# Inhibition of glycolysis-driven immunosuppression with a nano-assembly enhances response to immune checkpoint blockade therapy in triple negative breast cancer

Xijiao Ren[1], Zhuo Cheng[2], Jinming He[2], Xuemei Yao[2], Yingqi Liu[2], Kaiyong Cai[1], Menghuan Li ⬤[2] ✉, Yan Hu ⬤[1] ✉ & Zhong Luo ⬤[2] ✉

Immune-checkpoint inhibitors (ICI) are promising modalities for treating triple negative breast cancer (TNBC). However, hyperglycolysis, a hallmark of TNBC cells, may drive tumor-intrinsic PD-L1 glycosylation and boost regulatory T cell function to impair ICI efficacy. Herein, we report a tumor microenvironment-activatable nanoassembly based on self-assembled aptamer-polymer conjugates for the targeted delivery of glucose transporter 1 inhibitor BAY-876 (DNA-PAE@BAY-876), which remodels the immunosuppressive TME to enhance ICI response. Poly β-amino ester (PAE)-modified PD-L1 and CTLA-4-antagonizing aptamers (aptPD-L1 and aptCTLA-4) are synthesized and co-assembled into supramolecular nanoassemblies for carrying BAY-876. The acidic tumor microenvironment causes PAE protonation and triggers nanoassembly dissociation to initiate BAY-876 and aptamer release. BAY-876 selectively inhibits TNBC glycolysis to deprive uridine diphosphate N-acetylglucosamine and downregulate PD-L1 N-linked glycosylation, thus facilitating PD-L1 recognition of aptPD-L1 to boost anti-PD-L1 therapy. Meanwhile, BAY-876 treatment also elevates glucose supply to tumor-residing regulatory T cells (Tregs) for metabolically rewiring them into an immunostimulatory state, thus cooperating with aptCTLA-4-mediated immune-checkpoint inhibition to abolish Treg-mediated immunosuppression. DNA-PAE@BAY-876 effectively reprograms the immunosuppressive microenvironment in preclinical models of TNBC in female mice and provides a distinct approach for TNBC immunotherapy in the clinics.

Triple negative breast cancer (TNBC) is an aggressive breast cancer subtype with rapid disease progression and high risk of metastasis[1,2]. However, due to the lack of tumor-intrinsic actionable targets, TNBC treatment options are currently limited and their efficacy is still unsatisfactory[3,4]. Interestingly, immune checkpoint inhibition therapy (ICT), a form of immunotherapy that restores antitumor immunity by blocking negative immune checkpoints, has shown promise for effective TNBC treatment[5,6]. From a clinical perspective, TNBC presents a plethora of immunogenic traits including higher mutational burden and elevated programmed death ligand-1 (PD-L1) expression

[1]Key Laboratory of Biorheological Science and Technology, Ministry of Education, Chongqing University, Chongqing 400044, PR China. [2]School of Life Science, Chongqing University, Chongqing 400044, PR China. ✉e-mail: menghuanli@cqu.edu.cn; huyan303@cqu.edu.cn; luozhong918@cqu.edu.cn

levels, all of which may potentially improve the efficacy of ICT on TNBC patients[7,8]. Nevertheless, clinical data showed that only a limited number of TNBC patients could benefit from ICT, and the unsatisfactory response rate was due to the intrinsic immune escape capability of TNBC cells[9–11]. On one hand, PD-L1 on TNBC cell surface is usually highly glycosylated[12–14], which substantial reduces their binding affinity with ICIs to impair the antagonization performance[15]. On the other hand, TNBC microenvironment is populated by regulatory T cells (Tregs), which potently suppress the effector functions of cytotoxic T cells (CTLs) through a myriad of mechanisms including CTLA-4-mediated negative immunomodulation, secreting inhibitory cytokine, etc.[16,17]. Therefore, it is clinically desirable to develop ICT strategies with enhanced immune-checkpoint antagonization capacity as well as tumor immune microenvironment-reprograming ability to mount robust CTL-mediated antitumor immunity.

The oncogenesis process has profound impact on many fundamental behaviors of tumor cells, which are intrinsically linked with their malignant properties and therapeutic responses. Notably, tumor cells predominantly rely on glycolysis for ATP generation, which not only provides the bioenergy and biosynthesis substrates to support tumor progression, but also contributes to the development of the immunosuppressive TME[18–21]. Current insights reveal that glycolysis activity in tumor cells would generate D-fructose-6-phosphate intermediate product to synthesize UDP-GlcNAc through hexosamine biosynthetic pathway (HBP), which is the canonical glycan donor for glycosylation of proteins[22,23]. Meanwhile, hyperglycolytic tumor cells would also outcompete tumor-resident immune cells for sequestering glucose in TME, thus reinforcing Treg immunosuppressive functions to facilitate immunotolerance of tumor cells[24–26]. These insights highlighted the critical roles of tumor-intrinsic glycolysis in orchestrating the ICT resistance of TNBCs, whereas the selective inhibition of TNBC glycolysis could be a promising approach to remodel the immunosuppressive TME to cooperatively improve ICT efficacy[24,27,28].

Aptamers are a class of nucleic acid-based molecular therapeutics that can bind specifically to designated molecular targets[29,30], which offers opportunities to antagonize the immune checkpoint proteins for therapeutic purposes. From a biological perspective, the nucleic acid sequences in aptamers could form "stem-loop" structures through spatial folding to acquire antibody-like binding affinity[31,32]. Unlike antibodies that are highly susceptible to chemical alterations, the non-stem-loop sequences of aptamers could be modified in a versatile manner without affecting their biological functions, allowing the on-demand tailoring of their physicochemical features[33–35]. These emerging aptamer technologies provide simple and effective approaches to not only enhance the in vivo stability of aptamers but also endow additional functionalities[36,37]. For instance, there are already reports that chemically engineered aptamers could be complexed onto nanoparticle surface[37–39] or self-assemble into bio-functional nanostructures[35,40,41], which could enhance their resistance to nuclease-mediated aptamer degradation as well as integrate other therapeutic agents to create combinational therapies. Consequently, aptamer-based nanotherapeutics could be an alternative strategy to develop more advanced ICTs for overcoming the challenges in TNBC immunotherapy.

In this work, we present a TME-activatable aptamer-based nanoassembly for the efficient ICT treatment of TNBC, which is realized through tumor cell-selective glycolysis inhibition combined with bispecific immune checkpoint blockade. Based on that clinical evidence that TNBC cells frequently overexpress the glucose transporter 1 (GLUT1) for glucose import unlike other major tumor-residing immune cell populations[42,43], we select an experimental GLUT1 inhibitor BAY-876[44,45] for selective glycolysis inhibition of TNBC cells. Meanwhile, poly β-amino ester (PAE) is conjugated onto the 3′ terminal of PD-L1 and CTLA-4-antagonizing aptamers[46,47] as a multifunctional chemical handle (aptPD-L1 and aptCTLA-4). Due to the significant hydrophobicity of PAE under neutral pH, aptPD-L1 and aptCTLA-4 readily self-assemble into supramolecular nanostructures in biomimetic buffer solution, which not only stabilizes the aptamers but also enables the effective loading of hydrophobic BAY-876 molecules. After reaching the acidic TNBC microenvironment, the PAE handles rapidly become protonated and switch to a hydrophilic state, thus disrupting the hydrophobic/hydrophilic balance of the aptamer-PAE conjugates and triggering nanoassembly disintegration, eventually leading to the on-demand release of BAY-876, aptPD-L1 and aptCTLA-4 into TME (Fig. 1a). BAY-876-mediated GLUT1-inhibition in TNBC cells readily suppresses their glycolysis activities and further downregulates the glycosylation level of PD-L1 through Glu-F6P-UDP-GlcNAc axis, thus facilitating their recognition and binding by aptPD-L1 for effective PD-L1 blockade. Meanwhile, BAY-876 treatment also abolishes the over-competition of glucose by TNBCs and enhances glucose abundance in TME to metabolically reprogram immunosuppressive Tregs into an immunostimulatory state, which synergizes with aptCTLA-4-mediated CTLA-4 blockade to alleviate Treg-mediated CTL suppression (Fig. 1b). In vivo evaluations show that the bispecific aptamer nanoassemblies abolish TNBC growth through evoking robust CTL-mediated antitumor immune responses as well as prevent TNBC relapse and metastasis by establishing potent systemic anti-TNBC immune memory, which may provide insights on developing effective ICTs for TNBC treatment in the clinics.

## Results
### Preparation and characterization of DNA-PAE@BAY-876
To endow the PD-L1 and CTLA-4-targeting aptamers with self-assembly capacity and TME responsiveness, PAE, a biocompatible polymer with protonation capabilities[48,49], was modified onto the 3′-alkynyl terminals of the two aptamers as a biofunctional chemical handle. For this purpose, we first successfully synthesized N₃-PEG₂₀₀₀-PAE through the Michael addition reaction based on $N_3$-PEG$_{2000}$-ACA, 1,4-butanediol diacrylate (HDD) and 4,4′-trimethyldipiperidine (TDP) (Supplementary Figs. 1 and 2a–d) and thoroughly studied its chemical properties. Gel permeation chromatography analysis showed that the synthesized $N_3$-PEG$_{2000}$-PAE has an average molecular weight of 21,243 g·mol$^{-1}$ (Supplementary Fig. 2e). Meanwhile, selected aptamers were synthesized via established procedures with an average molecular weight of 21,412 g·mol$^{-1}$ and 21,633 g·mol$^{-1}$. To enable the non-invasive conjugation of PAE handles onto above aptamers, the aptamers were engineered with multiple T bases at their 3′ terminals as reactive sites. NUPACK analysis results showed no significant changes in secondary structures and ΔG values of the engineered aptamers compared to their original forms, indicating that the appending T bases did not affect their antagonizing function with cognate proteins (Fig. 2a, b). The PAE-based handles were conjugated onto aptamers via high-yield and non-invasive click reactions at a ratio of 1:1, which were denoted as aptPD-L1 and aptCTLA-4 for simplification. The affinity of aptPD-L1 and aptCTLA-4 was detected by a magnetic bead binding assay, for which biotin-modified mouse PD-L1 and mouse CTLA-4 were first separately conjugated onto streptavidin modified magnetic beads and then incubated with different concentrations of Cy5-aptPD-L1 or FAM-aptCTLA-4. The dissociation constants of aptPD-L1 and aptCTLA-4 to their cognate protein receptors were calculated to be 122.4 nM and 168.3 nM, respectively, supporting the binding affinity and specificity of the selected aptamers[46,47] with the designated immune checkpoints (Supplementary Fig. 4). In addition, using standard curve calibration method based on the fluorescence data, the grafting rates of aptPD-L1 and aptCTLA-4 with PAE handles were determined to be 81.24% and 84.79%, respectively (Supplementary Fig. 5a–d). The nanoassembly formation and spontaneous BAY-876 loading was realized through mixing aptPD-L1, aptCTLA-4 and BAY-876 in pH 7.4 PBS, for which the hydrophobic BAY-876 molecules were intercalated into the DNA-PAE cores through hydrophobic

interaction. TEM and SEM imaging results in Fig. 2c–e revealed that the aptamer-based nanoassemblies have a spherical shape with high monodispersity and uniformity, which was also consistently supported by the DLS analysis. Interestingly, DLS data revealed that the BAY-876 intercalation induced a slight size increase from 118 to 134 nm (Fig. 2f), which could be explained by the steric perturbation thereof. The integration of BAY-876 also elevated the zeta potential of the nanoassemblies from −15.87 to −9.5 mV (Fig. 2g). The BAY-876 loading was further investigated by UV-vis spectroscopy, which showed that the DNA-PAE@BAY-876 nanoassemblies have retained the characteristic peaks of both aptamers and BAY-876 (Fig. 2h), suggesting the successful construction of the composite self-assembly structures. Nevertheless, the characteristic peaks have slightly shifted compared with pristine DNA-PAE and BAY-876, which was ascribed to the π−π interaction in between[45] (Fig. 2h, i). Quantitative UV-vis spectroscopic analysis via a standard curve calibration approach showed that the relative encapsulation amount of BAY-876 in the final nanoassembly was around 51% and the encapsulation efficiency was around 0.1% (Supplementary Fig. 5e), which was attributed to the large molecular weight difference between BAY-876 and DNA-PAE. Further analysis showed that the aptamer-based nanoassembly was highly stable under neutral pH, which presented a critical micellar concentration (CMC) of around 7.26 μg·mL$^{-1}$ under pH 7.4 (Fig. 2j and Supplementary Fig. 6a) and could remain stable in

pH 7.4 PBS for more than 48 h (Fig. 2k and Supplementary Fig. 7), suggesting its feasibility for in vivo drug delivery applications.

Due to the rich protonatable tertiary amino groups in the PAE handles, they could switch to a hydrophilic positively-charged state under acidic pH, which would cause the rapid disintegration of the aptamer-based nanoassembly. Results of solubility analysis in Supplementary Fig. 3 suggested that N$_3$-PEG$_{2000}$-PAE showed low solubility in neutral or basic pH, but was more readily solubilized when the pH dropped below 7.0, immediately confirming its acid-triggerable solubilization capacity in TME-like conditions. Pyrene assay results indicated that the critical dissociation pH for the aptamer-based nanoassembly was around 6.91 (Fig. 2l and Supplementary Fig. 6b), which could be explained by the pH-induced solubility transition of PAE above. At the same time, DLS detection showed that the average hydrodynamic size of DNA-PAE increased from 122 to 328 nm (Fig. 2m) when pH dropped from 7.4 to 6.8. Meanwhile, the zeta potential of DNA-PAE increased from −16.26 to −2.99 mV (Supplementary Fig. 7a), accompanied with diminished Tyndall effect (Supplementary Fig. 7b). These observations indicated that the electrostatic repulsion between the charged PAE species under acidic conditions destabilized the DNA-PAE nanoassemblies. This is also consistent with our TEM observations that the DNA-PAE@BAY-876 nanoassemblies rapidly disintegrated under pH 6.8 (Fig. 2c, d). We further investigated the pH-actuated dissociation behavior of the DNA-PAE@BAY-876 nanoassembly on

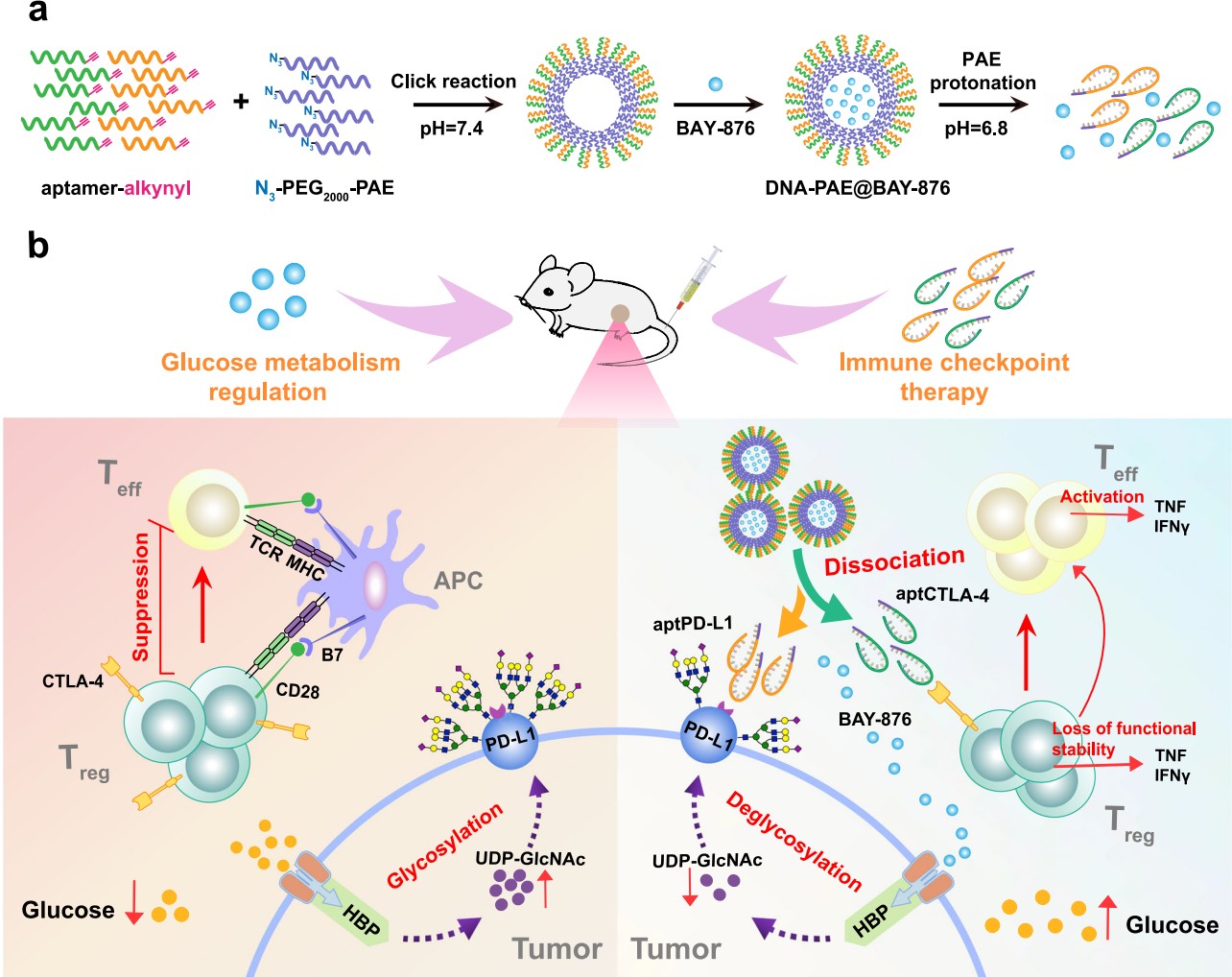

**Fig. 1 | DNA-PAE@BAY-876 nanoassemblies reprogram immunosuppressive TME through TNBC-selective glycolysis inhibition and bispecific PD-L1/CTLA-4 antagonization, leading to enhanced ICT efficacy. a** Synthetic route of DNA-PAE@BAY-876 nanoassemblies through the spontaneous organization of the molecular building blocks. **b** Schematic illustration of DNA-PAE@BAY-876-mediated TME remodeling and the associated immunostimulatory mechanisms.

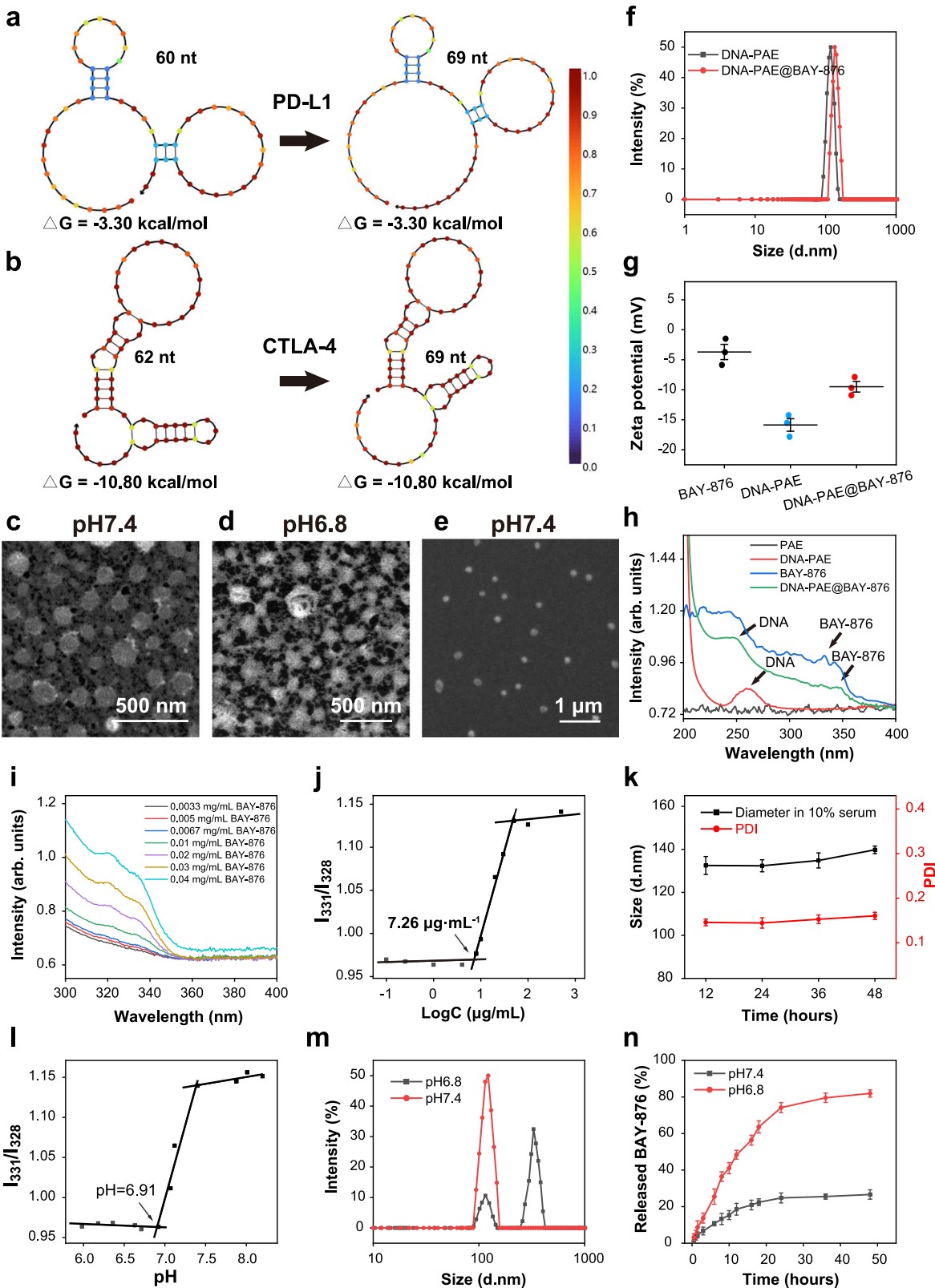

BAY-876 release kinetics (Fig. 2n). Remarkably, the DNA-PAE@BAY-876 nanoassemblies showed low drug release rate under pH 7.4, of which the final BAY-876 leakage ratio was only around 26% after 48 h of incubation. Contrastingly, the BAY-876 release rate increased significantly under pH 6.8, and the final drug release rate reached around 82% after 48 h incubation. The findings indicated that the DNA-PAE@BAY-876 nanoassembly has high stability under physiological conditions and releases the payload under acidic conditions for potentiating TME-responsive drug delivery.

## Targeting ability of DNA-PAE@BAY-876 in vitro

To test the bispecific binding capability of the DNA-PAE@BAY-876 nanoassemblies after mild acidity-mediated activation, we comprehensively investigated the cell binding and subcellular distribution of

**Fig. 2 | Physical and chemical characterization of DNA-PAE@BAY-876 nanoassemblies. a, b** Secondary structure of PD-L1 and CTLA-4-antagonizing aptamers by NUPACK analysis. Color intensity indicates base-pairing probability at equilibrium. **c, d** TEM image of DNA-PAE@BAY-876 nanoassemblies with different pH. **e** SEM image of DNA-PAE@BAY-876 nanoassemblies at pH 7.4. **f** Size changes of DNA-PAE nanoassemblies before and after drug loading. **g** Zeta potential changes of DNA-PAE nanoassemblies before and after drug loading. **h** UV-vis spectroscopic features of DNA-PAE nanoassemblies before and after drug loading. **i** Ultraviolet absorption of BAY-876 at different concentrations. **j** CMC of DNA-PAE nanoassemblies under pH 7.4. **k** Stability of DNA-PAE@BAY-876 nanoassemblies in PBS supplemented with 10% serum. **l** Critical pH value of DNA-PAE nanoassemblies showing their acidity-triggerable dissociation. **m** Size changes of DNA-PAE nanoassemblies at pH 7.4 and 6.8. **n** Release kinetics of BAY-876 from DNA-PAE@BAY-876 nanoassemblies at pH 7.4 and 6.8. TEM and SEM experiments in panels **c**–**e** were repeated three times independently with similar results. The characterizations in panels **f, h, i, j, l** and **m** were repeated three times independently with similar results. Data are presented as mean values ± SEM ($n = 3$ independent experiments for panels **g, k** and **n**). Source data are provided as a Source Data file.

the engineered aptamers under different incubation conditions. Quantitative flow cytometric data first revealed that aptPD-L1 and aptCTLA-4 with PAE handles have good binding affinity with their cognate molecular markers on targeted cell populations consistent with the NUPACK analysis, of which the binding performance was comparable to the corresponding antibodies (Fig. 3a, b). Furthermore, we established co-incubation systems comprising 4T1 cells and splenic immune cells and confirmed that PAE-modified aptPD-L1 and aptCTLA-4 could preferentially complex with 4T1 cells and Tregs, respectively, again validating the application potential of the PAE-modified aptamers for robust ICT against TNBCs (Supplementary Fig. 9). Specifically, we observed that the amount of aptPD-L1 bound to 4T1 cell surface was higher than HC11 cells, attributing to the upregulated PD-L1 expression level in 4T1 cells (Supplementary Fig. 10). Similar trends were also detected for aptCTLA-4, which preferentially bond to CTLA-4-overexpressing activated Tregs rather than their non-activated counterparts. In addition, flow cytometry and fluorescence spectroscopic data showed that the pH variations had no significant effect on the binding ability of the two aptamers (Supplementary Fig. 11), supporting their therapeutic robustness in the acidic environment. We then incubated 4T1/HC11 cells with DNA-PAE nanoassemblies under pH 7.4 or pH 6.8 to investigate their nano-biointeraction via confocal laser microscopy (CLSM) (Fig. 3c). Interestingly, abundant Cy5-labeled aptPD-L1 bond to 4T1 cell surface under pH 6.8, while the majority of aptPD-L1 was attached to 4T1 nuclei when incubated under pH 7.4. The distinct aptPD-L1 distribution patterns were immediate evidence for the acidity-responsiveness of DNA-PAE nanoassemblies. Under the TME-like acidic pH of 6.8, DNA-PAE nanoassemblies were rapidly dissociated into free aptamer-PAE conjugates, allowing their direct complexation with PD-L1 on 4T1 cytoplasmic membrane. However, under the physiological pH of 7.4, the DNA-PAE nanoassemblies were endocytosed by 4T1 cells as a whole and subsequently dissociated in the acidic tumor lysosomes, which would then complex with the PD-L1 in tumor nucleus[15]. Furthermore, PD-L1-negative HC11 cells generally showed low fluorescence deposition after treatment with both DNA-PAE nanoassemblies and AbPD-L1, which was in line with their intrinsically low PD-L1 expression status and again validated the therapeutic mechanism of DNA-PAE. To further elucidate the uptake mechanism of the DNA-PAE nanoassemblies, we used siRNA to knock down the PD-L1 expression on the 4T1 cell surface[50] and found that this caused a significant reduction in their uptake amount under neutral pH, showing that the DNA-PAE nanoassembly endocytosis by 4T1 cells under neutral pH was mediated by the interaction between aptPD-L1 and membrane PD-L1 (Supplementary Fig. 12). These observations collectively demonstrate that the DNA-PAE nanoassembly maintained the molecular affinity of pristine aptamers while further enabling their specific binding with the cognate receptors on cell surface through acidity-triggered dissociation.

In addition, dissociation of the DNA-PAE@BAY-876 nanoassembly in response to acidic treatment would also cause the spontaneous release of BAY-876. In line with the upregulated GLUT1 expression in 4T1 cells compared with HC11 cells, we found that DNA-PAE@BAY-876 treatment substantially reduced active GLUT1 levels in 4T1 cells (Fig. 3d). Furthermore, MTT results showed that DNA-PAE@BAY-876

could inhibit tumor proliferation in a dose dependent manner, of which the 4T1 cell viability gradually dropped to around 53% under the BAY-876 dose of 100 nM, while HC11 cell viability still maintained around 90% at the same dosing condition (Fig. 3e, f). Consequently, the final BAY-876 dose was determined at 100 nM for the subsequent experiment. Alternatively, we also measured the glucose levels and lactic acid secretion in DNA-PAE@BAY-876-treated 4T1 cells and found that they were both substantially suppressed (Supplementary Fig. 13), evidently supporting the glycolysis inhibition by DNA-PAE@BAY-876 through blocking GLUT1-mediated glucose uptake. The above results all demonstrated that DNA-PAE@BAY-876 could selectively inhibit glycolysis in GLUT1-overexpressing 4T1 cells while eliciting no obvious negative impact on GLUT1-normal cell populations.

To further investigate the cellular specificity and safety of the BAY-876-dependent glycolysis inhibition strategy in vitro, we isolated representative immune cell populations including CD4+ T cells, CD8+ T cells, Tregs, M1 macrophages and M2 macrophages and incubated them with BAY-876 at a therapeutically relevant concentration of 100 nM. MTT assay showed no significant changes in the viability of all major immune cell populations after BAY-876 treatment regardless of their activation status (Fig. 3g–i and Supplementary Fig. 14). Consistent with the glucose metabolic patterns in different cell populations in vitro, immunofluorescence imaging and quantitative data by flow cytometry on 4T1 tumors extracted from Balb/c mouse models indicated that tumor-infiltrating CD45+ lymphocytes were generally associated with higher GLUT3 expression levels, while tumor cells tended to express high levels of GLUT1 for glucose uptake (Fig. 3j, k and Supplementary Fig. 15). It is thus anticipated the that GLUT1-inhibition function of BAY-876 would allow 4T1 cell-specific glycolysis inhibition. For this purpose, we established co-incubation system consisting of 4T1 cells and mouse splenic immune cells and treated them with BAY-876 or KL-11743[51], a GLUT1/GLUT3 dual inhibitor. Notably, treating the co-culture system with 100 nM BAY-876 caused significant suppression of 4T1 cells while expanding the immune cell populations associated with the adaptive antitumor immunity including DCs, M1 macrophages, CD4+ T cells and CD8+ T cells, on account of the capacity of BAY-876 to selectively inhibit GLUT1 to block tumor cell-intrinsic glucose uptake. Contrastingly, both tumor cells and immune cells were substantially suppressed by KL-11743 treatment due to the simultaneous inhibition of GLUT1 and GLUT3, leading to universal glucose uptake blockade (Fig. 3l, m and Supplementary Figs. 16–18). The evidence above supported the therapeutic capacity of the BAY-876-containing nanoassembly to rebalance glucose competition between TNBC cells and immune cells for mounting adaptive antitumor immune responses.

## Antitumor effects of DNA-PAE@BAY-876 in vitro

The antitumor potency of the DNA-PAE@BAY-876 nanoassemblies using co-incubation system comprising 4T1 cells and splenic immune cells. According to flow cytometric analysis, western blotting and MTT assay (Fig. 4a–c and Supplementary Figs. 19 and 20a), DNA-PAE@BAY-876 + pH 6.8 showed the most pronounced antitumor effect among all groups, of which the survival rate of 4T1 cells was only around 24%. According to the optical microscopic analysis results, DNA-PAE@BAY-

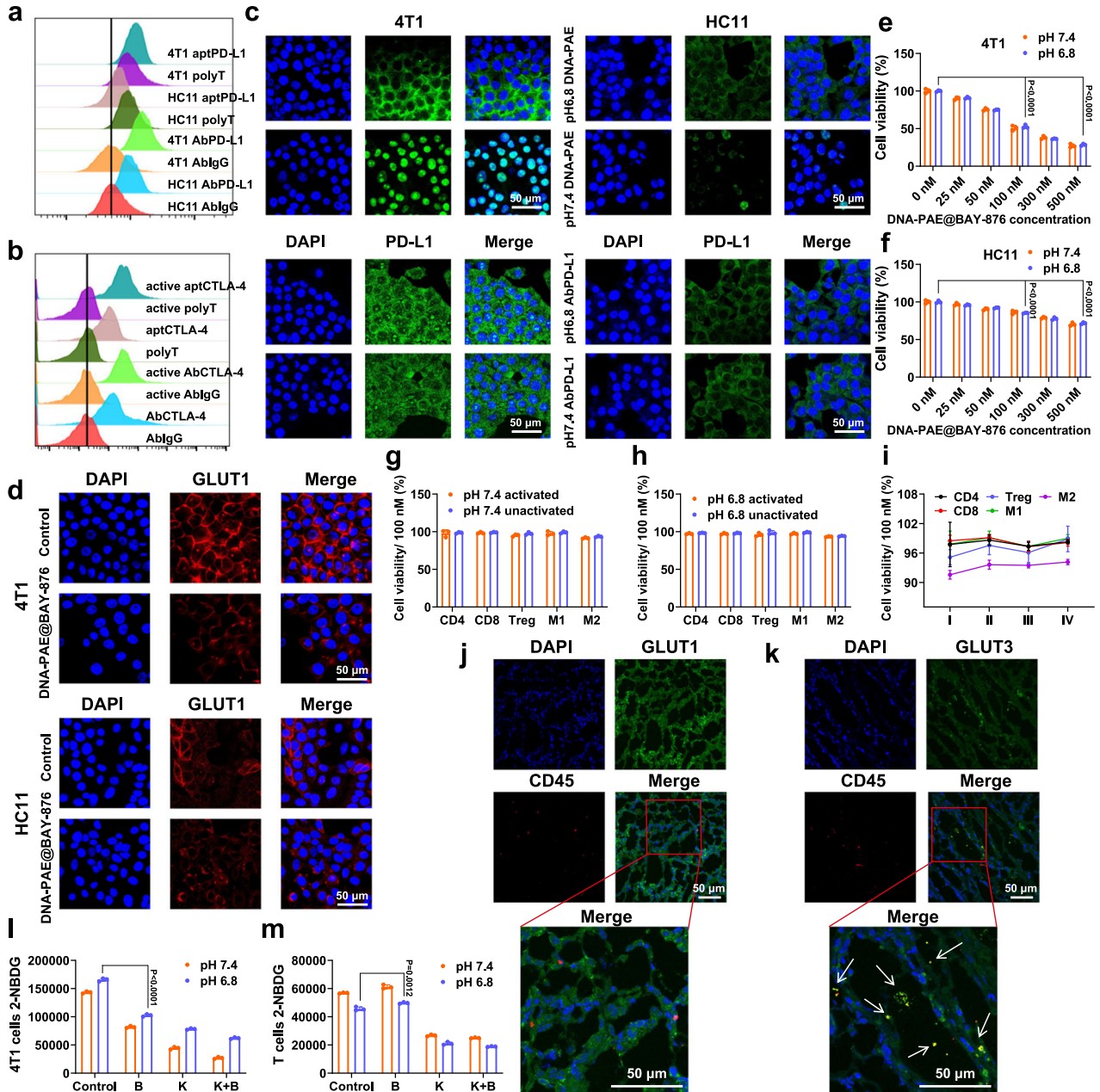

**Fig. 3 | Evaluation on the TNBC-targeting effect of DNA-PAE@BAY-876 in vitro.**
**a** Flow cytometric analysis on the cell binding behavior of aptPD-L1 to 4T1/
HC11 cells under pH 7.4. **b** Cell binding behavior of aptCTLA-4 to activated/inacti-
vated T cells under pH 7.4. **c** Interaction between 4T1 or HC11 cells and DNA-PAE
nanoassemblies under different environmental pH conditions at 37 °C for 2 h.
**d** DNA-PAE@BAY-876-medited GLUT1 inhibition in 4T1 cells and HC11 cells at 37 °C
for 24 h. **e, f** Dose-dependent toxicity of DNA-PAE@BAY-876 to 4T1 cells and
HC11 cells at 37 °C for 24 h. **g, i** Toxicity of BAY-876 to major immune cell popu-
lations at different pH conditions. (I) pH 7.4+activation, (II) pH 7.4+inactivation, (III)

pH 6.8+activation, (IV) pH 6.8+inactivation. **j, k** Immunofluorescence analysis of
GLUT1 and GLUT3 in 4T1 tumors. **l, m** Glucose uptake of 4T1 cells and T cells in the
co-incubation system after treatment with BAY-876 or KL-11743. Flow cytometry
and immunofluorescence experiments in panels **a–d**, **j** and **k** were repeated three
times independently with similar results. Data are presented as mean values ± SEM
($n$ = 3 biologically independent samples for panels **e–i**, **l** and **m**). Statistical analysis
was carried out via two-way ANOVA method. Source data are provided as a Source
Data file.

876 substantially inhibited inhibition of tumor cell invasion while
promoting immune cell migration, again validating the nanoassembly-
enabled impairment of tumor cell activity and immunosuppression
(Supplementary Fig. 20b, c). Meanwhile, 2-NBDG assay showed that
the DNA-PAE@BAY-876 nanoassemblies also significantly reduced the
glucose uptake by 4T1 cells while enhancing the glucose uptake by
immune cell populations (Fig. 4d, e), which is consistent with the
intrinsic tumor specificity of BAY-876 as well as supports our
hypothesis that the aptamer-based nanoassembly could improve the
delivery efficiency of hydrophobic BAY-876 molecules.

In line with the potent antitumor efficacy and glucose meta-
bolism modulatory effect of DNA-PAE@BAY-876 nanoassemblies
under pH 6.8, we found that DNA-PAE@BAY-876 under pH 6.8
showed the greatest enhancement on T cell-mediated antitumor
immune responses. As shown in the Fig. 4f, CD4+ and CD8+ T cell
ratios in the DNA-PAE@BAY-876 with pH 6.8 have increased by
24.3% and 10.5% compared with the control group. Meanwhile, DC
and M1 macrophage frequencies in the DNA-PAE@BAY-876 with
pH 6.8 group have increased by 35.38% and 33.89%, respectively
(Supplementary Fig. 21), while the ratio of IFN-γ+CD8+ T cells has

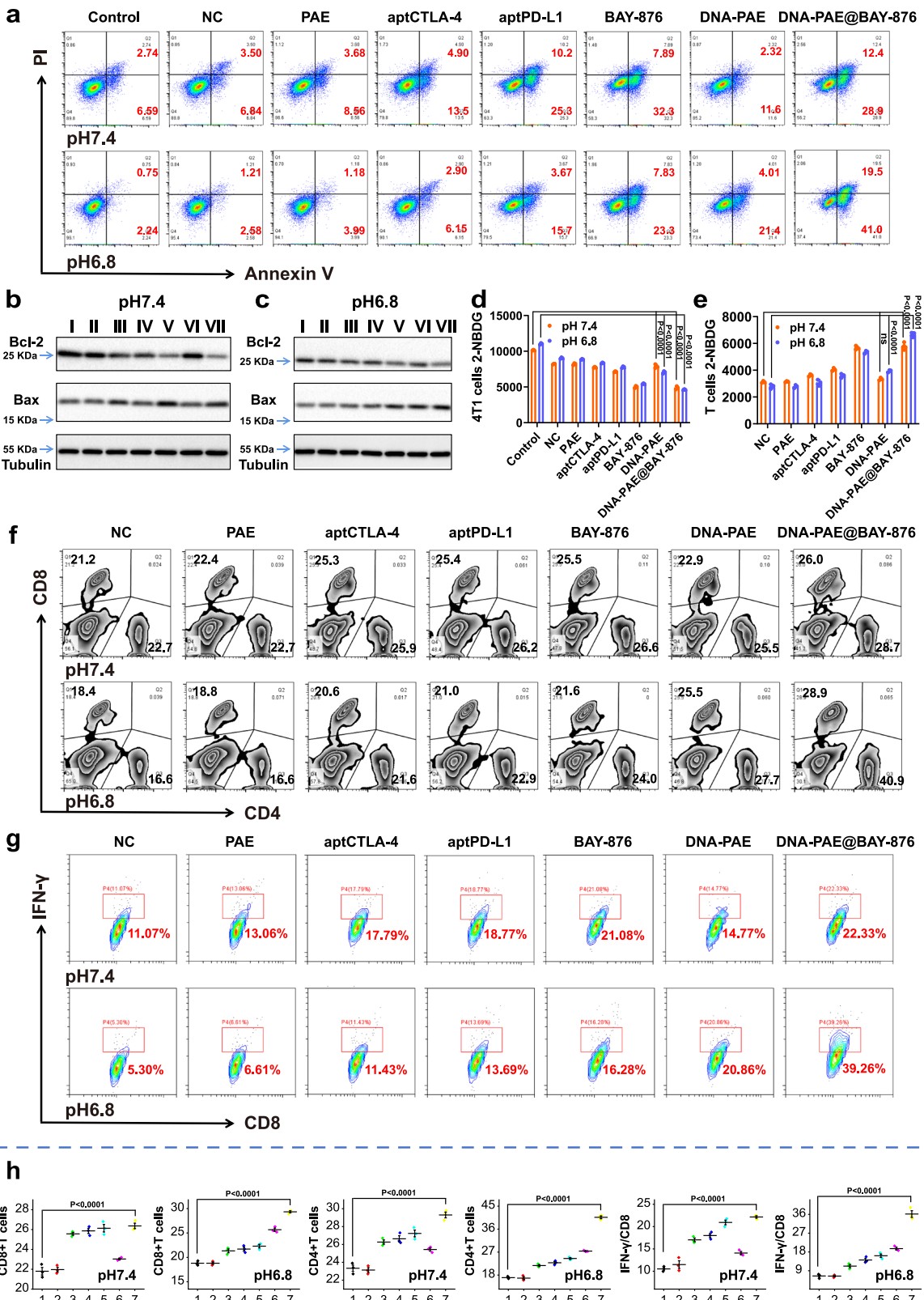

also increased by 33.96% (Fig. 4g). Furthermore, quantitative ELISA test revealed that immune cells in the DNA-PAE@BAY-876 with pH 6.8 had the highest secretion levels of pro-inflammatory cytokine IFN-γ, chemokine CXCL10, TNF-α and granzyme B (GZMB), while the secretion level of anti-inflammatory cytokine IL-10 was the lowest (Supplementary Fig. 22). These observations immediately suggested that the DNA-PAE@BAY-876 nanoassemblies could be

activated in TME-like conditions to boost T cell-mediated antitumor effect.

## Impact of DNA-PAE@BAY-876 on 4T1-intrinsic PD-L1 glycosylation

To elucidate the molecular mechanism regarding the immunostimulatory effect of DNA-PAE@BAY-876, we first studied its impact on the

**Fig. 4 | DNA-PAE@BAY-876 treatment mounted potent antitumor effect in 4T1-splenic immune cell co-incubation system. a** Apoptosis ratio of 4T1 cells in the co-incubation system after different treatment at 37 °C for 24 h. Apoptosis effect of 4T1 cells with pH 7.4 (**b**) and pH 6.8 (**c**) by western blotting at 37 °C for 24 h. (I) NC, (II) PAE, (III) aptCTLA-4, (IV) aptPD-L1, (V) BAY-876, (VI) DNA-PAE, (VII) DNA-PAE@BAY-876. Note: the molecular weight of Bcl-2 is 26 KDa; the molecular weight of Bax is 21 KDa; the molecular weight of Tubulin is 50 KDa. **d, e** Glucose uptake by 4T1 cells or T cells after different treatments under co-culture condition at 37 °C for 24 h. **f, g** Flow cytometric analysis on the expansion of CD4+/CD8+ T cells, IFN-γ +CD8+ T cells in the co-incubation system after different treatment at 37 °C for 24 h. **h** Statistical analysis of data shown in panels **f** and **g** by flow cytometry. (1) NC, (2) PAE, (3) aptCTLA-4, (4) aptPD-L1, (5) BAY-876, (6) DNA-PAE, (7) DNA-PAE@BAY-876. Flow cytometry and western blot experiments in panels **a**–**c**, **f** and **g** were repeated three times independently with similar results. Data are presented as mean values ± SEM (*n* = 3 biologically independent samples for panels **d**, **e** and **h**). Statistical analysis was carried out via two-way ANOVA method. Source data are provided as a Source Data file.

immunosuppressive potential of 4T1 cells. Indeed, glycosylation is one of the most common post-transcriptional modifications of protein in various cells[52], while some studies indicated that excessive glycosylation of PD-L1 on triple negative breast cancer is one of the reasons leading to ICT failure in the clinics[9,14]. Consistent with their observations, we detected that PD-L1 receptors in 4T1 cells have high glycosylation levels (Fig. 5b, c), which readily explained the low 4T1 inhibitory efficacy of pristine aptPD-L1 in the efficacy evaluation above. Western blot analysis showed that the PD-L1 glycosylation levels decreased significantly after BAY-876 and DNA-PAE@BAY-876 treatment, substantiating the negatively regulatory role of BAY-876 on the glycosylation of tumor-intrinsic PD-L1 as well as its potential contribution to ICT enhancement (Fig. 5b, c and Supplementary Fig. 23). The BAY-876-induced biochemical changes in 4T1 cells were further analyzed to study the underlying mechanisms. Specifically, HBP is an important branch of glycolysis pathway responsible for converting Glu-F6P into the canonical glycan donor UDP-GlcNAc (Fig. 5f)[22,23]. Interestingly, western blotting and ELISA assay results collectively demonstrated that the BAY-876-induced PD-L1 deglycosylation could be reversed by adding additional F6P or UDP-GlcNAc, evidently confirming that BAY-876 reduced PD-L1 glycosylation levels in 4T1 cells through negatively regulating the Glu-F6P-UDP-GlcNAc axis (Fig. 5d, e and Supplementary Fig. 24). However, it is also noteworthy that even a large UDP-GlcNAc dose at around 0.5 mM could only induce partial recovery of PD-L1 glycosylation levels. This is understood that protein glycosylation in tumor cells is largely sustained by the glycolysis activities both in terms of UDP-GlcNAc precursors[20,23,53,54] and metabolite supply[55,56] including ATP and NADPH, both of which are significantly suppressed by the BAY-876-mediated blockade of glucose uptake. This is also supported by the quantitative analysis on the biochemical changes in 2-NBDG-treated 4T1 cells, which revealed a positive correlation between 2-NBDG dosage and F6P/UDP-GlcNAc levels (Fig. 5g and Supplementary Fig. 25). Additionally, we have also tested the effectiveness of BAY-876-mediated tumor-intrinsic PD-L1 deglycosylation by comparing to PNGase F treatment, which is currently the most effective enzymatic methods for removing N-linked glycans from proteins[15]. Notably, the PD-L1 deglycosylation efficacy of BAY-876 and DNA-PAE@BAY-876 was at a comparable level to PNGase F (Supplementary Fig. 26). These data supported our hypothesis that the DNA-PAE@BAY-876 could negatively impact the Glu-F6P-UDP-GlcNAc axis via blocking GLUT1-mediated glucose intake, offering a highly selective and effective strategy deglycosylation of tumor-intrinsic PD-L1.

To test if the BAY-876-induced PD-L1 deglycosylation could facilitate the aptPD-L1-mediated immune checkpoint blockade, we carried out comprehensive flow cytometric analysis to determine the changes in aptPD-L1 and 4T1 cell binding performance after different treatment. Normally, immune checkpoint inhibitors such as antibodies or aptamers are usually obtained by in vitro screening methods. For instance, the aptPD-L1 commonly used in biomedical research is obtained by REIM-SELEX[46]. Consequently, these PD-L1-inhibiting agents are prone to showing reduced antagonizing efficiency with glycosylated PD-L1 receptors. As shown by the flow cytometric analysis, BAY-876 and DNA-PAE@BAY-876 treated 4T1 cells both showed enhanced binding capacity with aptPD-L1, leading to more effective PD-L1 antagonization in vitro. The trends revealed by the flow cytometric data were also consistently supported by confocal microscopic visualization results (Fig. 5h–k and Supplementary Fig. 27). The observations above immediately suggested that the DNA-PAE@BAY-876 nanoassembly could enhance the PD-L1 antagonization performance of co-delivered aptPD-L1 against 4T1 cells in a highly coordinated manner, thus ameliorating the tumor cell-induced immunosuppression for more effective immunotherapy (Fig. 5a).

## DNA-PAE@BAY-876 abolishes Treg-mediated immunosuppression

Tumor-residing Tregs are the major immunosuppressive cell populations in TME and play critical roles in impairing CTL-mediated antitumor immune reactions[16,24]. Interestingly, recent studies increasingly revealed that the CTL-suppressing functions of Tregs are profoundly affected by their glucose metabolism state[24,57,58]. Specifically, high glucose environment would cause Treg destabilization and alleviate their immunosuppressive capacity[24]. Consequently, it is anticipated that the DNA-PAE@BAY-876-mediated glycolysis blockade of 4T1 cells would increase glucose abundance in TME to metabolically destabilize tumor-residing Tregs, which would further cooperative with the aptCTLA-4-mediated immune checkpoint blockade to abolish Treg-dependent CTL suppression (Fig. 6a). Then we isolated Tregs from the nanoassembly-treated 4T1/splenic cell co-incubation system to analyze their phenotypical and functional properties. Remarkably, Tregs in the pH 6.8 DNA-PAE@BAY-876 group showed the lowest expression levels of Foxp3 and CD25+CTLA-4+, which was approximately 30% lower than that in the control group and suggested the significant reduction in their immunosuppressive capacity (Fig. 6b, c). The nanoassembly-induced reduction in Treg-mediated immunosuppression was further studied by monitoring the effector function of CD8+ T cells. The pH 6.8 DNA-PAE@BAY-876 group had the most pronounced expansion of CD8+ T cell population among all groups, which was 15.24% higher than the control group (Supplementary Fig. 28a). Moreover, we observed that the Tregs in the pH 6.8 DNA-PAE@BAY-876 group also showed the lowest suppression capacity to CD8+ T cells (Fig. 6d, e and Supplementary Fig. 28b, c), suggesting substantial enhancement in their cytotoxic capacity. The same treatment also induced enhanced secretion of IFN-γ as well as reducing the secretion of anti-inflammatory cytokine IL-10 (Fig. 6f, g), indicating the successful activation of adaptive immune responses. To illustrate the immunostimulatory potential of the nanoassemblies, we tested the frequency of IFN-γ CD3+ and IL-10 CD3+ T cells in the co-culture system of 4T1 cells and splenic immune cells after various treatment. Compared with the Control group, the population of IFN-γ CD3+ T cells has increased by 28.72% in the pH 6.8 + DNA-PAE@BAY-876 group, while the IL-10+CD3+ T cell population has decreased by 28.83% (Supplementary Fig. 29). In line with the potent immunostimulatory capacity of DNA-PAE@BAY-876, cytotoxicity assay results demonstrated that pH 6.8 + DNA-PAE@BAY-876 group had the strongest antitumor ability, of which the 4T1 cell viability in the 4T1 + activated Treg + activated CD8+ T cell co-incubation system was only around 23% (Fig. 6h–j). Based on the results above, the DNA-PAE@BAY-876 nanoassembly could effectively relieve the Treg-mediated CTL suppression and boost their tumor cell killing activity.

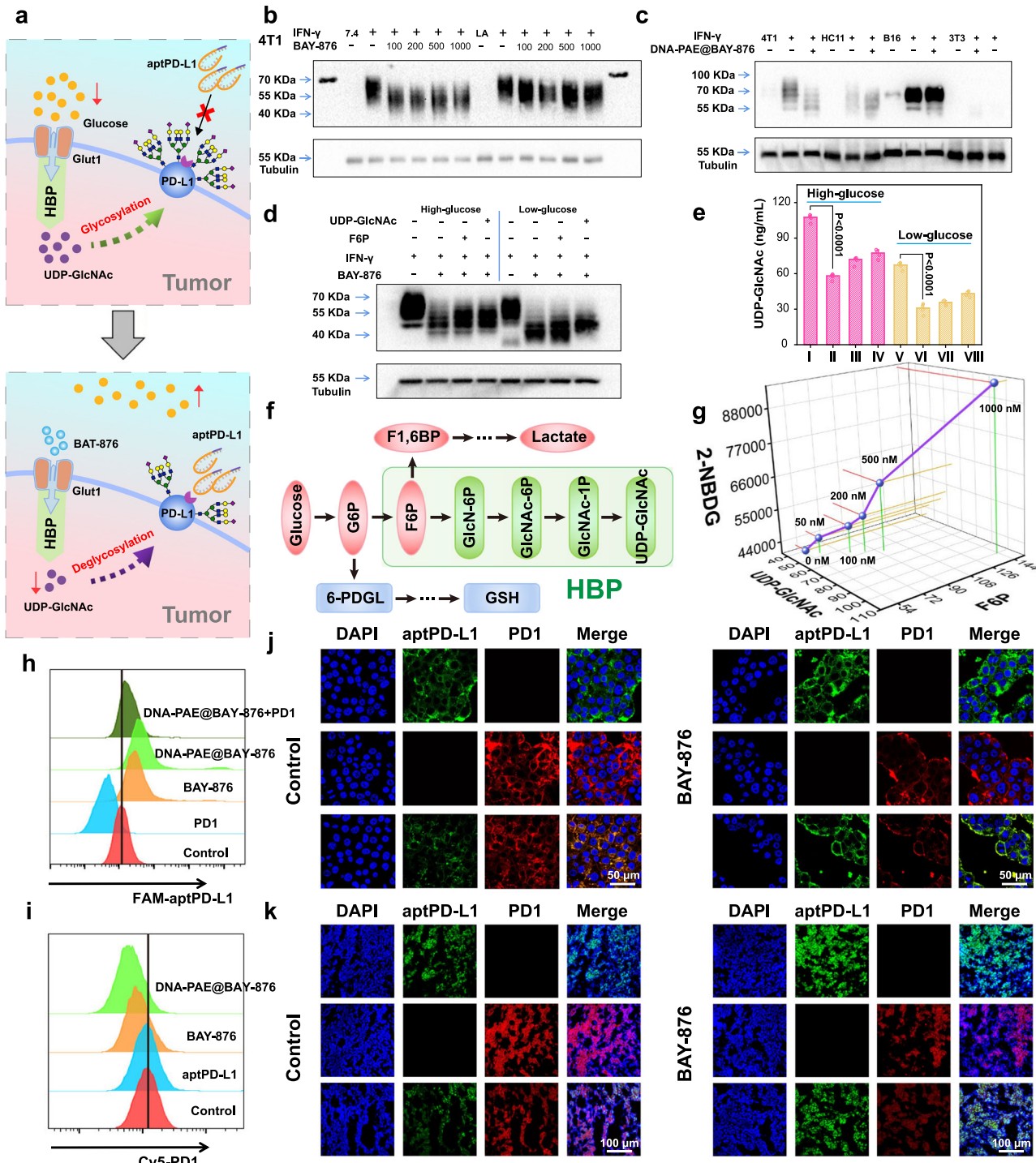

**Fig. 5 | Nanoassembly-enhanced 4T1 cell recognition and binding by aptPD-L1.**
**a** Schematic illustration on the nanoassembly-mediated metabolic rewiring of 4T1 cells for enhanced ICT. **b** Evaluation on the dose-dependent impact of BAY-876 on PD-L1 glycosylation in 4T1 cells with pH 7.4/6.8 at 37 °C for 24 h. The concentration of IFN-γ was 1 μg·mL⁻¹. Note: the molecular weight of PD-L1 is 33–70 KDa; the molecular weight of Tubulin is 50 KDa. **c** Impact of DNA-PAE@BAY-876 on the PD-L1 glycosylation level in multiple cell types at 37 °C for 24 h. The concentration of DNA-PAE@BAY-876 was 100 nM and the concentration of IFN-γ was 1 μg·mL⁻¹. **d** Impact of UDP-GlcNAc and F6P on the PD-L1 glycosylation levels in 4T1 cells after BAY-876 treatment with high/low-glucose media. The concentration of BAY-876, UDP-GlcNAc and IFN-γ was 100 nM, 0.1 mM and 1 μg·mL⁻¹, respectively. **e** UDP-

GlcNAc abundance in 4T1 cells under different conditions. (I, V) Control; (II, VI) BAY-876; (III, VII) BAY-876 + F6P; (IV, VIII) BAY-876 + UDP-GlcNAc. **f** Schematic illustration of Hexosamine Biosynthesis Pathway (HBP). **g** 3D plot on the correlation between UDP-GlcNAc and F6P with 2-NBDG after treatment by BAY-876 under graded concentrations. **h–k** Impact of BAY-876 on the competitive combination between aptPD-L1/PD1 with PD-L1 on 4T1 cell surface. Western blot, flow cytometry and immunofluorescence experiments in panels **b–d** and **h–k** were repeated three times independently with similar results. Data are presented as mean values ± SEM (*n* = 4 biologically independent samples for panel **e**). Statistical analysis was carried out via two-way ANOVA method. Source data are provided as a Source Data file.

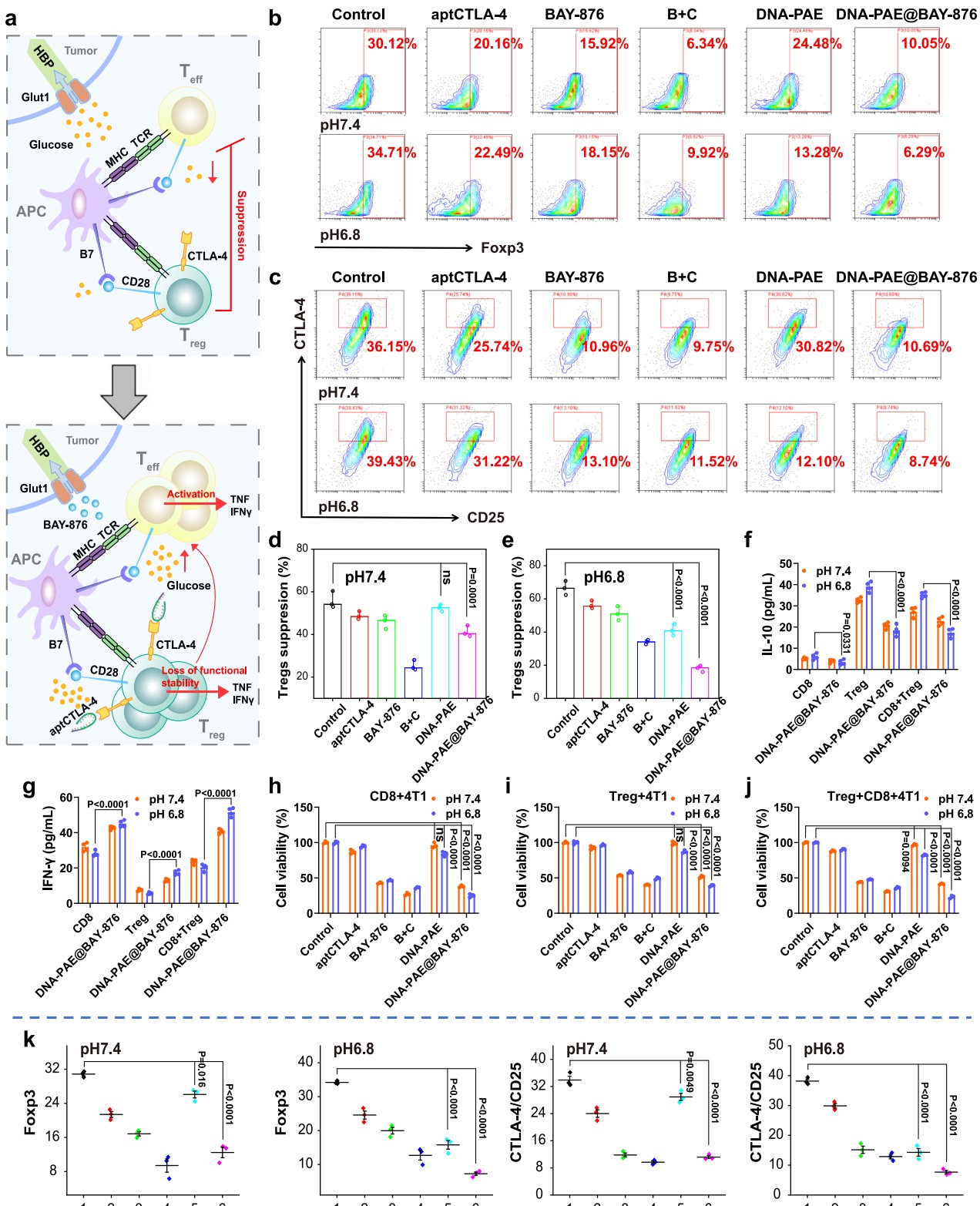

**Fig. 6 | Nanoassembly-mediated abolishment of Treg-mediated immunosuppression. a** Schematic illustration of the nanoassembly-mediated reprogramming of immunosuppressive Tregs. **b** Foxp3 expression in Tregs after different treatments at 37 °C for 24 h. **c** CTLA-4 and CD25 expression on Tregs after different treatments at 37 °C for 24 h. **d**, **e** Quantitative profiling of Treg-mediated CTL suppression after different treatments with pH 7.4/6.8 via measuring IFN-γ levels. Tregs suppression (%) = (IFN-γCD8-IFN-γTreg+ CD8)/IFN-γCD8 (**f**, **g**) ELISA assay on the secretion levels of key immune-related cytokines by immune cells treated with DNA-PAE@BAY-876 under different co-culture conditions at 37 °C for 24 h.

**h**–**j** Viability of 4T1 cells under different co-culture conditions at 37 °C for 24 h. **k** Statistical analysis of data shown in panels **b** and **c** by flow cytometry. (1) Control, (2) aptCTLA-4, (3) BAY-876, (4) B + C, (5) DNA-PAE, (6) DNA-PAE@BAY-876. B + C indicates the mixture of BAY-876 and aptCTLA-4. Flow cytometry experiments in panels **b** and **c** were repeated three times independently with similar results. Data are presented as mean values ± SEM (*n* = 3 biologically independent samples for panels **d**–**k**). Statistical analysis in panels **f** and **g** was carried out via one-way ANOVA method while two-way ANOVA method was used for panels **d**, **e** and **h**–**k**. Source data are provided as a Source Data file.

## Therapeutic evaluation of DNA-PAE@BAY-876 nanoassembly in vivo

On the basis of the immunostimulatory effects of the DNA-PAE@BAY-876 nanoassembly in vitro, we further investigated its antitumor efficacy in vivo on 4T1-tumor bearing Balb/c mouse models. We first investigated the pharmacokinetics properties of DNA-PAE@BAY-876 in Balb/c mice and found that the nanoassemblies could substantially enhance the circulation stability of the hydrophobic BAY-876 molecules, of which the blood half-life increased from 3 to 8 h (Supplementary Fig. 30a). In addition, the DNA-PAE@BAY-876 showed preferential deposition in the tumors owing to their tailored nanomorphology and tumor-responsive properties (Supplementary Fig. 30b). In addition, aptCTLA-4 and aptPD-L1 were labeled with FITC to synthesize DNA-PAE@BAY-876, and the fluorescence confocal microscopic data showed that aptCTLA-4 and aptPD-L1 successfully targeted CD45+ lymphocytes and 4T1 tumor cells, respectively (Supplementary Fig. 31a, b). Meanwhile, the concentration of BAY-876 in the tumor microenvironment was determined to be 126 nM using the liquid chromatography-triple quadrupole mass spectrometer (Supplementary Fig. 31c–e). The above experimental data supported the tumor-targeted delivery efficacy of the DNA-PAE@BAY-876 nanoassembly in vivo. Then, the mouse models were divided into seven groups and treated by NC, PAE, aptCTLA-4, aptPD-L1, BAY-876, DNA-PAE and DNA-PAE@BAY-876 at 2-3 days interval, respectively. The treatment period would last 21 days (Fig. 7a). Notably, treating tumor cells with PAE induced no significant impact on tumor growth, thus excluding its potential therapeutic contribution. Treating 4T1-tumor bearing mice with aptCTLA-4 and aptPD-L1 induced moderate tumor inhibition effect, of which the final average tumor volume was 1189.11 and 1082.36 mm$^3$, respectively. The modest antitumor responses of aptCTLA-4 and aptPD-L1 were consistent with the in vitro evaluations and immediately suggested the intrinsic resistance of TNBC to ICT. On the other hand, the BAY-876 group also showed significant tumor inhibition effect with a final average tumor volume of around 446.85 mm$^3$, attributing to the BAY-876-mediated glycolysis inhibition. DNA-PAE@BAY-876 showed the most pronounced tumor inhibition effect compared with all other groups, which completely abolished tumor growth with a final average tumor volume of around 69.53 mm$^3$. Analysis on tumor weight in different groups revealed a similar trend, for which the DNA-PAE@BAY-876 group presented the lowest tumor weight (Fig. 7e), which were evidently superior than all other groups. The tumor tissues were also extracted for H&E and TUNEL staining to analyze the histological details of individual treatment, and tumors from the DNA-PAE@BAY-876 group showed the largest apoptotic dead cell population (Fig. 7g, h and Supplementary Fig. 32) consistent with western blotting analysis (Supplementary Fig. 33), which again validated the potent antitumor potency of DNA-PAE@BAY-876 nanoassemblies. In addition to the therapeutic evaluations, we have also thoroughly investigated the safety of above seven groups in vivo. Notably, all sample groups induced no significant changes in mouse body weight, while histological inspections on major mouse organs including heart, liver, spleen, lung and kidney revealed no obvious pathological alterations (Supplementary Fig. 34). Taken together, the DNA-PAE@BAY-876 nanoassembly could effectively inhibit TNBC growth in vivo without inducing systemic toxicities.

To elucidate the tumor inhibition mechanism of the DNA-PAE@BAY-876 nanoassembly in vivo, the biochemical traits of 4T1 cells and the immune composition in the tumors were analyzed. Tumors in the DNA-PAE@BAY-876 group showed significantly enhanced glucose levels compared with the control group, while the lactic acid and UDP-GlcNAc abundance has been evidently downregulated (Fig. 7i–k), immediately suggesting that the DNA-PAE@BAY-876 nanoassemblies have successfully inhibited the glycolysis activity in tumor cells. As a resultant of the BAY-876-induced high glucose TME, flow cytometric analysis on the extracted tumors showed that the immunosuppressive

phenotype (Foxp3 and IL−10) of Tregs has decreased by 23.92% and 13.4% in the DNA-PAE@BAY-876 group compared with the control group (Fig. 8a, b), while the tumor infiltration of IFN-γ+/CD8+ T cells has increased by 30.33% (Fig. 8c, d). Meanwhile, DNA-PAE@BAY-876 treatment also promoted tumor infiltration of DCs and M1 macrophages by 26.94% and 15.42%, while reducing the frequency of M2 macrophages and MDSCs by 31.80% and 10.57% (Supplementary Fig. 35). The changes in the TNBC immune composition were also supported by immunohistochemical fluorescence imaging results, which revealed the enhanced infiltration of CD4+/CD8+ T cells and enhanced secretion of IFN-γ as well as reduced secretion of IL−10 in the TNBC tissues (Fig. 8f, g and Supplementary Figs. 36–38), thus validating the successful inflammation of TNBC microenvironment towards an anti-tumorigenic state.

In order to evaluate the overall effector immune cell deployment (EICD) in 4T1 tumor-bearing mice with DNA-PAE@BAY-876 treatment, the lymph nodes of 4T1 tumor-bearing mice to detect the phenotypical changes of DCs and central memory T cells (Tcms), while mouse serum was collected from the eyeballs of both mouse models to detect the secretion levels of key cytokines related to the immunotherapeutic effects. As shown in Supplementary Fig. 39, lymph nodes from the DNA-PAE@BAY-876 group showed significantly larger DC and Tcm populations compared with the control group, which has increased by 14.67% and 14.26%, respectively. Meanwhile, the secretion levels of pro-inflammatory IFN-γ and TNF-α in serum have increased by 33.65% and 30.87%, while the secretion level of anti-inflammatory IL−10 has decreased by 11.95%. These data showed that the DNA-PAE@BAY-876 treatment could regulate EICD to mount potent local and systemic adaptive anti-tumor immunity for effective TNBC treatment. The above analysis results on the immune composition in TNBCs confirmed that the DNA-PAE@BAY-876 treatment could transform the immunosuppressive TNBC into an immunoactivated phenotype.

TNBC is associated with high risk of metastasis and recurrence. Extending from the immunostimulatory activity of the DNA-PAE@BAY-876 nanoassemblies in vivo, we further investigated if the nanoassembly-evoked systemic antitumor immunity could eliminate metastatic TNBCs using a 4T1 lung metastasis mouse model. As shown in Fig. 8h, DNA-PAE@BAY-876 treatment effectively inhibited the lung metastasis of 4T1 tumors compared with the control group, for which the number of metastasis nodules was 71.42% lower than the PBS-treated controls. These observations evidently supported that treating TNBCs with DNA-PAE@BAY-876 nanoassemblies could elicit potent systemic antitumor immunity to reduce the risk of TNBC lung metastasis, which may add to the application potential of the nanoassembly-augmented ICT for TNBC management in the clinics. Meanwhile, we also tested if the nanoassembly could evoke systemic immune memory to prevent TNBC metastasis and relapse using bilateral 4T1 tumor-bearing mouse models. Notably, the DNA-PAE@BAY-876 nanoassemblies showed potent inhibitory effect on both the primary and secondary tumors (Supplementary Fig. 40a), thus significantly prolonging the survival of the mouse models. Mice in the DNA-PAE@BAY-876 group showed a final average volume of only around 59.14 mm$^3$ and average weight of only around 0.278 g for the secondary tumors (Fig. 9b and Supplementary Fig. 40b) and a median survival time of more than 60 days (Fig. 9c). To clarify the mechanism of DNA-PAE@BAY-876 nanoassembly to inhibit distal tumor growth, we employed flow cytometry to analyze the immune composition of the distal tumors. It was discovered that the DNA-PAE@BAY-876 nanoassembly treatment has also enhanced the immune cell infiltration in the distal tumors compared with the control group (Fig. 9d). Remarkably, the frequencies of tumor-infiltrating IFN-γ+/CD8+ T cells, F4/80+CD86+M1 macrophages and CD62−CD44+CD8+ effector memory T cells in the DNA-PAE@BAY-876 group were all higher than the control group by 14.99%, 20.72% and 33.48% (Fig. 9e–g), suggesting that the DNA-PAE@BAY-876 nanoassemblies could effectively generate

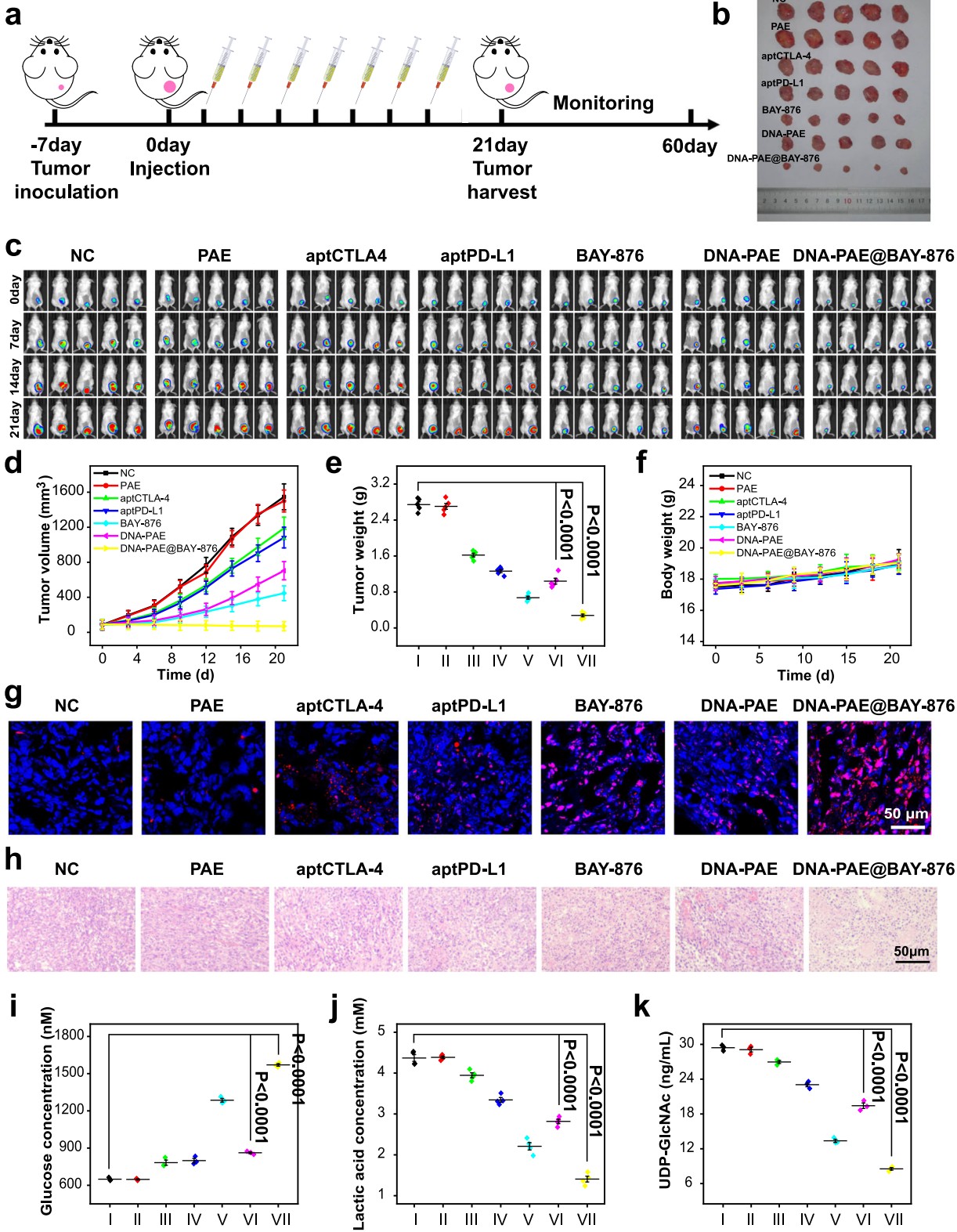

**Fig. 7 | Antitumor evaluation of DNA-PAE@BAY-876 in vivo. a** Schematic diagram of treatment schedule of DNA-PAE@BAY-876 nanoassemblies on 4T1 tumor-bearing mice. **b** Visual comparison of tumors from Balb/c mice after different treatment. **c** In vivo bioluminescence images on the tumor progression in Balb/c mice in different groups. **d** Tumor volume changes after different treatment. **e** Final tumor weight in different groups at the end of the 21 days period. **I** NC, (II) PAE, (III) aptCTLA-4, (IV) aptPD-L1, (V) BAY-876, (VI) DNA-PAE, (VII) DNA-PAE@BAY-876. **f** Body weight changes of Balb/c mice in different groups during treatment.

**g–h** TUNEL and H&E staining of tumor samples with five mice per group. **i–k** ELISA assay of glucose, lactic acid and UDP-GlcNAc abundance in tumor samples. In vivo histological experiments in panels **g–h** were repeated three times independently with similar results. Data are presented as mean values ± SEM ($n = 5$ mice for panels **d**–**f**, $n = 3$ mice for panels **i** and **k**, $n = 4$ mice for panel **j**). Statistical analysis was carried out via two-way ANOVA method. Source data are provided as a Source Data file.

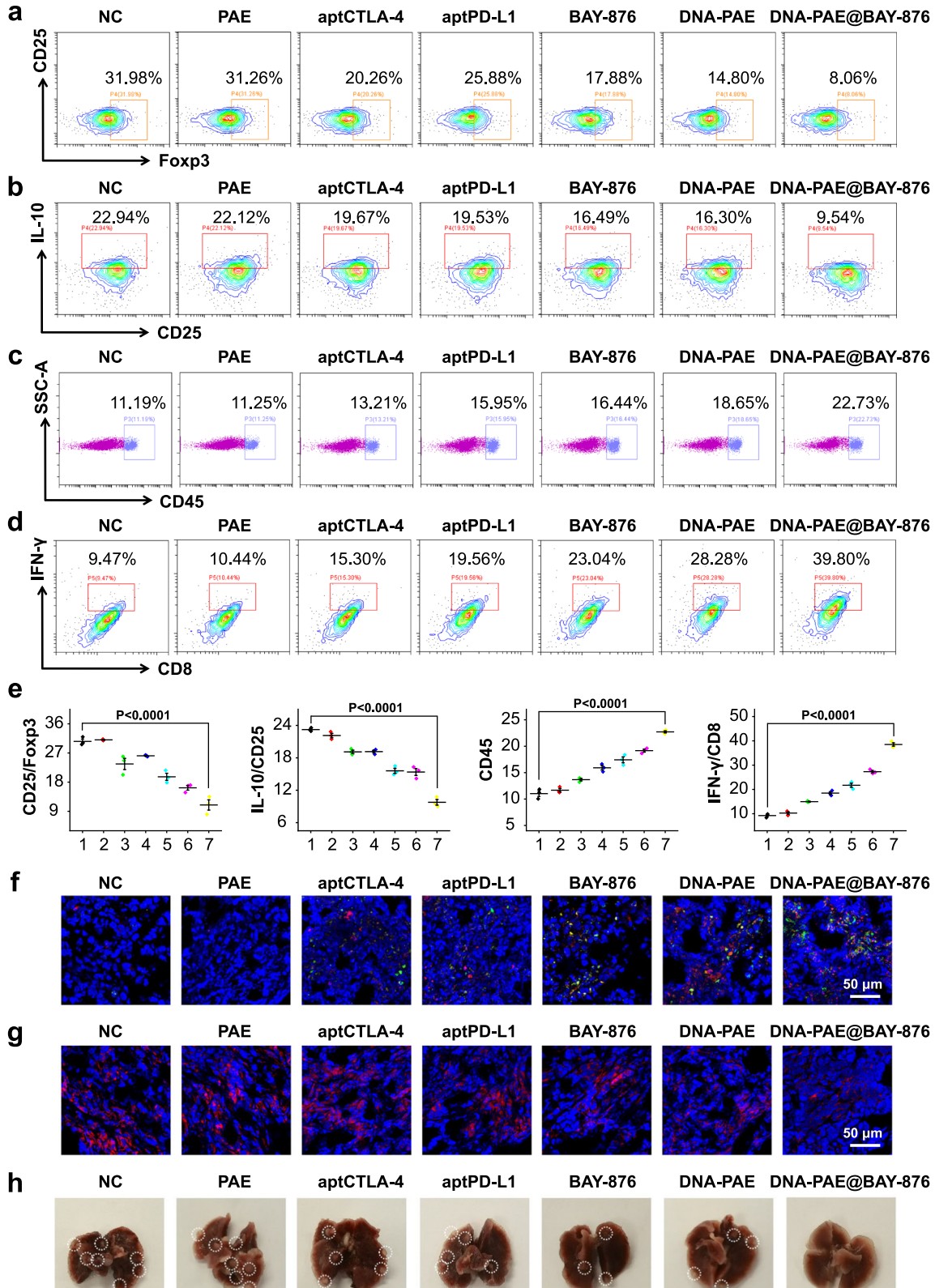

**Fig. 8 | Evaluation on DNA-PAE@BAY-876-mediated immunotherapy in vivo.**
**a**, **b** Flow cytometry detection on the tumor infiltration of Foxp3+ Tregs and IL−10+ Tregs. **c** Flow cytometry evaluation on the tumor-infiltration of total immune cells (CD45+). **d** Flow cytometry detection on the tumor infiltration of IFN-γ+CD8+ T cells. **e** Statistical analysis of data shown in panels **a**–**d** by flow cytometry (n = 3). (1) NC, (2) PAE, (3) aptCTLA-4, (4) aptPD-L1, (5) BAY-876, (6) DNA-PAE, (7) DNA-PAE@BAY-876. **f** Immunofluorescence images on the tumor-infiltration of CD4+ and CD8+ T cells in different groups with five mice per group.

**g** Immunofluorescence images of IL-10 in tumor tissues after different treatment with five mice per group. **h** Photographs of lung metastasis inhibition in 4T1 tumor-bearing mice after different treatments with five mice per group. Experiments in panels **a**–**d** and **f**–**h** were repeated three times independently with similar results. Data are presented as mean values ± SEM (n = 3 mice for panel **e**). Statistical analysis was carried out via two-way ANOVA method. Source data are provided as a Source Data file.

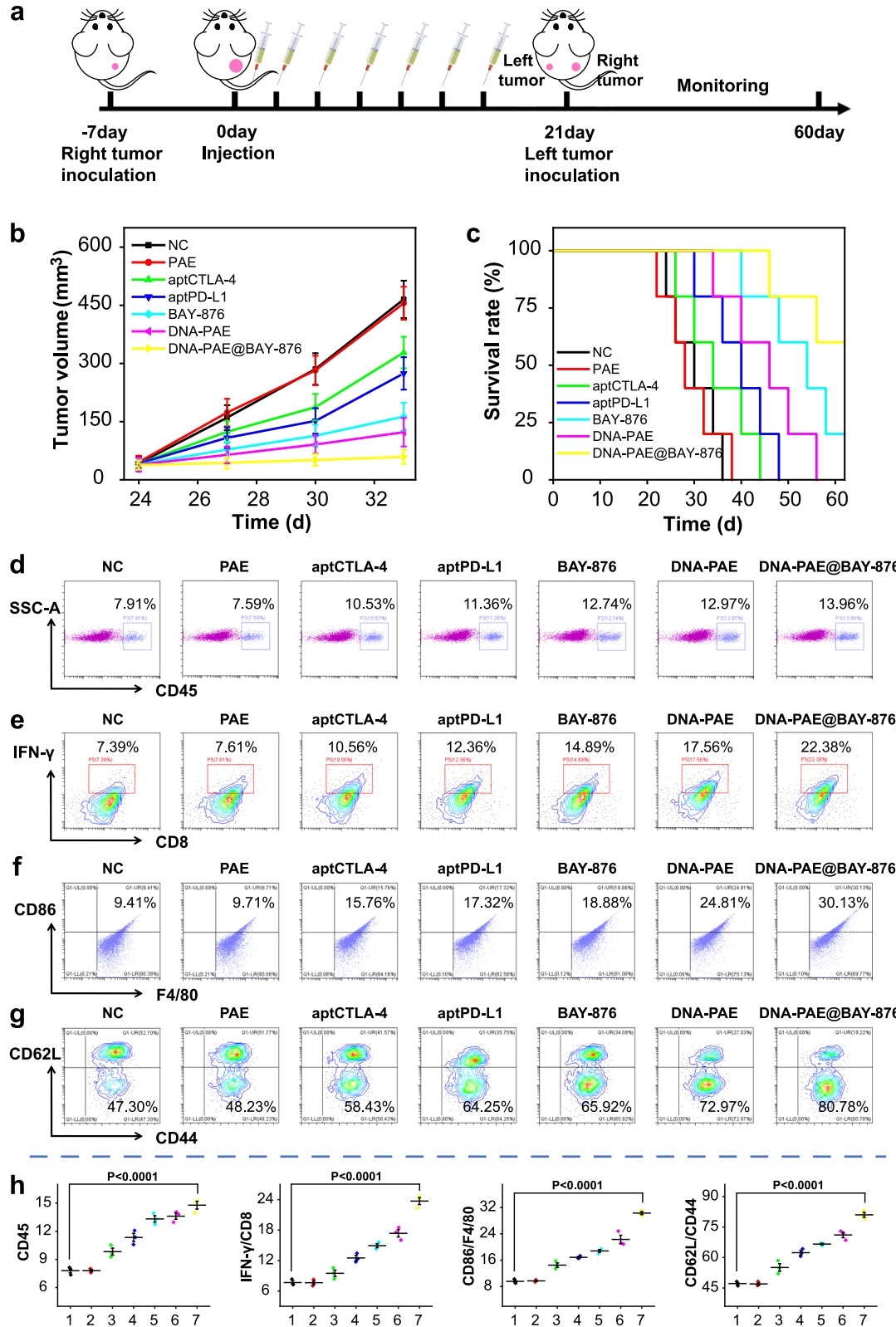

**Fig. 9 | Therapeutic evaluation of DNA-PAE@BAY-876 on bilateral tumor models. a** Schematic diagram of 4T1 bilateral tumor model construction and the treatment schedule. **b** Volume changes of the distal tumors after different treatment. **c** Survival curves of 4T1 bilateral tumor-bearing mice after different treatment. **d** Flow cytometry evaluation on the tumor-infiltration of total immune cells (CD45+). **e** Flow cytometry detection on the infiltration status of IFN-γ+CD8+T cells in distal tumor. **f**, **g** Flow cytometry evaluation on the tumor-infiltration of M1 macrophages (F4/80+CD86+) and CD8+ effector memory T cells (CD44+CD62L−). **h** Statistical analysis of flow cytometry data in panels **d**–**g**. (1) NC, (2) PAE, (3) aptCTLA-4, (4) aptPD-L1, (5) BAY-876, (6) DNA-PAE, (7) DNA-PAE@BAY-876. Flow cytometry experiments in panels **d**–**g** were repeated three times independently with similar results. Data are presented as mean values ± SEM (*n* = 3 mice for panels **b** and **h**, *n* = 5 mice for panel **c**). Statistical analysis was carried out via two-way ANOVA method. Source data are provided as a Source Data file.

high levels of effector memory T cells to mount antitumor immune memory for eliminating TNBC cells at systemic level.

## Therapeutic comparison DNA-PAE@BAY-876 to immune checkpoint inhibiting antibodies in vivo

The antitumor efficacy of DNA-PAE@BAY-876 nanoassembly was further compared to commonly investigated immune checkpoint inhibiting antibodies including anti-PD-L1 antibody /anti-CTLA-4 antibody in vivo for elucidating their clinical potential. Notably, the antitumor efficacy of aptPD-L1+aptCTLA-4 combination on 4T1 tumor-bearing mice was inferior to the abPD-L1+abCTLA-4 combination. In contrast, DNA-PAE@BAY-876 nanoassembly showed much superior therapeutic effect than antibody combination+BAY-876 treatment (Supplementary Fig. 41a–c). To elucidate the difference in the therapeutic performance of individual groups, we carried out comprehensive analysis on the pharmacokinetic behavior of individual components and found that most of anti-PD-L1 antibody reached the tumor site at 6 h, while BAY-876 were mainly enriched in the kidney site at 6 h (Supplementary Fig. 41d), as BAY-876 lacks intrinsic tumor targeting capacity and therefore failed to synchronize the antitumor actions of BAY-876 and immune checkpoint inhibiting antibodies. Indeed, the DNA-PAE@BAY-876 group showed superior tumor inhibition efficacy than the antibody-based treatments in terms of the mouse survival rate, tumor volume, and tumor weight (Supplementary Fig. 41e–g), accompanied with more pronounced tumor cell death according to H&E and TUNEL staining results (Supplementary Fig. 41h, i). The trends above were further supported by the evaluations on the immunostimulatory capacity of DNA-PAE@BAY-876 nanoassembly compared with antibody combination+BAY-876 by flow cytometry. Remarkably, compared with immune checkpoint inhibiting antibodies, DNA-PAE@BAY-876 nanoassembly more effectively promoted the infiltration and proliferation of CD8+ T cells, CD4+ T cells, DCs and M1 macrophages at the tumor site (Supplementary Fig. 42a–c), while potently inhibiting M2 macrophages, Tregs and MDSCs (Supplementary Fig. 42d–f), evidently supporting its potential utility for enhanced ICT against TNBCs in the clinics.

## Therapeutic evaluation of DNA-PAE@BAY-876 in humanized MDA-MB-231 tumor-bearing HSC-NOG-EXL mice

To further demonstrate the clinical translational potential of the DNA-PAE@BAY-876 nanoassembly, we constructed the humanized HSC-NOG-EXL mouse model (Supplementary Fig. 43) and thoroughly investigate its therapeutic impact thereof. To start with, we first extracted splenic immune cells from the humanized HSC-NOG-EXL mice to establish co-incubation system with MDA-MB-231 cells to investigate the targeting capacity of aptPD-L1 and aptCTLA-4 to human TNBC and immune cells. Notably, both aptamers showed good binding affinity with the designated cellular targets with high specificity (Supplementary Fig. 44), providing the mechanistic basis for the subsequently evaluations on humanized TNBC mouse models. Subsequently, by adapting the experimental set-up of 4T1 tumor-bearing mouse models, we divided humanized MDA-MB-231 tumor-bearing mice into 7 groups ($n = 3$) and then treated them with NC, PAE, aptCTLA-4, aptPD-L1, BAY-876, DNA-PAE, DNA-PAE@BAY-876 periodically until 21 days. Evaluations on the anti-TNBC efficacy of all sample groups revealed a similar trend like that on 4T1 tumor-bearing mouse models, where the DNA-PAE@BAY-876 nanoassembly showed the most pronounced antitumor effect with a final mean tumor volume and weight of 93 mm$^3$ and 0.24 g, respectively (Supplementary Fig. 45c, d). The nanoassembly-induced potent inhibition of MDA-MB-231 tumors in humanized mouse models was further supported by the H&E and TUNEL staining on the extracted tumor tissues, which revealed that DNA-PAE@BAY-876 nanoassembly induced the largest dead MDA-MB-231 cell population among all groups (Supplementary Fig. 45e, f). Extending from the antitumor efficacy of individual groups on humanized TNBC mouse models, we further analyzed the immune status in MDA-MB-231 tumors

to elucidate the therapeutic mechanisms of the nanoassembly. The DNA-PAE@BAY-876 substantially boosted the tumor infiltration of immune cells mediating the anti-tumorigenic immune responses. Specifically, the frequencies of mature DCs and M1 macrophages in the DNA-PAE@BAY-876 have increased by 17.89% and 21.96% compared with the control group, while frequencies of CD8+/CD4+ T cells have increased by 9.03% and 18.06% (Supplementary Fig. 46a–c). In contrast, frequencies of tumor-residing M2 macrophages, Tregs and MDSCs in DNA-PAE@BAY-876-treated MDA-MB-231 tumors have decreased by 21.50%, 12.00% and 12.45%, respectively (Supplementary Fig. 46d–f). In addition, the EICD in humanized MDA-MB-231 tumor-bearing mice after treatment with the nanoassemblies were evaluated using a similar set-up to the 4T1 tumor-bearing mouse models. Specifically, DC and Tcm frequencies in lymph nodes from the DNA-PAE@BAY-876 group were 11.29% and 15.39% higher than the control group, respectively, accompanied with a 28.97% and 25.00% increase in serum IFN-γ and TNF-α levels as well as a 11.50% decrease in serum IL-10 levels, indicating the successful nanoassembly-mediated regulation of EICD to boost the antitumor immunity. These observations collectively confirmed the antitumor potency of DNA-PAE@BAY-876 nanoassemblies against MDA-MB-231 tumors on humanized mouse models and supported its clinical potential for TNBC treatment on real-life patients.

## Discussion

In summary, we have developed a TME-responsive bispecific aptamer-based nanoassembly for enhancing the response of TNBC to ICT. The nanoassembly was constructed through the self-assembly of PAE-tagged aptPD-L1 and aptCTLA-4 combined with BAY-876. Upon reaching the acidic TME, the PAE handles would become protonated and switch from hydrophobic to a hydrophilic state, triggering the rapid dissociation of the nanoassemblies and in-situ release of aptPD-L1, aptCTLA-4 and BAY-876. BAY-876 could reduce glucose intake in GLUT1-overexpressing TNBC cells and block the glycolysis activities therein. On one hand, this would inhibit the Glu-F6P-UDP-GlcNAc axis in TNBC cells to reduce the glycosylation level of PD-L1 receptors on TNBC surface and enhance their binding affinity with aptPD-L1 for more effective immune checkpoint blockade. On the other hand, the delivered BAY-876 would abolish the overcompetition of environmental glucose by hyperglycolytic TNBC cells and establish a high-glucose environment for tumor-residing Tregs, which would induce their functional destabilization and cooperate with aptCTLA-4-mediated immune checkpoint blockade, eventually transforming the immunosuppressive Tregs into an immunoactivated phenotype. The nanoassembly-enabled metabolic rewiring of TNBC cells combined with bispecific immune checkpoint blockade cooperatively abolished the immune escape capacity of TNBCs and significantly inhibited TNBC growth in mouse models, providing an effective strategy for TNBC immunotherapy in the clinics.

## Methods
### Materials

N$_3$-PEG$_{2000}$-ACA was purchased from MeloPEG (Shengzhen, China). 1,4-butanediol diacrylate (HDD), 4,4′-trimethyldipiperidine (TDP), tris (3-hydroxypropyltriazolyl methyl) amine (THPTA), fructose-6-phosphate (F6P), IR-780 were purchased from Aladdin (Shanghai, China). BAY-876 and UDP-GlcNAc were purchased from MedChemExpress (Shanghai, China). Annexin V-FITC apoptosis detection kit, DAPI, and TUNEL detection kit were purchased from Beyotime (Shanghai, China). ELISA kit of IFN-γ, TNF-α, CXCL-10 and granzyme were purchased from ABclonal (Wuhan, China). Fructose-6-Phosphate (F6P) Assay Kit was purchased from Sigma-Aldrich. Red blood cell lysis buffer and Glycerol anhydrous were obtained from Solarbio (Beijing, China). Mouse-PD1 protein, Mouse-biotin-PD-L1 and Mouse-biotin-CTLA-4 were purchased from ACROBiosystems (Beijing, China). Streptavidin-coated magnetic beads were purchased from BEAVER

(Suzhou, China). DNA was purchased from Sangon (Shanghai, China), and the corresponding sequence information was provided in Supplementary Table 1. siNC and siPD-L1 were purchased from Biosyntech (Suzhou, China), and the corresponding sequence information was provided in Supplementary Table 2. The information of antibodies was provided in Supplementary Tables 3 and 4.

## Cells and mouse models
4T1, HC11, B16F10, 3T3, 4T1-luc, MDA-MB-231 cell lines were purchased from Shanghai Zeye Biotechnology Co., Ltd. with the catalog number of ZY-C6054M, ZY-C6027M, ZY-C6002M, ZY-C6050M, ZY-C6101M, ZY-C6044H, respectively. C57BL/6J and Balb/c mice (female, 6-week-old) were provided by Hunan Slake Jingda Experimental Animal Co., Ltd. The humanized HSC-NOG-EXL mice (female, 17-week-old) were provided by Beijing Charles River Experimental Animal Technology Co., Ltd.

All mice were kept in the animal house of Chongqing Medical University. Mice were housed in cages with five mice per cage and kept on in a regular 12 h light: 12 h dark cycle (9:00–21:00; 21:00–9:00). The temperature was $22 \pm 1$ degree Celsius and humidity was 40–68%. All animal tests have been reviewed and approved by the Animal Care and Use Committee of Laboratory Animals Administration of Chongqing Medical University, which were carried out following the Animal Management Rules of the Ministry of Health of the People's Republic of China.

## Material synthesis
100 mg $N_3$-$PEG_{2000}$-ACA was dissolved in 5 mL $CHCl_3$ and placed in 100 mL single-mouth flask, then stirred in an oil bath at 50 °C for 1 h under nitrogen environment. 100 mg $N_3$-$PEG_{2000}$-ACA were mixed with HDD and TDP at a ratio of $N_3$-$PEG_{2000}$-ACA: HDD: TDP = 0.1: 1: 1.1 in 10 mL $CHCl_3$, and this mixture was added to the flask drop by drop. The reaction was conducted at 50 °C for 48 h under nitrogen environment. The raw product was recovered through rotary evaporation at 40 °C at $6.7 \times g$, and the final product ($N_3$-$PEG_{2000}$-PAE) was freeze-dried for 48 h after ether precipitation.

The apt8–60 (120 nmol), aptCTLA-4-62 (120 nmol) and $N_3$-$PEG_{2000}$-PAE (240 nmol) were added to 2.5 mL PBS (pH = 7.4, 20 mM) in a 10 mL flask according to a ratio of 1:1:2. In addition, 1.0 μmol $CuSO_4 \cdot 5H_2O$/1.1 μmol tris (3-hydroxypropyltriazolyl methyl) amine (THPTA) and 4.0 μmol sodium ascorbate were added into above solution, then stirred at 15 °C for 24 h, put on dialysis (MWCO = 13,000 Da) for 48 h, and finally lyophilized under vacuum to obtain DNA-PAE powder.

The DNA-PAE and BAY-876 (100 mg·mL$^{-1}$ in DMSO) were added into PBS (pH = 7.4, 20 mM) in a 10 mL flask at the ratio of 1:0.1. The mixture was stirred for 6 h at room temperature, followed by dialysis (MWCO = 1000 Da) for 48 h, and finally lyophilized under vacuum to obtain DNA-PAE@BAY-876 powder.

## GPC characterization of $N_3$-$PEG_{2000}$-PAE
The lyophilized powder of $N_3$-$PEG_{2000}$-PAE was dissolved in $CHCl_3$ and filtered by 0.22 μm organic filter membrane, then determined by GPC (Gel Permeation Chromatography, Waters 1525, Suzhou Hechuan Chemical Technology Service Co., Ltd).

## TEM/SEM imaging of DNA-PAE@BAY-876
A small amount of DNA-PAE@BAY-876 lyophilized powder was dissolved in PBS (pH = 7.4, 20 mM), thoroughly mixed, then dripped onto copper mesh or silicon wafer, dried at 37 °C, and photographed by electron microscope.

## Dynamic light scattering (DLS) measurement
A small amount of samples were dissolved in PBS (pH = 7.4, 20 mM), thoroughly mixed, and then measured for determining particle size and Zeta potential.

## UV-vis spectroscopic analysis
A small amount of samples were dissolved in PBS (pH = 7.4, 20 mM) and thoroughly mixed. Characteristic peaks of each sample were measured and quantitatively analyzed by referring to BAY-876 standard curve.

## Determination of CMC and critical pH value
1 mg·mL$^{-1}$ DNA-PAE solution was added into PBS solution with different pH values or different concentrations. 0.05 mmol·L$^{-1}$ pyrene - acetone solution was added into 1.5 mL centrifugation tube and ventilated overnight. The above PBS solution was added into pyrene - acetone solution, then ultrasonicated (60 W) for 15 min, with a 2 min interval every 5 min. The above solutions were stood overnight and dried at room temperature. Finally, the characteristic peaks of pyrene were determined by a fluorescence spectrophotometer.

## Drug release from DNA-PAE@BAY-876
1 mg·mL$^{-1}$ DNA-PAE@BAY-876 solutions with different pH values were placed in dialysis bag (MWCO = 1000 Da) in a 100 mL beaker, followed by the addition of PBS and constant stirring. 1 mL PBS was extracted at predetermined time points while 1 mL fresh PBS was added back. Finally, the absorption spectra of extracted samples were determined by an ultraviolet spectrophotometer.

## Cell culture
4T1 cells (mouse triple negative breast cancer cells) were cultured in high-glucose DMEM medium. HC11 cells (mouse normal breast cells) were cultured in RPMI 1640 medium. MDA-MB-231 cells (human triple negative breast cancer cells) were cultured in high-glucose DMEM medium. All immune cells were cultured in RPMI 1640 medium. All media contained 10% fetal bovine serum (FBS) and 100 μg·mL$^{-1}$ double antibiotic solution (penicillin-streptomycin mixture).

4T1 cells or HC11 cells were cultured in the 5% $CO_2$ incubator at 37 °C, cell passage was required every day, and the time of pancreatic enzyme digestion was 5 min. All immune cells were cultured in the 5% $CO_2$ incubator at 37 °C and used as soon as possible within 4 days.

## Preparation of pH 6.8 medium
Lactic acid solution (20 mM) was added into high-glucose DMEM medium (pH 7.4) until the pH value dropped to 6.8, during which the medium pH was monitored using a pH detector.

## Splenocyte extraction
C57BL/6J mice were sacrificed and immersed into 75% alcohol, while the scissors, tweezers and filter membrane were sterilized by ultraviolet in advance. Mouse spleen was extracted and placed on an ultra-clean table, washed twice with PBS, put into 0.4 μm filter membrane with adding RPMI 1640 medium, and carefully ground with the syringe head. The splenocytes were centrifuged at $666.7 \times g$ for 5 min in the 50 mL centrifuge tube, mixed with 5 mL red blood cell lysis buffer and stood for 10 min, and again centrifuged at $666.7 \times g$ for 5 min with 5 mL RPMI 1640 medium. The collected cells were resuspended with 10 mL PBS and centrifuged. Finally, the splenocyte were incubated with RPMI 1640 media in an incubator at 37 °C.

## Sorting of immune cells
The splenocytes were filtered twice by 5 mL flow tubes with cell-strainer cap, and RPMI 1640 medium containing fluorescent antibody was added for processing. 1× sterilized PBS was prepared as the sheath solution and the above cells were sorted using BD FACSAria II (Becton Dickinson & Company, USA). After sorting, cells were immediately re-cultured with RPMI 1640 medium. CD4+ T cells were sorted using APC-anti-CD3 antibody and FITC-anti-CD4 antibody; CD8+ T cells were sorted using APC-anti-CD3 antibody and PE-anti-CD8 antibody; Treg cells were sorted using APC-antiCD3 antibody, FITC-anti-CD4 antibody and PE-anti-CD25

antibody; M1 macrophages were sorted using APC-anti-F4/80 antibody and FITC-anti-CD86 antibody; M2 macrophages were sorted using APC-anti-F4/80 antibody and PE-anti-CD206 antibody.

## T cell activation

The concentrations of anti-CD3 antibody and anti-CD28 antibody were respectively diluted to 5.5 µg·mL$^{-1}$ and 2 µg·mL$^{-1}$ by PBS. The above antibody diluent was added to 96-well plate at 70 µL per well and incubated at 4 °C overnight. After incubation, the liquid was drained and the plate was cleaned twice with PBS. The sorted T cells were added and cultured in an 37 °C incubator for 48 h or 96 h.

## Cytotoxicity determination using MTT assay

MTT powder was dissolved by the sterilized PBS at 5 mg·mL$^{-1}$ via ultrasonication, filtered by 0.22 µm filter membrane, and stored at 4 °C in the dark.

4T1 cells/HC11 cells were inoculated into 96-well plate with $1.0 \times 10^4$ per well, cultured for 24 h, cleaned twice with PBS and then added with different samples. The treated cells were cultured in an 37 °C incubator for 24 h, cleaned twice with PBS, followed by the addition of 0.5 mg·mL$^{-1}$ MTT solution at 100 µL per well. The incubation was at 37 °C for 2 h. The MTT solution was drained and each well was added with 100 µL DMSO. The plate was gently shaken for 10 min on a shaker. The absorbance was measured at 490 nm using the microplate reader.

## CCK-8 assay

To measure the cell viability of various immune cells after BAY-876 treatment, the immune cell populations were inoculated into 96-well plate at $1.0 \times 10^5$ units per well, and BAY-876 solutions with different concentrations and different pH values were added into the above plate for 24 h.10% CCK-8 solution was then added and incubated at 37 °C for 2 h. The absorbance was measured at 450 nm using a microplate reader.

To observe splenocyte migration under different conditions, 4T1 cells were inoculated into 12-well plate with $1.5 \times 10^5$ per well. After 24 h culture, splenocytes were added into above plate at $3 \times 10^6$ units per well with different samples. The cells were cultured at 37 °C for 24 h, centrifuged at $666.7 \times g$ for 5 min, and the supernatant was collected. Matrigel was added to the upper chamber, while the collected supernatant was added to the lower chamber. Subsequently, the splenocytes were transferred to the upper chamber at $1.0 \times 10^6$ per well. After 24 h, the upper chamber was taken out and the splenocytes in the lower chamber were collected and assayed by adding 10% CCK-8, followed by incubation at 37 °C for 2 h. Finally, the absorbance was measured at 450 nm using the microplate reader.

## Flow cytometry

To detect the dissociation constant of aptCTLA-4 or aptPD-L1 with their designated receptors, 1 mL streptavitin-coated magnetic beads were stood for 3 min with the magnet, then washed by PBS for three times. The biotin-modified mouse CTLA-4 or mouse PD-L1 protein with a final concentration of 50 nM was mixed with the above magnetic beads, and incubated on the rotator for 60 min. After cleaning with PBS for three times, FAM-aptCTLA-4 or Cy5-aptPD-L1 with different concentrations were added into the above mixture, then incubated at room temperature for 30 min. After washing twice with PBS, the fluorescence level was measured by flow cytometry (CytoFLEX, Beckman Coulter). The dissociation constants of aptPD-L1 and aptCTLA-4 were calculated by the formula $Y = B_{max} X / (K_D + X)$.

To measure the binding ability of aptPD-L1 and aptCTLA-4 with different cell populations, 4T1 cells/HC11 cells were inoculated into 12-well plate with $1.5 \times 10^5$ per well. After culturing for 24 h, the cells were blocked with 5% BSA for 30 min. Subsequently, different fluorescently labeled samples were added into above cells. After incubated for

30 min, these cells were detected by flow cytometry (CytoFLEX, Beckman Coulter).

T cells were sorted from splenocytes of C57BL/6J mice, and some of them were activated by antibody treatment. T cells were stored in 1.5 mL centrifuge tubes and then blocked with 5% BSA for 30 min. Different fluorescently labeled samples were added into above T cells and incubated for 30 min. Finally, these T cells were detected by flow cytometry (CytoFLEX, Beckman Coulter).

To measure the binding ability of aptPD-L1 and aptCTLA-4 in co-culture system of immune cells and TNBC cells, 4T1 cells and splenic immune cells were mixed in the 1.5 mL centrifuge tube at a ratio of 1:30 and treated with 5% BSA for 30 min, followed by the treatment of Cy5-aptPD-L1 or FAM-aptCTLA-4 for another 30 min of incubation. After washing twice with PBS, PC7-anti-CD45 antibody, PE-anti-CD4 antibody and APC-anti-CD25 antibody was added into above cells. Finally, the fluorescence intensity on 4T1 cells or immune cells was measured by flow cytometry (CytoFLEX, Beckman Coulter).

To quantitatively analyze the expression of GLUT1 and GLUT3 in tumor-residing cell populations, the 4T1 tumor-bearing Balb/c mouse model was constructed, and the tumors sized about 600 mm$^3$ were collected. The 4T1 tumor was washed twice with PBS, placed on a 0.4 µm filter membrane with adding high-glucose DMEM medium, and carefully pulverized with the syringe head, then mixed with 5 mL red blood cell lysis buffer and stood for 10 min, and centrifuged at $666.7 \times g$ for 5 min. The cells were collected and sealed by 10% FBS for 2 h, then mixed with anti-GLUT1 antibody (rabbit host) and anti-GLUT3 antibody (mouse host) overnight at 4 °C, followed by incubation with Cy5-labeled rabbit second antibody or FAM-labeled mouse second antibody for 2 h at room temperature. After washing twice with PBS, the cells were assayed by flow cytometry (CytoFLEX, Beckman Coulter).

To observe glucose uptake of 4T1 cells and T cells under different conditions, 4T1 cells were inoculated in 12-well plate with $1.5 \times 10^5$ cells per well, cultured for 24 h, added with different samples and T cells of 30 times the amount of tumor cells. 30 mM 2-NBDG was subsequently added and incubated at 37 °C for 24 h. After centrifuging at $666.7 \times g$ for 5 min, all cells were added with APC-anti-CD45 antibody, incubated at room temperature for 30 min in the dark, then detected by flow cytometry (CytoFLEX, Beckman Coulter).

4T1 cells were inoculated in 12-well plate with $1.5 \times 10^5$ cells per well, then added with different concentrations of BAY-876 after cultured for 24 h. 30 mM 2-NBDG was subsequently added and incubated at 37 °C for 24 h. Finally, 4T1 cells were collected and detected by flow cytometry (CytoFLEX, Beckman Coulter).

To evaluate the effects of BAY-876 and KL-11743 on major immune cell populations in the co-incubation system, 4T1 cells were inoculated into 12-well plate with $1.5 \times 10^5$ units per well. The cells were cultured for 24 h, added with splenocytes of 30 times the amount of tumor cells, and then incubated with 100 nM BAY-876 or 5 µM KL-11743 at 37 °C for 24 h. The splenocytes were collected and processed with the corresponding fluorescent antibodies at room temperature for 30 min in the dark. Finally, the cells were detected by flow cytometry (CytoFLEX, Beckman Coulter).

To detect the apoptosis of 4T1 cells in co-culture system after various treatment, 4T1 cells were inoculated into 12-well plate with $1.5 \times 10^5$ per well. The cells were cultured for 24 h, added with different samples and splenocytes of 30 times the amount of tumor cells, and incubated at 37 °C for 24 h. After centrifuging at $666.7 \times g$ for 5 min, all cells were added with 5 µL FITC-Annexin V and APC-anti-CD45 antibody, incubated at room temperature for 30 min in the dark, then added with 10 µL PI and incubated at room temperature for 10 min in the dark. The stained cells were finally detected by flow cytometry (CytoFLEX, Beckman Coulter).

To detect the apoptosis of T cells after different treatment, T cells were separated from splenocytes and added into 12-well plate with equal amount, followed by the addition of different samples. The cells

were incubated in an incubator for 24 h. The T cells were centrifuged at 666.7 × g for 5 min and added with 5 μL FITC-Annexin V and APC-anti-CD3 antibody. After incubation at room temperature for 30 min in the dark, the T cells were added with 10 μL PI and then incubated at room temperature for 10 min in the dark before the detection by flow cytometry (CytoFLEX, Beckman Coulter).

To evaluate phenotypic changes on major immune cell populations after various treatments, 4T1 cells were inoculated into 12-well plate with $1.5 \times 10^5$ units per well. The cells were cultured for 24 h, added with different samples and splenocytes of 30 times the amount of tumor cells, and then incubated at 37 °C for 24 h. The splenocytes were collected and processed by the corresponding fluorescent antibodies at room temperature for 30 min in the dark. Finally, the cells were detected by flow cytometry (CytoFLEX, Beckman Coulter).

To elucidate PD-1/aptPD-L1 competition in vitro, 4T1 cells were inoculated into 12-well plate with $1.5 \times 10^5$ per well, cultured for 24 h and incubated at 37 °C with different samples for 24 h. FAM-aptPD-L1/Cy5-PD1 were subsequently added and the cells were detected using flow cytometry (CytoFLEX, Beckman Coulter).

To evaluate the IFN-γ/IL-10 expression levels in CD3+ T cells, 4T1 cells were inoculated into 12-well plate with $1.5 \times 10^5$ per well. The cells were cultured for 24 h, added with different samples and splenocytes of 30 times the amount of tumor cells, and then incubated at 37 °C for 24 h. The splenocytes were collected and incubated with PC7-anti-CD45 antibody, APC-anti-CD3 antibody, FITC-anti-IFN-γ antibody and PE-anti-IL-10 antibody at room temperature for 30 min in the dark. Finally, the cells were detected by flow cytometry (CytoFLEX, Beckman Coulter).

### Binding of aptPD-L1/aptCTLA-4 to mouse PD-L1/CTLA-4 proteins via magnetic bead assay
1 mL streptavitin-coated magnetic beads were stood for 3 min with the magnet, then washed by PBS for three times. The biotin-modified mouse PD-L1 or mouse CTLA-4 protein with a final concentration of 50 nM was mixed with the above magnetic beads, and incubated on the rotator for 60 min. After cleaning with PBS for three times, 150 nM Cy5-aptPD-L1 or FAM-aptCTLA-4 with different pH was added into the above mixture, then incubated at room temperature for 30 min. After washing twice with PBS, the fluorescence level was measured by a fluorescence spectrophotometer.

### Optical microscopic analysis
For the optical analysis of treatment-induced cytotoxicity on TNBC cells, 4T1 cells were inoculated into 12-well plates at $1.5 \times 10^5$ per well. After culturing for 24 h, the cells were added with different samples and splenocytes of 30 times the amount of tumor cells at 37 °C for 24 h. After washing three times with PBS, the cells were re-suspended and observed using an optical microscope.

To test the invasion capacity of 4T1 cells after various treatment, 4T1 cells were inoculated into 12-well plate with $1.5 \times 10^5$ per well. After culturing for 24 h, the cells were added with different samples and splenocytes of 30 times the amount of tumor cells at 37 °C for 24 h, centrifuged at 666.7 × g for 5 min to collect the supernatant. The supernatant was put into the lower chamber while 4T1 cells were inoculated into the upper chamber with $1.0 \times 10^5$ per well. After incubation for 24 h, the 4T1 cells were stained by crystal violet at room temperature for 30 min, washed several times with PBS, and observed using an optical microscope.

### Enzyme-linked immunosorbent assay (ELISA)
4T1 cells were inoculated into 12-well plate with $1.5 \times 10^5$ per well. After culturing for 24 h, the cells were added with different materials and 30 times the amount of splenocyte at 37 °C for 24 h, centrifuged at 666.7 × g for 5 min, and the supernatant was collected. Finally, the collected supernatant was tested according to the ELISA kit instructions.

### ATP and NADPH abundance in 4T1 cells
4T1 cells were inoculated into 6-well plates at $5 \times 10^5$ units per well. After culturing for 24 h, the cells were treated by BAY-876 with different concentration for 24 h. After cleaning with PBS, the cells were lysed with RIPA lysate (containing 1% PMSF) for 30 min at 4 °C and centrifuged at 4000 × g for 10 min. Eventually, the collected supernatant was assayed by the ATP or NADPH kit to determine the cellular ATP and NADPH abundance.

### CLSM analysis
To observe the pH-dependent alterations of DNA-PAE nanoassemblies and the impact on cell interactions, 4T1 cells or HC11 cells were inoculated into the 20 mm confocal dish with $1.5 \times 10^5$ per dish. After culturing for 24 h, the cells were added with pH 7.4 or pH 6.8 medium containing FITC-DNA-PAE nanoassemblies or anti-PD-L1 antibody at 37 °C for 2 h, then fixed with 4% paraformaldehyde for 30 min. After washing twice with PBS, the cells were treated with FITC-secondary antibody corresponding to anti-PD-L1 antibody at room temperature for 2 h. The stained cells were washed twice with PBS and stained for 10 min with DAPI in the dark, and finally imaged by a laser confocal microscope (Leica TCS SP8, Germany).

To evaluate the effect of DNA-PAE@BAY-876 on GLUT1 activity, 4T1 cells or HC11 cells were inoculated into the 20 mm confocal dish with $1.5 \times 10^5$ per dish. The cells were cultured for 24 h, then added with 48.87 μg·mL$^{-1}$ DNA-PAE@BAY-876 at 37 °C for 24 h. The media were drained while the cells were fixed with 4% paraformaldehyde for 30 min. Anti-GLUT1 antibody was added into the confocal dish for incubation at 4 °C overnight. After washing twice with PBS, the cells were treated with fluorescent secondary antibody at room temperature for 2 h. The stained cells were washed twice with PBS and stained for 10 min with DAPI in the dark, and finally imaged by a laser confocal microscope (Leica TCS SP8, Germany).

For immunofluorescence imaging of GLUT1/GLUT3 in vivo, the 4T1 tumor-bearing Balb/c mouse model was constructed, and the tumors sized about 600 mm³ were collected. After the tumors were frozen-sectioned, the sections were fixed with 4% paraformaldehyde at room temperature for 30 min. After cleaning with PBS, the sections were blocked with 5% BSA for 2 h, then added with anti-GLUT1 antibody/anti-GLUT3 antibody at 4 °C overnight, washed twice with PBS, added with fluorescent secondary antibody at room temperature for 2 h and stained for 10 min with DAPI in the dark, finally sealed with anhydrous glycerol for observation by CLSM (Leica TCS SP8, Germany).

To elucidate the aptPD-L1/PD-1 binding, 4T1 cells were inoculated into the 20 mm confocal dish with $1.5 \times 10^5$ per dish. The cells were cultured for 24 h, and then treated with 100 nM BAY-876 at 37 °C for 24 h. After washing with PBS, the cells were added with FAM-aptPD-L1/Cy5-PD1 protein for 30 min in the dark. After staining for 10 min with DAPI, the cells were photographed by laser confocal microscope (Leica TCS SP8, Germany).

### Protein expression detection by Western Blotting
4T1 cells or HC11 cells were inoculated into 6-well plates with $5 \times 10^5$ per well. After culturing for 24 h, the cells were treated by the corresponding samples for 24 h. After incubation, the cells were lysed with RIPA lysate (containing 1% PMSF) to extract total protein, and the protein was quantified by BCA method. Finally, the expression of corresponding proteins was observed by 12% SDS -polyacrylamide gel electrophoresis.

### UDP-GlcNAc-mediated recovery of PD-L1 glycosylation in BAY-876-treated 4T1 cells
4T1 cells were inoculated into 6-well plates at $5 \times 10^5$ units per well. After culturing for 24 h, the cells were treated by 100 nM BAY-876 for 12 h, then continually incubated with UDP-GlcNAc or F6P for 12 h. Finally, the PD-L1 glycosylation level was observed by Western Blotting and the cellular UDP-GlcNAc level was detected by the ELISA kit.

### In vivo pharmacokinetic analysis of DNA-PAE@BAY-876

BAY-876 and DNA-PAE@BAY-876 were injected into Balb/c mice through the tail vein, and blood was collected from the tail at different time points. After centrifugation, the supernatants were extracted to determine the concentration of BAY-876.

### In vivo delivery analysis of DNA-PAE@BAY-876

Firstly, FITC-DNA-PAE@BAY-876 nanoassembly was synthesized using FITC-labeled aptPD-L1 or FITC-labeled aptCTLA-4. Meanwhile, the 4T1 tumor-bearing Balb/c mouse model was constructed and treated with FITC-DNA-PAE@BAY-876 through intravenous injection. Then, the tumors at about 600 mm$^3$ were collected.

A portion of the collected tumor tissue was frozen-sectioned and these sections were fixed with 4% paraformaldehyde at room temperature for 30 min. After cleaning with PBS, the sections were blocked with 5% BSA for 2 h, then added with APC-anti-CD45 antibody or APC-anti-PD-L1 antibody at 4 °C overnight, then stained for 10 min with DAPI in the dark, finally sealed with anhydrous glycerol for observation by laser confocal microscope (Leica TCS SP8, Germany).

Another batch of the tumor tissue was ground to powder using liquid nitrogen, and then lysed with RIPA lysate (containing 1% PMSF) for 30 min at 4 °C and centrifuged at 4000 × $g$ for 10 min. The collected supernatant was filtered by ultrafiltration tube (MWCO = 600 Da) and the BAY-876 content in tumors was quantified by a liquid chromatography-triple quadrupole mass spectrometer. At the same time, BAY-876-methanol solutions with different concentrations were prepared to establish the standard curve by liquid chromatography-triple quadrupole mass spectrometer.

### PD-L1 gene knock-down

4T1 cells were inoculated into 6-well plates with 5 × 10$^5$ per well or the 20 mm confocal dish with 1.5 × 10$^5$ per dish. After culturing for 24 h, the cells were treated by siNC or siPD-L1 with PEI for 8 h, then washed twice with PBS and incubated at 37 °C for 24 h, finally observed by Western Blotting or photographed by a laser confocal microscope (Leica TCS SP8, Germany).

### Establishment of animal models

All animal tests have been reviewed and approved by the Animal Care and Use Committee of Laboratory Animals Administration of Chongqing Medical University, which carried out following the Animal Management Rules of the Ministry of Health of the People's Republic of China. According to the national and institutional guidelines, the maximum tumor size allowed was 2000 mm$^3$, and mice were euthanized when the tumor burden exceeded the threshold.

To establish the 4T1 tumor-bearing Balb/c mouse model, 100 μL 2 × 10$^6$ 4T1 cells were injected subcutaneously into each Balb/c mouse, when the tumor grew to about 100 mm$^3$, the mice were randomly divided into 7 groups at 6 mice per group and subjected to different treatment (PBS, 24.01 μg·mL$^{-1}$·g$^{-1}$ PAE, 12.35 μg·mL$^{-1}$·g$^{-1}$ aptCTLA-4, 12.22 μg·mL$^{-1}$·g$^{-1}$ aptPD-L1, 0.05 μg·mL$^{-1}$·g$^{-1}$ BAY-876, 48.31 μg·mL$^{-1}$·g$^{-1}$ DNA-PAE, 48.87 μg·mL$^{-1}$·g$^{-1}$ DNA-PAE@BAY-876) through intravenous injection once every two days, and tumor volume and body weight of every mouse were recorded periodically. The tumor size was calculated using the formula Vtumor = L × W$^2$/2 (L: longitudinal diameter of the tumor, W: cross-sectional diameter of the tumor). After 21 days of treatment, all mice were sacrificed, their tumor tissues, major organs and lymph nodes were collected for post-analysis while the blood of mice from the eyeball was collected using the heparin tube and then serum was collected after centrifugation for ELISA assay of multiple cytokines. For the 4T1 tumor-bearing bilateral Balb/c mouse model, 100 μL 5 × 10$^6$ 4T1 cells were injected subcutaneously into the opposite side of the primary tumors after 21 days of treatment, and the murine survival curves were plotted across a period of 60 days.

A portion of the collected tumor tissue and lymph nodes were washed twice with PBS, put into 0.4 μm filter membrane with adding high-glucose DMEM medium, and carefully ground with the syringe head, then mixed with 5 mL red blood cell lysis buffer and stood for 10 min, and centrifuged at 666.7 × $g$ for 5 min. The supernatant was removed, the cells were re-suspended with PBS, then stained with corresponding fluorescent antibodies for 30 min to detect the infiltration of various immune cells by flow cytometry.

Moderate amount of the tumor tissue was ground to the powder using liquid nitrogen, and then lysed with RIPA lysate (containing 1% PMSF) for 30 min at 4 °C and centrifuged at 4000 × $g$ for 10 min. The collected supernatant was used for Western Blotting or ELISA to detect the expression of corresponding protein or the abundance of glycolysis-related metabolites.

Another portion of the tumor tissue was frozen-sectioned, fixed with 4% paraformaldehyde at room temperature for 30 min. After permeation with 0.5% Triton X-100 for 10 min, the sections were incubated with TUNEL test solution at 37 °C for 60 min, then added with DAPI for 10 min in the dark and sealed with glycerol anhydrous after washing with PBS. The mounted tissue sections were finally imaged by a laser confocal microscope (Leica TCS SP8, Germany).

The remaining tumor tissues were frozen-sectioned, fixed with 4% paraformaldehyde at room temperature for 30 min. After cleaning with PBS, the sections were blocked with 5% BSA for 2 h, incubated with fluorescent antibodies at 4 °C overnight, added with DAPI for 10 min in the dark, then sealed with glycerol anhydrous after washing with PBS. The mounted tissue samples were finally imaged by a laser confocal microscope (Leica TCS SP8, Germany).

To evaluate the lung metastasis of TNBC after nanoassembly treatment, 100 μL 2 × 10$^6$ 4T1 cells were injected subcutaneously into each Balb/c mouse, when the tumor grew to about 100 mm$^3$, the mice were randomly divided into 7 groups at 6 mice per group and subjected to different treatment (PBS, 24.01 μg·mL$^{-1}$·g$^{-1}$ PAE, 12.35 μg·mL$^{-1}$·g$^{-1}$ aptCTLA-4, 12.22 μg·mL$^{-1}$·g$^{-1}$ aptPD-L1, 0.05 μg·mL$^{-1}$·g$^{-1}$ BAY-876, 48.31 μg·mL$^{-1}$·g$^{-1}$ DNA-PAE, 48.87 μg·mL$^{-1}$·g$^{-1}$ DNA-PAE@BAY-876) through intravenous injection once every two days. On day 50, the mice were sacrificed to extract the lungs, which fixed with 4% paraformaldehyde and photographed to count the number of lung metastasis nodules.

For the purpose of therapeutic comparison between DNA-PAE@BAY-876 nanoassembly and immune checkpoint inhibiting antibodies in vivo, 100 μL 2 × 10$^6$ 4T1 cells were injected subcutaneously into each Balb/c mouse, when the tumor grew to about 100 mm$^3$, the mice were randomly divided into 5 groups at 6 mice per group and subjected to different treatment (Control, 50 μg·mL$^{-1}$·g$^{-1}$ anti-PD-L1 antibody and 50 μg·mL$^{-1}$·g$^{-1}$ anti-CTLA-4 antibody, 12.35 μg·mL$^{-1}$·g$^{-1}$ aptCTLA-4 and 12.22 μg·mL$^{-1}$·g$^{-1}$ aptPD-L1, 50 μg·mL$^{-1}$·g$^{-1}$ antibodies and 0.05 μg·mL$^{-1}$·g$^{-1}$ BAY-876, 48.87 μg·mL$^{-1}$·g$^{-1}$ DNA-PAE@BAY-876) through intravenous injection once every two days. After 21 days of treatment, all mice were sacrificed, their tumors were collected for analysis. The murine survival curves were plotted across a period of 60 days.

A portion of the collected tumor tissue was washed twice with PBS, placed on a 0.4 μm filter membrane, added with high-glucose DMEM medium, carefully pulverized with syringe head, mixed with 5 mL red blood cell lysis buffer, stood for 10 min, and centrifuged at 666.7 × $g$ for 5 min. The supernatant was removed, the cells were re-suspended with PBS and then stained with fluorescent antibodies for 30 min to detect the infiltration of various immune cells in the tumor by flow cytometry (CytoFLEX, Beckman Coulter).

Another portion of the tumor tissue was frozen-sectioned, fixed with 4% paraformaldehyde at room temperature for 30 min. After permeation with 0.5% Triton X-100 for 10 min, the sections were incubated with TUNEL test solution at 37 °C for 60 min, then added with DAPI for 10 min in the dark and sealed with glycerol anhydrous after washed with PBS. The mounted tissue sections were finally imaged by a laser confocal microscope (Leica TCS SP8, Germany).

To establish the humanized MDA-MB-231 tumor-bearing HSC-NOG-EXL mouse model, 100 μL 2 × 10⁶ MDA-MB-231 cells were injected subcutaneously into each humanized HSC-NOG-EXL mouse. When the tumors grew to about 100 mm³, the mice were randomly divided into 7 groups ($n = 3$) and subjected to different treatment (PBS, 24.01 μg·mL⁻¹·g⁻¹ PAE, 12.35 μg·mL⁻¹·g⁻¹ aptCTLA-4, 12.22 μg·mL⁻¹·g⁻¹ aptPD-L1, 0.05 μg·mL⁻¹·g⁻¹ BAY-876, 48.31 μg·mL⁻¹·g⁻¹ DNA-PAE, 48.87 μg·mL⁻¹·g⁻¹ DNA-PAE@BAY-876) through intravenous injection once every two days. After 21 days of treatment, all mice were sacrificed, their tumor tissues and lymph nodes were collected for post-analysis while the blood of mice from the eyeball was collected using the heparin tube and then serum was collected after centrifugation for ELISA assay of multiple cytokines.

A portion of the collected tumor tissue and lymph nodes were washed twice with PBS, placed on 0.4 μm filter membrane, added with high-glucose DMEM medium, carefully pulverized with the syringe head, then mixed with 5 mL red blood cell lysis buffer and stood for 10 min, and centrifuged at 666.7 × $g$ for 5 min. The supernatant was removed, the cells were re-suspended with PBS, then stained with corresponding fluorescent antibodies for 30 min to detect the infiltration of various immune cells by flow cytometry.

Another portion of the tumor tissue was frozen-sectioned, fixed with 4% paraformaldehyde at room temperature for 30 min. After permeation with 0.5% Triton X-100 for 10 min, the sections were incubated with TUNEL test solution at 37 °C for 60 min, then added with DAPI for 10 min in the dark and sealed with glycerol anhydrous after washed with PBS. The mounted tissue sections were finally imaged by a laser confocal microscope (Leica TCS SP8, Germany).

### Statistics and reproducibility
All measurements contained three or more independent replicates from separate experiments. The exact sample size and statistical test for each experiment are described in the relevant figure legends. All statistical analysis results are presented as mean ± standard error (S.E.M.). All statistical data were processed in GraphPad Prism (version 9.5 for Windows) by Student's $t$ test, one-way ANOVA or two-way ANOVA.

### Reporting summary
Further information on research design is available in the Nature Portfolio Reporting Summary linked to this article.

## Data availability
The authors declare that all data generated in this study are available within the Article, Supplementary Information or Source data file. Source data are provided with this paper.

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

## Acknowledgements

The authors would like to thank the Analytical and Testing Center of Chongqing University for their assistance during sample characterization. This study is financially supported by National Natural Science Foundation of China (32122048, 11832008, 92059107, and 51825302), Chongqing Outstanding Young Talent Supporting Program (cstc2021ycjh-bgzxm0124) and Natural Science Foundation of Chongqing Municipal Government (cstb2022nscq-msx0488).

## Author contributions

M.H.L., Y.H. and Z.L. jointly supervised this study. Y.H, Z.L. and M.H.L. conceptualized this study. X.J.R. and Z.C. performed the experiment. Y.H, Z.L, M.H.L., X.J.R., J.M.H., X.M.Y., Y.Q.L. and K.Y.C. analyzed and interpreted the data. Y.H., Z.L., M.H.L and X.J.R. wrote the paper. Y.H, Z.L., M.H.L. and X.J.R. helped with the revision of the draft.

## Competing interests

The authors declare no competing interests.
