## [Peer Review File · Nature Communications]

Inhibition of glycolysis-driven immunosuppression with a nano-assembly enhances response to immune checkpoint blockade therapy in triple negative breast cancerREVIEWER COMMENTS

Reviewer #1 (Remarks to the Author): with expertise in nanotechnology, aptamers, cancer immunotherapy

In this work, β -amino ester (PAE)-modified PD-L1 and CTLA-4-antagonizing aptamers (aptPD-L1 and aptCTLA-4) was co-assembled for the targeted delivery of CLUT1 inhibitor BAY-876. The as-synthesized DNA-PAE@BAY-876 could respond to acidic condition to initiate BAY-876 and aptamer release, and then reshaped the TME to enhance ICI response.

So many multifunctional nanoplatfroms have been explored for combinational ICT and TME remodeling in cancer treatment. Actually, this work is not the first effort to integrate tumor-glycolysis inhibition and ICT (e.g. ACS Nano, 2020, 14, 11055). Additionally, the aptamer and the BAY-876 used in this study also have been reported earlier. Therefore, it is hard to evaluate the significance and value of the designed systems. Some critical issues should be addressed as follow:

1. All flow cytometric analysis lack parallel tests and statistic analysis. Only single test cannot draw a solid conclusion. Meanwhile, the gating strategies in all flow cytometric analysis must be provided. For example, in Figure 4a, the tumor cells were co-incubated with splenocytes. How the authors separate tumor cells from the immune cells and spleen cells?
2. In Figure 2g, the name of the DNA-PAE was mislabeled.
3. In Figure 2h, the UV-Vis spectrum of pure BAY-876 should be provided.
4. In Page 9 Line 178, the authors mentioned that "the DNA-PAE@BAY-876 nanoassembly has high stability under physiological conditions...". The reviewer think that the 26% leakage is not negligible. The reviewer doubts that the nanoassembly could be defined as a high stability system.
5. In Figure 3c, AbPD-L1 could bond to 4T1 cell surface under pH 7.4 but not pH 6.8. Why? In addition, no DNA-PAE and AbPD-L1 could enter HC11 cells? No endocytosis occurred? Importantly, the different cellular uptakes in two different cell lines cannot confirm the specific interaction between PD-L1 and aptPD-L1.
6. The high PD-L1 expression of the 4T1 cell line used in this work must be verified.
7. The CD45 fluorescence in Figure 3j-3k is very weak. It cannot be concluded from these data that "CD45+ lymphocytes showed high GLUT3 expression..." (Line 233-234).
8. Values of x and y coordinate axis in Figure 5g were very unclear.

Reviewer #2 (Remarks to the Author): with expertise in immuno-metabolism

Ren et al. present a new strategy to target triple negative breast cancer using PD-L1 and CTLA-4-antagonizing aptamers in nanoassemblies that carry BAY-876 (an inhibitor of GLUT1). They suggest that their strategy specifically targets the tumour microenvironment, as the low pH triggers PAE protonation and nanoassembly dissociation. This strategy reduces tumour growth in mice inoculated with 4T1 cells in the flank. The authors suggest that GLUT-1 inhibition targets 4T1 cells specifically as these cells have high expression of GLUT-1. GLUT-1 inhibition reduces the viability of 4T1 cells and reduces glycosylation of PD-L1. I have several major concerns that I have summarised below.

1. It is unclear to me when the authors are using cell lines or tumours. If the authors are using cell lines this is not the 'tumor microenvironment'. How the authors setup their experiments generally needs clarifying.
2. There are grammar, punctuation, and spelling mistakes throughout the manuscript that make it difficult to understand the points the authors are trying to make. One example is towards the end of the introduction where it is unclear to me whether the authors are discussing published research, their hypothesis, or their findings in this manuscript.
3. The authors use different pHs to demonstrate that their construct is dissociated. How are these pHs created and at what timepoint is the pH altered. Does this affect the 4T1/immune compartment on its own?
4. The conditions in each experiment are not always clear to me and need to be clearly stated in either the results section or figure legend. Example include Figure 3a, Figure 4b.
5. The authors activate CD4/CD8 T cells for 48h and conclude that BAY does not affect viability/function. This does not agree with previous findings (PMID: 24930970), and it is important to know if longer activation times to generate T_{eff} (e.g. 96h) affect this.
6. Figure 5a is not cited or discussed.
7. What is the difference experimentally between Figure S10A and Figure S17A? Figure S10A suggested to me that BAY hardly affects 2-NBDG uptake (MFI 5000->4500), while Figure 17A suggests a striking reduction in 2-NBDG uptake (95,000->50,000). Are more experiments needed to resolve this?
8. The UDP-GlnNac rescue in Figure 5 is not convincing- perhaps it is not uptaken well?

9. I cannot find the proliferation data mentioned on line 338.
10. In figure 6h-j it seems the majority of the 4T1 killing is mediated by BAY, with a minor effect of the CD8 T cells. Is this because the CD8 T cells are inhibited? How functional are these cells?
11. In figure 7 it would be helpful to match the colours used across panels.
12. It is not clear to me how much the nanoassembly is important here. Would combined PD-L1/CTLA4 antibodies plus BAY have a similar affect in vivo? The authors only compare only BAY, or only aptamer to the combination.

Reviewer #3 (Remarks to the Author): with expertise in nano-technology for cancer therapy

Immune checkpoint inhibition therapy (ICT) is a novel therapeutic option for triple negative breast cancer (TNBC) that shows promising efficacy in clinical practice. However, post-transcriptional glycosylation of PD-L1 on TNBC cells may significantly impede the immune checkpoint blockade efficacy. In view of the central role of TNBC-intrinsic glycolysis in driving ICT resistance, this manuscript by Ren et al described a nanointegrated strategy for enhancing the ICT efficacy against TNBC with high PD-L1 glycosylation level, which is realized through the combined TNBC-targeted glycolysis inhibition and bispecific PD-L1/CTLA4 blockade using bioresponsive aptamer-based nanoassemblies. The on-demand activation of the nanoassembly in TME could release GLUT1-inhibiting BAY-876 drug and selectively abolish TNBC glycolysis, thus reshaping the immunosuppressive TME to facilitate the bispecific ICT. This study is interesting and of significance, which offers a novel multidisciplinary approach to address clinical problems. I recommend it for Nat Comm after addressing the following issues.

Major issues:

1. Drug release from the nanoassembly was triggered through PAE protonation in acidic environment. The charge status of PAE under pH 7.4 should also be investigated to validate the drug release mechanism.
2. It is important to examine if the acidic pH would affect the bioactivity of the aptamers, including their secondary structures and binding affinity with cognate receptors.
3. The binding of PD-L1 and CTLA-4 aptamers to 4T1 cells and Tregs was measured under

separated conditions, which could not adequately illustrate the targeting ability in the complex tumor microenvironment, the authors need to investigate cell-aptamer binding under co-culture condition.

4. Based on the comparison between KL-11743 and BAY-876 in the co-culture system, the authors concluded that 100 nM BAY-876 could effectively promote the antitumor function of immune cells. However, only CD4+ T and CD8+ T cells were examined, the authors should measure other major relevant immune cell populations such as dendritic cells and M1/M2 macrophages.

5. Only CD4+ T cells and CD8+ T cells were measured in Figure 4. The authors should also monitor the activation status of major APC populations, such as dendritic cells and M1 macrophages.?

6. In Figure 6, the authors detected Foxp3, CD25 and CTLA-4 expression in Tregs. The changes in cytokine expressions should also be included via flow cytometry.

Minor issues:

1. In FigureS2 H1-NMR, the characteristic peak positions in the compounds should be assigned and labeled.
2. Western blot data in Figure4b should be statistically analyzed.
3. The figure texts in Figure5f is too small and needs to be adjusted.
4. Some literatures about the glucose metabolism of macrophages should be added.
5. H&E staining in Figure7 is unclear and should be redone.

Reviewer #4 (Remarks to the Author): with expertise in breast cancer, immunology, metabolism

Ren et al develop a TME-responsive bispecific aptamer-based nanoassemblies encapsulating GLUT1 inhibitor BAY-876 (DNA-PAE@BAY-876), which could synergistically inhibit TNBC-selective glycolysis and promote immunostimulatory in the tumor microenvironment (TME). Although the theory is supported by plenty of experiments, the data presented are generally unconvinced, especially the exclusive tumor model used in this study. The study would benefit from additional experiments to establish the therapeutic relevance of DNA-PAE@BAY-876 in triple negative breast cancer (TNBC).

Major points:

1. How efficient are the PAE-based handles conjugated onto aptPD-L1 and aptCTLA-4? What about the encapsulation efficiency of the BAY-876? And how efficiently is the released BAY-876 entering tumor cells in the TME?
2. The topic of the manuscript focuses on TNBC, however, just a 4T1 tumor model resembling human TNBC is used in in vitro and in vivo studies. Thus, the current data are not convincing enough to illustrate the clinical implication of DNA-PAE@BAY-876.
3. Figure 3c: As the DNA aptamer and the cell membrane are both negatively charged, how are the DNA-PAE nanoassemblies endocytosed by cells under pH of 7.4?
4. Figure 4g: The ratio of IFN-g+CD8+ T cells in the control group under pH of 7.4 is two-fold higher than that under pH of 6.8, why that?
5. TNBC is associated with high risk of distant metastasis such as lung. Can DNA-PAE@BAY-876 treatment reduce metastatic lung burden through remodeling glycometabolism in TME?
6. Effector immune cell deployment (EICD) is very important in immune checkpoint blockade therapy of cancers. The authors can discuss the role of DNA-PAE@BAY-876 in regulating EICD to enhance ICI efficacy.

Minor points:

1. The reference citations in the text should be in accordance with the instructions for authors.
2. In Figure 1b, the CD28 icon is confused with that of BAY-876 due to their identical color.
3. In Figures 2a and 2b, color scales should be shown to understand what shades with different colors indicate.
4. Figure 2e: Which PH condition does the SEM image correspond to?
5. Figure 4b and 4c, what does the "I II III IV V VI VII" mean?
6. For western blotting, dashes should be added to locate the exact position of the marker bands indicated.
7. The authors should revise the language and correct grammatical mistakes as well as formal errors in the text.

REVIEWER COMMENTS

Reviewer #1 (Remarks to the Author): with expertise in nanotechnology, aptamers, cancer immunotherapy

In this work, β -amino ester (PAE)-modified PD-L1 and CTLA-4-antagonizing aptamers (aptPD-L1 and aptCTLA-4) was co-assembled for the targeted delivery of CLUT1 inhibitor BAY-876. The as-synthesized DNA-PAE@BAY-876 could respond to acidic condition to initiate BAY-876 and aptamer release, and then reshaped the TME to enhance ICI response. So many multifunctional nanoplatfroms have been explored for combinational ICT and TME remodeling in cancer treatment. Actually, this work is not the first effort to integrate tumor-glycolysis inhibition and ICT (e.g. ACS Nano, 2020, 14, 11055). Additionally, the aptamer and the BAY-876 used in this study also have been reported earlier. Therefore, it is hard to evaluate the significance and value of the designed systems. Some critical issues should be addressed as follow:

A: Thank you for the thorough review of our manuscript as well as the constructive criticisms. We are grateful for the insights you kindly provided and would like to apologize for the ambiguous descriptions that failed to highlight the novelty of this study. We have carefully read the suggested references and agree with the reviewer that TME remodeling is an emerging paradigm for improving the antitumor efficacy of immunotherapies, which has the potential to substantially improve the response rates to various immunotherapeutic modalities as well as enhancing the treatment efficacy [Adv Mater 34, e2206200 (2022); ACS Nano 14, 11055-11066 (2020); Adv Mater 35, e2300216 (2023); ACS Nano 17, 13461-13473 (2023)]. For instance, in the work by Kolb et al [ACS Nano 14, 11055-11066 (2020)], the authors developed a tumor mitochondrion-targeting nanoparticle for the delivery of PDK-1-inhibiting dichloroacetate prodrugs, which could inhibit the glycolytic activity in tumor cells and reverse the acidic tumor microenvironment through blocking lactate efflux, thus abolishing the immunosuppressive function of Tregs to boost T cell-mediated adaptive antitumor immunity. Overall, these studies collectively demonstrated that therapeutic modulation of the tumor microenvironment through tumor-intrinsic metabolic rewiring could provide novel opportunities for achieving multifaceted anti-tumorigenic benefit including inhibition of immune suppressor cells, enhanced T cell priming efficiency, etc. Interestingly, recent insights revealed that the post-transcriptional glycosylation of membrane-bound PD-L1 has essential roles in inhibiting the binding with the cognate immune checkpoint inhibitors [Cancer

Cell 36, 168-178 e164 (2019)]. As the glycosylation level of membrane PD-L1 is tightly regulated by the tumor-intrinsic glucose metabolism [Mol Carcinog 59, 691-700 (2020); Am J Cancer Res 8, 1837-1846 (2018)]. We postulate that blocking the glucose uptake of TNBC cells could suppress the glycosylation of membrane-bound PD-L1 receptors and facilitate their binding with commercially available PD-L1 antagonizing agents. Surprisingly, we further observed that blocking tumor-intrinsic glucose uptake could profoundly modify the tumor metabolic landscape towards an anti-tumorigenic state, as the elevated glucose levels could induce the metabolic destabilization of tumor-infiltrating Tregs and abolish their immune suppressive function. Therefore, it is anticipated that the tumor-intrinsic glucose metabolism could be a nexus linking their own immunotherapeutic susceptibility and the robustness of adaptive antitumor immunity. Based on this concept, we further developed a nanointegrative platform for enabling the tumor-specific rewiring of glucose metabolism with synchronized CTLA-4 blockade in Tregs, which effectively induced systemic tumor regression in vivo. On the other hand, we agree with the reviewer that immune checkpoint antagonizing aptamers and BAY-876 have indeed been increasingly exploited for immunotherapeutic applications on account of their therapeutic efficacy and safety in the clinics [Small 17, e2102695 (2021); Cancers (Basel) 11, 33 (2019)]. Therefore, the choice of molecularly engineered PD-L1/CTLA-4-antagonizing aptamers and BAY-876 was based on a balanced consideration of (1) molecular cooperativity for constructing biostable nanoassemblies, (2) therapeutic synergism for boosting the antitumor immune responses and (3) enhanced translational potential by using clinically tested components. We have also carried out substantially amount of additional characterizations and experiments to highlight the novelty of the DNA-PAE@BAY-876 nanoassemblies, Typically, we have determined the drug release rate of DNA-PAE@BAY-876 nanoassemblies at different pH conditions as well as explored the targeting ability of DNA-PAE@BAY-876 in depth, extending from the acidity-triggered activation of the nanoassembly in aqueous environment. Moreover, the in vivo pharmacokinetic features of BAY-876 and DNA-PAE@BAY-876 have been comparatively analyzed to substantiate the therapeutic advantage of the nanoassembly. Furthermore, we ensure that all the concerns you kindly pointed out have been properly addressed in the revised manuscript, of which the detailed

responses are outlined below point by point. We hope that these revisions could address your concerns and mobilize this manuscript closer to the high standard of Nature Communications.

1. All flow cytometric analysis lack parallel tests and statistic analysis. Only single test cannot draw a solid conclusion. Meanwhile, the gating strategies in all flow cytometric analysis must be provided. For example, in Figure 4 a, the tumor cells were co-incubated with splenocytes. How the authors separate tumor cells from the immune cells and spleen cells?

A: Thank you for the insights. We are grateful for your advice and would like to explain that all flow cytometric analysis included three independent replicates. However, due to formatting considerations, only one of the dot plots or histograms was shown in the main text or SI. Therefore, we have prepared a data set that included all the original data to enhance the reproducibility of this study, which is resubmitted as one of the supplementary files for your review. Meanwhile, statistical analysis has been conducted for all relevant flow cytometric tests and the figures have been incorporated into the manuscript. In addition, the gating strategies for flow cytometric tests in the present study have been added for clarification. The associated figures are shown here below for your review:

Figure S48. Gating strategy for the FACS tests. a-b were applied for Figure 3a-b; c-d were applied for Figure S29; e was applied for Figure 4f, Figure S18a and Figure S42a; f was applied for Figure S18b, Figure S21a, Figure S35a, Figure S39a and Figure S42b; g was applied for Figure S39b; h was applied for Figure 9g, Figure S18c, Figure S21b, Figure S35b and Figure S42c; i was applied for Figure S35c and Figure S42d; j was applied for Figure S35d and Figure S42f; k was applied for Figure 4g, Figure 8d, Figure 9f and Figure S28b-c; l was applied for Figure 4a and Figure S17.

Figure S49. Gating strategy for the FACS tests. a was applied for Figure 6b; b was applied for Figure S28a; c was applied for Figure 8c and Figure 9e; d was applied for Figure 6c; e was applied for Figure 8a and Figure S42; f was applied for Figure 8b; g was applied for Figure 9h.

Figure S50. Gating strategy for the FACS tests applied for Figure S46 and Figure S47.

is 26 KDa; the molecular weight of Bax is 21 KDa; the molecular weight of Tubulin is 50 KDa. (d-e) Glucose uptake by 4T1 cells or T cells after different treatments under co-culture condition at 37°C for 24 h (n=3). (f-g) Flow cytometric analysis on the expansion of CD4+/CD8+ T cells, IFN- γ +CD8+ T cells in the co-incubation system after different treatment at 37°C for 24 h. (h) Statistical analysis of data shown in panel f-g by flow cytometry (n=3). (1) NC, (2) PAE, (3) aptCTLA-4, (4) aptPD-L1, (5) BAY-876, (6) DNA-PAE, (7) DNA-PAE@BAY-876.

Figure 6. Nanoassembly-mediated abolishment of Treg-mediated immunosuppression. (a) Schematic illustration of the nanoassembly-mediated reprogramming of immunosuppressive Tregs.

(b) Foxp3 expression in Tregs after different treatments at 37°C for 24 h. (c) CTLA-4 and CD25 expression on Tregs after different treatments at 37°C for 24 h. (d-e) Quantitative profiling of Treg-mediated CTL suppression after different treatments with pH7.4/6.8 (n=3) via measuring IFN- γ levels. Tregs suppression (%) = (IFN- γ CD8-IFN- γ Treg+CD8) / IFN- γ CD8 (f-g) ELISA assay on the secretion levels of key immune-related cytokines by immune cells treated with DNA-PAE@BAY-876 under different co-culture conditions at 37°C for 24 h (n=3). (h-j) Viability of 4T1 cells under different co-culture conditions at 37°C for 24 h (n=3). (k) Statistical analysis of data shown in panel b-c by flow cytometry (n=3). (1) Control, (2) aptCTLA-4, (3) BAY-876, (4) B+C, (5) DNA-PAE, (6) DNA-PAE@BAY-876. B+C indicates the mixture of BAY-876 and aptCTLA-4.

Figure 8. Evaluation on DNA-PAE@BAY-876-mediated immunotherapy in vivo. (a-b) Flow cytometry detection on the tumor infiltration of Foxp3⁺ Tregs and IL-10⁺ Tregs. (c) Flow cytometry evaluation on the tumor-infiltration of total immune cells (CD45⁺). (d) Flow cytometry detection on the tumor infiltration of IFN- γ ⁺ CD8⁺ T cells. (e) Statistical analysis of data shown in panel a-d by flow cytometry (n=3). (1) NC, (2) PAE, (3) aptCTLA-4, (4) aptPD-L1, (5) BAY-876, (6) DNA-

PAE, (7) DNA-PAE@BAY-876. (f) Immunofluorescence images on the tumor-infiltration of CD4+ and CD8+ T cells in different groups. (g) Immunofluorescence images of IL-10 in tumor tissues after different treatment. (h) Photographs of lung metastasis inhibition in 4T1 tumor-bearing mice after different treatments.

Figure 9. Therapeutic evaluation of DNA-PAE@BAY-876 on bilateral tumor models. (a) Schematic diagram of 4T1 bilateral tumor model construction and the treatment schedule. (b) Photographical images of the bilateral tumors through the treatment period. (I) NC, (II) PAE, (III) aptCTLA-4, (IV) aptPD-L1, (V) BAY-876, (VI) DNA-PAE, (VII) DNA-PAE@BAY-876. (c) Volume changes of the distal tumors after different treatment (n=3). (d) Survival curves of 4T1 bilateral tumor-bearing mice after different treatment (n=5). (e) Flow cytometry evaluation on the tumor-infiltration of total immune cells (CD45+). (f) Flow cytometry detection on the infiltration status of IFN- γ +CD8+T cells in distal tumor. (g-h) Flow cytometry evaluation on the tumor-infiltration of M1 macrophages (F4/80+CD86+) and CD8+ effector memory T cells (CD44+CD62L-). (i) Statistical analysis of flow cytometry data in panel e-h (n=3). (1) NC, (2) PAE, (3) aptCTLA-4, (4) aptPD-L1, (5) BAY-876, (6) DNA-PAE, (7) DNA-PAE@BAY-876.

Figure S18. Phenotypical changes of various immune cells in the co-incubation system of 4T1 cells and splenocytes after BAY-876 or KL-11743 treatment. (a) Flow cytometry evaluation on the expansion of CD8+/CD4+ T cells. (b) Flow cytometry evaluation of DC maturation (CD11c+MHC-II+). (c) Flow cytometry evaluation on the polarization state of macrophages (F4/80+CD86+). (d) Statistical analysis of flow cytometry data in panel a-c (n=3). (1) PBS, (2) BAY-876, (3) KL-11743, (4) BAY-876+KL-11743.

Figure S21. Nanoassembly treatment boosted the activation of APCs. Flow cytometry evaluation of DC maturation (CD11c⁺MHC-II⁺) (a) and M1 polarization of macrophages (F4/80⁺CD86⁺) after various treatment. (b) in the co-incubation system after different treatment. (c) Statistical analysis of flow cytometry data in panel a-b (n=3).

Figure S28. Phenotypical changes of CD8+ T cells under different co-culture conditions with different treatment. (a) Frequency of CD8+ T cells in the co-incubation system of 4T1 cells, CD8+ T cells and Tregs after different treatment (n=3). (b) Frequency of IFN- γ + CD8+ T cells in the co-incubation system of 4T1 cells and CD8+ T cells after different treatment (n=3). (c) Frequency of IFN- γ + CD8+ T cells in the co-incubation system of 4T1 cells, Tregs and CD8+ T cells after different treatment (n=3). (1) Control, (2) aptCTLA-4, (3) BAY-876, (4) B+C, (5) DNA-PAE, (6) DNA-PAE@BAY-876. B+C was BAY-876 combined with aptCTLA-4.

Figure S29. Activation status of T cells in the co-incubation system of 4T1 cells and spleen cells after different treatment. (a) IFN- γ secretion in the co-incubation system of 4T1 cells and spleen cells after different treatment. (b) IL-10 secretion in the co-incubation system of 4T1 cells and spleen cells after different treatment. (c) Statistical analysis the flow cytometric data in panel a-b (n=3). B+C means BAY-876 combined with aptCTLA-4.

Figure S35. Evaluation on DNA-PAE@BAY-876-mediated immunotherapy in 4T1 tumor-bearing mice. (a) Flow cytometry detection on the tumor infiltration of DCs (CD11c⁺MHC-II⁺). (b) Flow cytometry detection on the tumor-infiltration of M1 macrophages (F4/80⁺CD86⁺). (c) Flow cytometry detection on the tumor infiltration of M2 macrophages (F4/80⁺CD206⁺). (d) Flow cytometry detection on the tumor infiltration of MDSCs (CD11b⁺GR1⁺). (e) Statistical analysis of a-d data by flow cytometry (n=3). (1) NC, (2) PAE, (3) aptCTLA-4, (4) aptPD-L1, (5) BAY-876, (6) DNA-PAE, (7) DNA-PAE@BAY-876.

Figure S39. Effector immune cell deployment (EICD) in 4T1 tumor-bearing mice after various treatment. (a) Flow cytometry analysis on the maturation status of lymph node-resident DCs (CD11c+MHC-II+). (b) Flow cytometry analysis on the lymph node-resident central memory T cell populations (CD197+CD62L+). (c-e) Serum cytokine levels in mice after 21 days of different treatment (n=4). (f-g) Statistical analysis of the flow cytometric data in panel a-b (n=3).

Figure S42. Comparative analysis on the antitumor efficacy of DNA-PAE@BAY-876 and anti-PD-L1/anti-CTLA-4 antibody in 4T1 tumor-bearing mice. (a) Flow cytometry detection on the tumor infiltration of CD8⁺ T cells and CD4⁺ T cells. (b) Flow cytometry detection on the tumor infiltration of mature DCs (CD11c⁺MHC-II⁺). (c) Flow cytometry detection on the tumor-infiltration of M1 macrophages (F4/80⁺CD86⁺). (d) Flow cytometry detection on the tumor infiltration of M2 macrophages (F4/80⁺CD206⁺). (e) Flow cytometry detection on the tumor infiltration of Tregs (CD4⁺CD25⁺Foxp3⁺). (f) Flow cytometry detection on the tumor infiltration of MDSCs (CD11b⁺GR1⁺). (1) Control, (2) ab, (3) apt, (4) ab+BAY-876, (5) DNA-PAE@BAY-876. ab was anti-PD-L1 antibody and anti-CTLA-4 antibody; apt was aptPD-L1 and aptCTLA-4.

Figure S46. Evaluation on DNA-PAE@BAY-876-mediated immunotherapeutic effect in MDA-MB-231 tumor-bearing humanized HSC-NOG-EXL mice. (a) Flow cytometry detection on the tumor infiltration of CD8⁺ T cells and CD4⁺ T cells. (b) Flow cytometry detection on the tumor infiltration of DCs (CD11c+HLA-DR⁺). (c) Flow cytometry detection on the tumor-infiltration of M1 macrophages (CD14+CD80⁺). (d) Flow cytometry detection on the tumor infiltration of M2 macrophages (CD14+CD206⁺). (e) Flow cytometry detection on the tumor infiltration of Tregs (CD4+CD25+Foxp3⁺). (f) Flow cytometry detection on the tumor infiltration of MDSCs (HLA-DR-CD11b⁺). (g) Statistical analysis of the flow cytometric data in panel a-f (n=3). (1) NC, (2) PAE, (3) aptCTLA-4, (4) aptPD-L1, (5) BAY-876, (6) DNA-PAE, (7) DNA-PAE@BAY-876.

Figure S47. Effector immune cell deployment (EICD) in MDA-MB-231 tumor-bearing humanized HSC-NOG-EXL mice. (a) Flow cytometry detection on the lymph node-resident DCs (CD11c+HLA-DR+). (b) Flow cytometry detection on the lymph node-resident central memory T cells (CD45RA-CD197+). (c-e) Cytokine secretion levels in humanized mice serum after 21 days of different treatment (n=3). (f-g) Statistical analysis of the flow cytometric data in panel a-b (n=3).

2. In Figure 2 g, the name of the DNA-PAE was mislabeled.

A: Thank you for the correction. We are really sorry for the careless mistakes and corrected the label accordingly, which should be DNA-PAE. The corrected figure is also shown here below for your review:

Figure 2. Physical and chemical characterization of DNA-PAE@BAY-876 nanoassemblies. (a-b) Secondary structure of PD-L1 and CTLA-4-antagonizing aptamers. Color intensity indicates base-pairing probability at equilibrium. (c-d) TEM image of DNA-PAE@BAY-876 nanoassemblies with different pH. (e) SEM image of DNA-PAE@BAY-876 nanoassemblies at pH7.4. (f) Size changes of DNA-PAE nanoassemblies before and after drug loading. (g) Zeta potential changes of DNA-PAE nanoassemblies before and after drug loading. (h) UV-vis spectroscopic features of

DNA-PAE nanoassemblies before and after drug loading. (i) Ultraviolet absorption of BAY-876 at different concentrations. (j) CMC of DNA-PAE nanoassemblies under pH 7.4. (k) Stability of DNA-PAE@BAY-876 nanoassemblies in PBS supplemented with 10% serum. (l) Critical pH value of DNA-PAE nanoassemblies showing their acidity-triggerable dissociation. (m) Size changes of DNA-PAE nanoassemblies at pH7.4 and 6.8. (n) Release kinetics of BAY-876 from DNA-PAE@BAY-876 nanoassemblies at pH7.4 and 6.8.

3. In Figure 2h, the UV-Vis spectrum of pure BAY-876 should be provided.

A: Thank you for the advice. We agree with the reviewer that adding the UV-vis spectrum of pristine BAY-876 could more accurately depict the construction of the nanoassemblies and have redone the UV-vis analysis as suggested, which is represented by the blue line in Figure 2h. As shown in Figure 2h, pristine BAY-876 showed an evident characteristic absorption peak at around 330 nm. However, the BAY-876-characteristic peak in DNA-PAE@BAY-876 showed a slight blue shift, while the DNA-characteristic absorption peak in DNA-PAE@BAY-876 has slightly red-shifted compared with pristine DNA-PAE (red line). The peak shift was attributed to the π - π interaction between DNA and BAY-876 [European J Org Chem 2020, 2321-2329 (2020)], which confirmed the successful integration of BAY-876 in the self-assembled nanostructure.

The related data and discussions are shown here below for your review:

“The BAY-876 loading was further investigated by UV-vis spectroscopy, which showed that the DNA-PAE@BAY-876 nanoassemblies have retained the characteristic peaks of both aptamers and BAY-876 (Figure 2h), suggesting the successful construction of the composite self-assembly structures. Nevertheless, the characteristic peaks have slightly shifted compared with pristine DNA-PAE and BAY-876, which was ascribed to the π - π interaction in between (Figure 2h-i).”

Figure 2. Physical and chemical characterization of DNA-PAE@BAY-876 nanoassemblies. (a-b) Secondary structure of PD-L1 and CTLA-4-antagonizing aptamers. Color intensity indicates base-pairing probability at equilibrium. (c-d) TEM image of DNA-PAE@BAY-876 nanoassemblies with different pH. (e) SEM image of DNA-PAE@BAY-876 nanoassemblies at pH7.4. (f) Size changes of DNA-PAE nanoassemblies before and after drug loading. (g) Zeta potential changes of DNA-PAE nanoassemblies before and after drug loading. (h) UV-vis spectroscopic features of

DNA-PAE nanoassemblies before and after drug loading. (i) Ultraviolet absorption of BAY-876 at different concentrations. (j) CMC of DNA-PAE nanoassemblies under pH 7.4. (k) Stability of DNA-PAE@BAY-876 nanoassemblies in PBS supplemented with 10% serum. (l) Critical pH value of DNA-PAE nanoassemblies showing their acidity-triggerable dissociation. (m) Size changes of DNA-PAE nanoassemblies at pH7.4 and 6.8. (n) Release kinetics of BAY-876 from DNA-PAE@BAY-876 nanoassemblies at pH7.4 and 6.8.

4. In Page 9 Line 178, the authors mentioned that “the DNA-PAE@BAY-876 nanoassembly has high stability under physiological conditions...”. The reviewer think that the 26% leakage is not negligible. The reviewer doubts that the nanoassembly could be defined as a high stability system.

A: Thank you for your concern. We fully understand the reviewer’s concern regarding the stability of the aptamer-based self-assembled nanostructure and would like to explain that the relatively high drug leakage (~26%) happened after prolonged incubation of 48 h, which is due to the spontaneous passive diffusion of the BAY-876 molecules in aqueous environment despite its hydrophobicity [Nano Lett 22, 8735-8743 (2022); Nano Lett 22, 6418-6427 (2022)]. Importantly, we would like to note that the potential safety risks associated with the passive BAY-876 diffusion was avoided by the efficient tumor-targeting effect of the nanoassembly in vivo. According to the pharmacokinetic and biodistribution analysis in vivo, the DNA-PAE@BAY-876 nanoassemblies showed significantly longer blood half-life compared with pristine BAY-876, and showed efficient tumor-specific accumulation within 6 h post intravenous injection. By referring to the drug release profiles of DNA-PAE@BAY-876 under pH 7.4, the relative drug leakage rate was less than 9% at 6 h. We also would like to state that drug leakage from DNA-PAE@BAY-876 nanoassemblies gradually reached a plateau after 20 h at pH 7.4 (Figure 2n). Meanwhile, pharmacokinetic analysis showed that most of the DNA-PAE@BAY-876 nanoassemblies have already been cleared from the blood after 10 h and only less than 10% of the injected dose remained after 20 h (Figure S30a), immediately suggesting that the risk of potential health risks of uncontrollable drug leakage was negligible. This is also consistent with the in vivo biocompatibility analysis that the DNA-PAE@BAY-876 nanoassemblies did not cause any obvious toxic effect after intravenous injection.

Overall, these data collectively demonstrated that the relatively high drug leakage from the nanoassembly in the long term would not affect its antitumor efficacy and safety in vivo.

Figure S30. In vivo pharmacokinetic analysis. (a) Pharmacokinetic profiles of DNA-PAE@BAY-876 and BAY-876 in mice after intravenous injection. (b) Systemic distribution of DNA-PAE@BAY-876/BAY-876 at 0/6/12/24h post intravenous injection.

Figure 2. Physical and chemical characterization of DNA-PAE@BAY-876 nanoassemblies. (a-b) Secondary structure of PD-L1 and CTLA-4-antagonizing aptamers. Color intensity indicates base-pairing probability at equilibrium. (c-d) TEM image of DNA-PAE@BAY-876 nanoassemblies with different pH. (e) SEM image of DNA-PAE@BAY-876 nanoassemblies at pH7.4. (f) Size changes of DNA-PAE nanoassemblies before and after drug loading. (g) Zeta potential changes of DNA-PAE nanoassemblies before and after drug loading. (h) UV-vis spectroscopic features of

DNA-PAE nanoassemblies before and after drug loading. (i) Ultraviolet absorption of BAY-876 at different concentrations. (j) CMC of DNA-PAE nanoassemblies under pH 7.4. (k) Stability of DNA-PAE@BAY-876 nanoassemblies in PBS supplemented with 10% serum. (l) Critical pH value of DNA-PAE nanoassemblies showing their acidity-triggerable dissociation. (m) Size changes of DNA-PAE nanoassemblies at pH7.4 and 6.8. (n) Release kinetics of BAY-876 from DNA-PAE@BAY-876 nanoassemblies at pH7.4 and 6.8.

5. In Figure 3c, AbPD-L1 could bond to 4T1 cell surface under pH 7.4 but not pH 6.8. Why? In addition, no DNA-PAE and AbPD-L1 could enter HC11 cells? No endocytosis occurred? Importantly, the different cellular uptakes in two different cell lines cannot confirm the specific interaction between PD-L1 and aptPD-L1.

A: Thank you for your advice. We have carefully examined the experimental procedures for the cellular experiment in Figure 3c and concluded that the seemingly weak red fluorescence in the AbPD-L1+pH 6.8 group was due to prolonged incubation of the fluorescent antibodies under the acidic pH of 6.8 for 24 h, which may significantly enhance the risk of spontaneous fluorescence quenching and antibody denaturation [J Phys Chem Lett 14, 3898-3906 (2023)]. We are really sorry for our mistakes in experiment setup and have redone the fluorescence imaging using an indirect immunocytochemical staining approach, in which the cells were first incubated with anti-PD-L1 antibody for 2 h and then treated with secondary fluorescent antibodies for 2 h, thus avoiding the quenching of the fluorescent moieties under acidic conditions. Indeed, the results showed that AbPD-L1 could efficiently bind to 4T1 cell surface under both pH 7.4 and 6.8. Meanwhile, we would like to explain that the cellular fluorescence regarding sample uptake may be difficult to differentiate due to the insufficient resolution. Therefore, we have redone the experiment in Figure 3c using a high-resolution fluorescence microscope to enhance the data quality. Generally speaking, the fluorescence levels in HC11 cells after treatment with DNA-PAE and AbPD-L1 both remained low, which was in line with their low PD-L1 expression status. In contrast, substantial amount of FITC fluorescence has appeared in the cellular compartment of the DNA-PAE group after 2 h of incubation, showing that DNA-PAE entered 4T1 cells through PD-L1-mediated endocytosis at pH 7.4 and then complexed with PD-L1 in the nucleus, while aptPD-L1 predominantly complexed with membrane-bound PD-L1 when the environmental pH of the culture media was adjusted to 6.8,

indicating that the acidity-triggered disassembly of DNA-PAE was crucial for enabling the spatial-temporally synchronized BAY-876 delivery and PD-L1 immune checkpoint blockade against 4T1 cells while sparing healthy cells.

The related data and discussions are shown here below for your review:

“We then incubated 4T1/HC11 cells with DNA-PAE nanoassemblies under pH 7.4 or pH 6.8 to investigate their nano-biointeraction via confocal laser microscopy (CLSM) (Figure 3c). Interestingly, abundant Cy5-labeled aptPD-L1 bond to 4T1 cell surface under pH 6.8, while the majority of aptPD-L1 was attached to 4T1 nuclei when incubated under pH 7.4. The distinct aptPD-L1 distribution patterns were immediate evidence for the acidity-responsiveness of DNA-PAE nanoassemblies. Under the TME-like acidic pH of 6.8, DNA-PAE nanoassemblies were rapidly dissociated into free aptamer-PAE conjugates, allowing their direct complexation with PD-L1 on 4T1 cytoplasmic membrane. However, under the physiological pH of 7.4, the DNA-PAE nanoassemblies were endocytosed by 4T1 cells as a whole and subsequently dissociated in the acidic tumor lysosomes, which would then complex with the PD-L1 in tumor nucleus⁶. Furthermore, PD-L1-negative HC11 cells generally showed low fluorescence deposition after treatment with both DNA-PAE nanoassemblies and AbPD-L1, which was in line with their intrinsically low PD-L1 expression status and again validated the therapeutic mechanism of DNA-PAE.”

Figure 3. Evaluation on the TNBC-targeting effect of DNA-PAE@BAY-876 in vitro. (a) Flow cytometric analysis on the cell binding behavior of aptPD-L1 to 4T1/HC11 cells under pH 7.4. (b) Cell binding behavior of aptCTLA-4 to activated/inactivated T cells under pH 7.4. (c) Interaction between 4T1 or HC11 cells and DNA-PAE nanoassemblies under different environmental pH conditions at 37°C for 2 h. (d) DNA-PAE@BAY-876-mediated GLUT1 inhibition in 4T1 cells and HC11 cells at 37°C for 24 h. (e-f) Dose-dependent toxicity of DNA-PAE@BAY-876 to 4T1 cells and HC11 cells at 37°C for 24 h (n=3). (g-i) Toxicity of BAY-876 to major immune cell populations at different pH conditions (n=3). (I) pH7.4+activation, (II) pH7.4+inactivation, (III) pH6.8+activation, (IV) pH6.8+inactivation. (j-k) Immunofluorescence analysis of GLUT1 and GLUT3 in 4T1 tumors. (l-m) Glucose uptake of 4T1 cells and T cells in the co-incubation system after treatment with BAY-876 or KL-11743 (n=3).

Meanwhile, to verify the molecular-specific affinity between aptPD-L1/ unglycosylated PD-L1 and aptCTLA-4/unglycosylated CTLA-4 protein pairs, we proactively conjugated 50 nM biotin-modified mouse PD-L1 and mouse CTLA-4 to streptavidin modified magnetic beads, respectively, followed by incubation with different concentrations of Cy5-aptPD-L1 or FAM-aptCTLA-4. Owing to the conjugation of the fluorescent moieties, the binding ability of the aptamers could thus be quantitatively determined by flow cytometry. As shown in Figure S4, the dissociation constant of aptCTLA-4 to unglycosylated CTLA-4 was 168.3 nM, while the dissociation constant of aptPD-L1 to unglycosylated PD-L1 was 122.4 nM. The fluorescent-based flow cytometric analysis was in line with the observations in previous reports [Anal Chim Acta 1185, 339066 (2021); Mol Ther Nucleic Acids 8, 520-528 (2017)] confirming that both aptCTLA-4 and aptPD-L1 had high affinity with their cognate receptors.

The related data and discussions are shown here below for your review:

“The affinity of aptPD-L1 and aptCTLA-4 was detected by a magnetic bead binding assay, for which biotin-modified mouse PD-L1 and mouse CTLA-4 were first separately conjugated onto streptavidin modified magnetic beads and then incubated with different concentrations of Cy5-aptPD-L1 or FAM-aptCTLA-4. The dissociation constants of aptPD-L1 and aptCTLA-4 to their cognate protein receptors were calculated to be 122.4 nM and 168.3 nM, respectively, supporting the binding affinity and specificity of the selected aptamers^{13, 14} with the designated immune checkpoints (Figure S4).”

“Determination of dissociation constant of aptCTLA-4 or aptPD-L1: 1 mL streptavidin-coated magnetic beads were stood for 3 min with the magnet, then washed by PBS for three times. The biotin-modified mouse CTLA-4 or mouse PD-L1 protein with a final concentration of 50 nM was mixed with the above magnetic beads, and incubated on the rotator for 60 min. After cleaning with PBS for three times, FAM-aptCTLA-4 or Cy5-aptPD-L1 with different concentrations were added into the above mixture, then incubated at room temperature for 30 min. After washing twice with PBS, the fluorescence level was measured by flow cytometry (CytoFLEX, Beckman Coulter). The dissociation constants of aptPD-L1 and aptCTLA-4 were calculated by the formula $Y = B_{\max} X / (K_D + X)$.”

Figure S4. Evaluation on the binding affinity of aptCTLA-4 and aptPD-L1 with their cognate receptors. (a) The dissociation constant of FAM-aptCTLA-4 to CTLA-4 under graded concentrations (n=3). (b) The dissociation constant of Cy5-aptPD-L1 to PD-L1 protein under graded concentrations (n=3).

In addition, to further validate the targeting capacity of aptPD-L1 and aptCTLA-4 under clinically relevant conditions, we established co-incubation systems comprising 4T1 cells and splenic immune cells, which were then subjected to the separate treatment of aptPD-L1 and aptCTLA-4. As shown in Figure S9, aptPD-L1 predominantly complexed with 4T1 cells (red line) rather than splenic immune cells (green line), while aptCTLA-4 preferentially bond to Tregs (red line) instead of 4T1 cells (green line), indicating again that these two aptamers had good target specificity.

The related data and discussions are shown here below for your review:

“Furthermore, we established co-incubation systems comprising 4T1 cells and splenic immune cells and confirmed that PAE-modified aptPD-L1 and aptCTLA-4 could preferentially complex with 4T1 cells and Tregs, respectively, again validating the application potential of the PAE-modified aptamers for robust ICT against TNBCs (Figure S9).”

“Binding of aptPD-L1 and aptCTLA-4 in co-culture system: 4T1 cells and splenic immune cells were mixed in the 1.5 mL centrifuge tube at a ratio of 1:30 and treated with 5% BSA for 30 min, followed by the treatment of Cy5-aptPD-L1 or FAM-aptCTLA-4 for another 30 min of incubation. After washing twice with PBS, PC7-anti-CD45 antibody, PE-anti-CD4 antibody and APC-anti-CD25 antibody was added into above cells. Finally, the fluorescence intensity on 4T1 cells or immune cells was measured by flow cytometry (CytoFLEX, Beckman Coulter).”

Figure S9. Targeting ability of aptPD-L1 (a) and aptCTLA-4 (b) in the co-incubation system of 4T1 cells and splenic immune cells.

6. The high PD-L1 expression of the 4T1 cell line used in this work must be verified.

A: Thank you for your advice. To validate the proposed therapeutic mechanism of the nanoassembly in the present study, PD-L1 expression in 4T1 cells and HC11 cells was comparatively analyzed using three different methods. As shown by the flow cytometric tests in Figure S10a, PD-L1 expression (red line) in 4T1 cells was evidently higher than HC11 cells (green line). This was also supported by CLSM analysis that 4T1 cells showed much stronger PD-L1-associated fluorescence than HC11 cells (Figure S10b). We further carried out quantitative WB analysis on PD-L1 abundance and confirmed that the amount of PD-L1 in 4T1 cells was around 1.24-fold higher than HC11 cells (Figure S10c). These data collectively validated the elevated PD-L1 expression in 4T1 cells than normal cells and supported our proposed ICT mechanism.

The related data and discussions are shown here below for your review:

“Specifically, we observed that the amount of aptPD-L1 bound to 4T1 cell surface was higher than HC11 cells, attributing to the upregulated PD-L1 expression level in 4T1 cells (Figure S10).”

Figure S10. Evaluation on the PD-L1 expression in TNBC cells. The PD-L1 expression in 4T1 cells and HC11 cells detected by flow cytometry (a), fluorescence confocal microscopy (b) and western blot (c).

7. The CD45 fluorescence in Figure 3j-3k is very weak. It cannot be concluded from these data that “CD45+ lymphocytes showed high GLUT3 expression...” (Line 233-234).

A: Thank you for your insight. We agree with the reviewer’s concern regarding the GLUT3 expression in CD45+ lymphocytes and carried out substantial amount of additional characterizations to address this issue, which was possibly due to the low resolution of the IHC images. Firstly, we carried out immunofluorescence staining of GLUT1/GLUT3 and CD45 on extracted tumor tissues using a high-resolution microscope. As shown by the immunofluorescence images, the CD45 fluorescence showed negligible overlap with GLUT1 fluorescence, but was highly overlapped with GLUT3 fluorescence. In addition to the visual observations via immunofluorescence staining, we have extracted the tumor tissues and quantitatively analyzed GLUT1 and GLUT3 expression in TNBC and immune cell populations via flow cytometry. As shown in Figure S15, the GLUT1 levels in 4T1 tumor cells were 8.22-fold higher than CD45+ lymphocytes, while the GLUT3 expression in the two cell populations remained at the similar level, indicating that tumor cells tended to express GLUT1 while CD45+ lymphocytes were associated with elevated GLUT3 expression. These results collectively demonstrated that high GLUT3 expression is a common trait of tumor-infiltrating CD45+ lymphocytes, which supported the TME remodeling effect of the nanoassemblies for boosting the immunotherapeutic efficacy.

The related data and discussions are shown here below for your review:

“Consistent with the glucose metabolic patterns in different cell populations in vitro, immunofluorescence imaging and quantitative data by flow cytometry on 4T1 tumors extracted from Balb/c mouse models indicated that tumor-infiltrating CD45+ lymphocytes were generally associated with higher GLUT3 expression levels, while tumor cells tended to express high levels of GLUT1 for glucose uptake (Figure 3j-k and Figure S15).”

“GLUT1/GLUT3 immunofluorescence imaging: The 4T1 tumor-bearing Balb/c mouse model was constructed, and the tumors sized about 600 mm³ were collected. After the tumors were frozen-sectioned, the sections were fixed with 4% paraformaldehyde at room temperature for 30 min. After cleaning with PBS, the sections were blocked with 5% BSA for 2 h, then added with anti-GLUT1 antibody/anti-GLUT3 antibody at 4°C overnight, washed twice with PBS, added with fluorescent secondary antibody at room temperature for 2 h and stained for 10 min with DAPI in the dark, finally sealed with anhydrous glycerol for observation by CLSM (Leica TCS SP8, Germany).”

“Quantitative analysis of immunofluorescence images for determining GLUT1 and GLUT3 expression: The 4T1 tumor-bearing Balb/c mouse model was constructed, and the tumors sized about 600 mm³ were collected. The 4T1 tumor was washed twice with PBS, placed on a 0.4 μm filter membrane with adding high-glucose DMEM medium, and carefully pulverized with the syringe head, then mixed with 5 mL red blood cell lysis buffer and stood for 10 min, and centrifuged at 2000 rpm for 5 min. The cells were collected and sealed by 10% FBS for 2 h, then mixed with anti-GLUT1 antibody (rabbit host) and anti-GLUT3 antibody (mouse host) overnight at 4°C, followed by incubation with Cy5-labeled rabbit second antibody or FAM-labeled mouse second antibody for 2 h at room temperature. After washing twice with PBS, the cells were assayed by flow cytometry (CytoFLEX, Beckman Coulter).”

Figure 3. Evaluation on the TNBC-targeting effect of DNA-PAE@BAY-876 in vitro. (a) Flow cytometric analysis on the cell binding behavior of aptPD-L1 to 4T1/HC11 cells under pH 7.4. (b) Cell binding behavior of aptCTLA-4 to activated/inactivated T cells under pH 7.4. (c) Interaction between 4T1 or HC11 cells and DNA-PAE nanoassemblies under different environmental pH conditions at 37°C for 2 h. (d) DNA-PAE@BAY-876-mediated GLUT1 inhibition in 4T1 cells and HC11 cells at 37°C for 24 h. (e-f) Dose-dependent toxicity of DNA-PAE@BAY-876 to 4T1 cells and HC11 cells at 37°C for 24 h (n=3). (g-i) Toxicity of BAY-876 to major immune cell populations at different pH conditions (n=3). (I) pH7.4+activation, (II) pH7.4+inactivation, (III) pH6.8+activation, (IV) pH6.8+inactivation. (j-k) Immunofluorescence analysis of GLUT1 and GLUT3 in 4T1 tumors. (l-m) Glucose uptake of 4T1 cells and T cells in the co-incubation system after treatment with BAY-876 or KL-11743 (n=3).

Figure S15. GLUT1 (a) and GLUT3 (b) distribution in different cell populations in vivo by flow cytometry.

8. Values of x and y coordinate axis in Figure 5g were very unclear.

A: Thank you for your reminder. We have reformatted the values for both x and y axes in Figure 5g and ensure that they are clearly readable under default 100% zoom view.

Figure 5. Nanoassembly-enhanced 4T1 cell recognition and binding by aptPD-L1. (a) Schematic illustration on the nanoassembly-mediated metabolic rewiring of 4T1 cells for enhanced ICT. (b) Evaluation on the dose-dependent impact of BAY-876 on PD-L1 glycosylation in 4T1 cells with pH7.4/6.8 at 37°C for 24 h. The concentration of IFN- γ was 1 μ g·mL⁻¹. Note: the molecular weight of PD-L1 is 33-70 KDa; the molecular weight of Tubulin is 50 KDa. (c) Impact of DNA-PAE@BAY-876 on the PD-L1 glycosylation level in multiple cell types at 37°C for 24 h. The concentration of DNA-PAE@BAY-876 was 100 nM and the concentration of IFN- γ was 1 μ g·mL⁻¹. (d) Impact of UDP-GlcNAc and F6P on the PD-L1 glycosylation levels in 4T1 cells after BAY-876 treatment with high/low-glucose media. The concentration of BAY-876, UDP-GlcNAc and

IFN- γ was 100 nM, 0.1 mM and 1 $\mu\text{g}\cdot\text{mL}^{-1}$, respectively. (e) UDP-GlcNAc abundance in 4T1 cells under different conditions (n=4). (I, V) Control; (II, VI) BAY-876; (III, VII) BAY-876+F6P; (IV, VIII) BAY-876+UDP-GlcNAc. (f) Schematic illustration of Hexosamine Biosynthesis Pathway (HBP). (g) 3D plot on the correlation between UDP-GlcNAc and F6P with 2-NBDG after treatment by BAY-876 under graded concentrations. (h-k) Impact of BAY-876 on the competitive combination between aptPD-L1/PD1 with PD-L1 on 4T1 cell surface.

Reviewer #2 (Remarks to the Author): with expertise in immuno-metabolism

Ren et al. present a new strategy to target triple negative breast cancer using PD-L1 and CTLA-4-antagonizing aptamers in nanoassemblies that carry BAY-876 (an inhibitor of GLUT1). They suggest that their strategy specifically targets the tumour microenvironment, as the low pH triggers PAE protonation and nanoassembly dissociation. This strategy reduces tumour growth in mice inoculated with 4T1 cells in the flank. The authors suggest that GLUT-1 inhibition targets 4T1 cells specifically as these cells have high expression of GLUT-1. GLUT-1 inhibition reduces the viability of 4T1 cells and reduces glycosylation of PD-L1. I have several major concerns that I have summarised below.

A: Thank you for the constructive comments and suggestions. Based on your advices we have carried out substantial amount of characterizations to address your concerns. Meanwhile, the relevant descriptions have been optimized to more clearly demonstrate the therapeutic effect of the nanoassembly system while avoiding misunderstanding. We ensure that every point you raised has been properly addressed in the revised manuscript. The detailed responses are shown here below for your review.

1. It is unclear to me when the authors are using cell lines or tumours. If the authors are using cell lines this is not the 'tumor microenvironment'. How the authors setup their experiments generally needs clarifying.

A: Thanks for your correction. We are really sorry for the lack of detailed descriptions regarding the experimental setup as both 4T1 cell lines and 4T1 tumor-bearing mice have been used in the present study for mechanistic and efficacy evaluations. Therefore, we have revamped the wording in relevant places to avoid misunderstanding. Specifically, we agree with the reviewer that abnormal acidity is but one of the fundamental traits of the tumor microenvironment and thus changed the term "tumor microenvironment" in the in vitro experiment to other more objective forms such as "with mild acidity resembling TNBCs in the clinics".

Representative examples:

“2. Targeting ability of DNA-PAE@BAY-876 in vitro”

“To test the bispecific binding capability of the DNA-PAE@BAY-876 nanoassemblies after mild acidity-mediated activation...”

“supporting their therapeutic robustness in the acidic environment...”

“To further investigate the cellular specificity and safety of the BAY-876-dependent glycolysis inhibition strategy in vitro...”

“The evidence above supported the therapeutic capacity of the BAY-876-containing nanoassembly to rebalance glucose competition between TNBC cells and immune cells for mounting adaptive antitumor immune responses.”

Meanwhile, we have also added detailed descriptions regarding the experimental setup of these experiments for clarification. Some of the representative examples are shown here below for your review:

“**Preparation of pH6.8 medium:** Lactic acid solution (20 mM) was added into high-glucose DMEM medium (pH7.4) until the pH value dropped to 6.8, during which the medium pH was monitored using a pH detector.”

“**The establishment of 4T1 tumor-bearing Balb/c mouse model:** Firstly, 100 μL 2×10^6 4T1 cells were injected subcutaneously into each Balb/c mouse, when the tumor grew to about 100 mm^3 , the mice were randomly divided into 7 groups at 6 mice per group and subjected to different treatment (PBS, 24.01 $\mu\text{g} \cdot \text{mL}^{-1} \cdot \text{g}^{-1}$ PAE, 12.35 $\mu\text{g} \cdot \text{mL}^{-1} \cdot \text{g}^{-1}$ aptCTLA-4, 12.22 $\mu\text{g} \cdot \text{mL}^{-1} \cdot \text{g}^{-1}$ aptPD-L1, 0.05 $\mu\text{g} \cdot \text{mL}^{-1} \cdot \text{g}^{-1}$ BAY-876, 48.31 $\mu\text{g} \cdot \text{mL}^{-1} \cdot \text{g}^{-1}$ DNA-PAE, 48.87 $\mu\text{g} \cdot \text{mL}^{-1} \cdot \text{g}^{-1}$ DNA-PAE@BAY-876) through intravenous injection once every two days, and tumor volume and body weight of every mouse were recorded periodically. After 21 days of treatment, all mice were sacrificed, their tumor tissues, major organs and lymph nodes were collected for post-analysis while the blood of mice from the eyeball was collected using the heparin tube and then serum was collected after centrifugation for ELISA assay of multiple cytokines. For the 4T1 tumor-bearing bilateral Balb/c mouse model, 100 μL 5×10^6 4T1 cells were injected subcutaneously into the opposite side of the

primary tumors after 21 days of treatment, and the murine survival curves were plotted across a period of 60 days.

A portion of the collected tumor tissue and lymph nodes were washed twice with PBS, put into 0.4 μm filter membrane with adding high-glucose DMEM medium, and carefully ground with the syringe head, then mixed with 5 mL red blood cell lysis buffer and stood for 10 min, and centrifuged at 2000 rpm for 5 min. The supernatant was removed, the cells were re-suspended with PBS, then stained with corresponding fluorescent antibodies for 30 min to detect the infiltration of various immune cells by flow cytometry.

Moderate amount of the tumor tissue was ground to the powder using liquid nitrogen, and then lysed with RIPA lysate (containing 1% PMSF) for 30 min at 4°C and centrifuged at 12000 rpm for 10 min. The collected supernatant was used for Western Blotting or ELISA to detect the expression of corresponding protein or the abundance of glycolysis-related metabolites.

Another portion of the tumor tissue was frozen-sectioned, fixed with 4% paraformaldehyde at room temperature for 30 min. After permeation with 0.5% Triton X-100 for 10 min, the sections were incubated with TUNEL test solution at 37°C for 60 min, then added with DAPI for 10 min in the dark and sealed with glycerol anhydrous after washing with PBS. The mounted tissue sections were finally imaged by a laser confocal microscope (Leica TCS SP8, Germany).

The remaining tumor tissues were frozen-sectioned, fixed with 4% paraformaldehyde at room temperature for 30 min. After cleaning with PBS, the sections were blocked with 5% BSA for 2 h, incubated with fluorescent antibodies at 4°C overnight, added with DAPI for 10 min in the dark, then sealed with glycerol anhydrous after washing with PBS. The mounted tissue samples were finally imaged by a laser confocal microscope (Leica TCS SP8, Germany).”

2. There are grammar, punctuation, and spelling mistakes throughout the manuscript that make it difficult to understand the points the authors are trying to make. One example is towards the end of the introduction where it is unclear to me whether the authors are discussing published research, their hypothesis, or their findings in this manuscript.

A: Thank you for the reminder. We are really sorry for the linguistic mistakes and made corrections accordingly to improve the readability of the manuscript. Specifically, we have revamped the wording in the introduction section to highlight our own findings and avoid misunderstanding. Some representative examples are shown here below for your review.

Linguistic corrections:

“Interestingly, immune checkpoint inhibition therapy (ICT), a form of immunotherapy that restores antitumor immunity by blocking negative immune checkpoints, has shown promise for effective TNBC treatment.”

“From a clinical perspective, TNBC presents a plethora of immunogenic traits...”

“Consequently, aptamer-based nanotherapeutics could be an alternative strategy to develop more advanced ICTs for overcoming the challenges in TNBC immunotherapy.”

“which may provide new insights on developing effective ICTs for TNBC treatment in the clinics.”

“To endow the PD-L1 and CTLA-4-targeting aptamers with self-assembly capacity and TME responsiveness...”

Changes in the introduction section:

“Herein, we presented a TME-activatable aptamer-based nanoassembly for the efficient ICT treatment of TNBC, which was realized through tumor cell-selective glycolysis inhibition combined with bispecific immune checkpoint blockade. Based on that clinical evidence that TNBC cells are frequently overexpressed with glucose transporter 1 (GLUT1) for glucose import unlike other major tumor-residing immune cell populations, we selected an experimental GLUT1 inhibitor BAY-876 for selective glycolysis inhibition of TNBC cells. Meanwhile, poly β -amino ester (PAE) was conjugated onto the 3' terminal of PD-L1 and CTLA-4-antagonizing aptamers as a multifunctional chemical handle (aptPD-L1 and aptCTLA-4). Due to the significant hydrophobicity of PAE under neutral pH, aptPD-L1 and aptCTLA-4 readily self-assembled into supramolecular nanostructures in biomimetic buffer solution, which not only stabilized the aptamers but also enabled the effective loading of hydrophobic BAY-876 molecules. After reaching the acidic TNBC microenvironment, the PAE handles rapidly became protonated and switched a hydrophilic state, thus disrupting the hydrophobic/hydrophilic balance of the aptamer-PAE conjugates and triggering nanoassembly disintegration, eventually leading to the on-demand release of BAY-876, aptPD-L1 and aptCTLA-4 into TME (Figure 1a). BAY-876-mediated GLUT1-inhibition in TNBC cells readily suppressed their glycolysis activities and further downregulated the glycosylation level of PD-L1 through Glu-F6P-UDP-GlcNAc axis, thus facilitating their recognition and binding by aptPD-L1 for effective PD-L1 blockade. Meanwhile, BAY-876 treatment also abolished the overcompetition of glucose by

TNBCs and enhanced glucose abundance in TME to metabolically reprogram immunosuppressive Tregs into an immunostimulatory state, which synergized with aptCTLA-4-mediated CTLA-4 blockade to substantially alleviate Treg-mediated CTL suppression (Figure 1b). In vivo evaluations showed that the bispecific aptamer nanoassemblies abolished TNBC growth through evoking robust CTL-mediated antitumor immune responses as well as prevented TNBC relapse and metastasis by establishing potent systemic anti-TNBC immune memory, which may provide new insights on developing effective ICTs for TNBC treatment in the clinics.”

3. The authors use different pHs to demonstrate that their construct is dissociated. How are these pHs created and at what timepoint is the pH altered. Does this affect the 4T1 /immune compartment on its own?

A: Thank you for your concern. We are sorry for the lack of detailed descriptions regarding the setup of pH gradient for in vitro experiment and expanded the experimental protocols accordingly. Specifically, we would like to note that the therapeutic activities of the nanoassemblies were comparatively measured under the neutral pH of 7.4 and slight acidic pH of 6.8. Notably, pristine DMEM culture media have a neutral pH of 7.4 and are thus used for the pH 7.4 group. Meanwhile, the pH 6.8 group was established by adding lactic acid into DMEM media. The pH-altered media were used for the whole experiment period. The rationale of choosing lactic acid for adjusting the pH of culture media to slight acidic conditions is that lactic acid is the metabolic by-product of glycolytic tumor cells and well-recognized as the primary bioacid for orchestrating the acidic TME in various solid [Cell Rep 39, 110792 (2022)]. Indeed, lactic acid-supplemented culture media have been widely used in biomedical research to recapitulate the acidic TME under clinical conditions [Nat Metab 5, 314-330 (2023); J Vis Exp, 56660 (2017)]. Meanwhile, there is abundant evidence that lactic acid is capable of driving tumor progression and immunosuppression [Nat Cancer 3, 1464-1483 (2022); Cell 171, 358-371 e359 (2017); Cell Rep 38, 110451 (2022)]. This is consistent with our observations that 4T1 cells incubated under pH 6.8 showed faster proliferation rate than that under pH 7.4, while the antitumor activity of activated T cells under pH 6.8 was lower than that under pH 7.4 (Figure 4). Notably, the DNA-PAE@BAY-876 nanoassembly could effectively overcome the acidity-induced pro-tumorigenic effects and evoke potent antitumor immunity to

abolish tumor growth both in vitro and in vivo, substantiating its therapeutic utility in a clinical context.

The related experimental procedure is shown here below for your review:

“Preparation of pH6.8 medium: Lactic acid solution (20 mM) was added into high-glucose DMEM medium (pH7.4) until the pH value dropped to 6.8, during which the medium pH was monitored using a pH detector.”

4. The conditions in each experiment are not always clear to me and need to be clearly stated in either the results section or figure legend. Example include Figure 3 a, Figure 4 b.

A: Thank you for the correction. Based on your suggestion we have revamped the descriptions in the figure legends to elucidate the experimental conditions for relevant data. Some of the optimized descriptions are shown here below for your review:

Figure 3. Evaluation on the TNBC-targeting effect of DNA-PAE@BAY-876 in vitro. (a) Flow cytometric analysis on the cell binding behavior of aptPD-L1 to 4T1/HC11 cells under pH 7.4. (b) Cell binding behavior of aptCTLA-4 to activated/inactivated T cells under pH 7.4. (c) Interaction between 4T1 or HC11 cells and DNA-PAE nanoassemblies under different environmental pH conditions at 37°C for 2 h. (d) DNA-PAE@BAY-876-mediated GLUT1 inhibition in 4T1 cells and HC11 cells at 37°C for 24 h. (e-f) Dose-dependent toxicity of DNA-PAE@BAY-876 to 4T1 cells and HC11 cells at 37°C for 24 h (n=3). (g-i) Toxicity of BAY-876 to major immune cell populations at different pH conditions (n=3). (I) pH7.4+activation, (II) pH7.4+inactivation, (III) pH6.8+activation, (IV) pH6.8+inactivation. (j-k) Immunofluorescence analysis of GLUT1 and GLUT3 in 4T1 tumors. (l-m) Glucose uptake of 4T1 cells and T cells in the co-incubation system after treatment with BAY-876 or KL-11743 (n=3).

Figure 4. DNA-PAE@BAY-876 treatment mounted potent antitumor effect in 4T1-splenic immune cell co-incubation system. (a) Apoptosis ratio of 4T1 cells in the co-incubation system after different treatment at 37°C for 24 h. (b-c) Apoptosis effect of 4T1 cells with pH7.4 (b) and

pH6.8 (c) by western blotting at 37°C for 24 h. (I) NC, (II) PAE, (III) aptCTLA-4, (IV) aptPD-L1, (V) BAY-876, (VI) DNA-PAE, (VII) DNA-PAE@BAY-876. Note: the molecular weight of Bcl-2 is 26 KDa; the molecular weight of Bax is 21 KDa; the molecular weight of Tubulin is 50 KDa. (d-e) Glucose uptake by 4T1 cells or T cells after different treatments under co-culture condition at 37°C for 24 h (n=3). (f-g) Flow cytometric analysis on the expansion of CD4+/CD8+ T cells, IFN- γ +CD8+ T cells in the co-incubation system after different treatment at 37°C for 24 h. (h) Statistical analysis of data shown in panel f-g by flow cytometry (n=3). (1) NC, (2) PAE, (3) aptCTLA-4, (4) aptPD-L1, (5) BAY-876, (6) DNA-PAE, (7) DNA-PAE@BAY-876.

Figure 5. Nanoassembly-enhanced 4T1 cell recognition and binding by aptPD-L1. (a) Schematic illustration on the nanoassembly-mediated metabolic rewiring of 4T1 cells for enhanced ICT. (b) Evaluation on the dose-dependent impact of BAY-876 on PD-L1 glycosylation in 4T1 cells with pH7.4/6.8 at 37°C for 24 h. The concentration of IFN- γ was 1 $\mu\text{g}\cdot\text{mL}^{-1}$. Note: the molecular weight of PD-L1 is 33-70 KDa; the molecular weight of Tubulin is 50 KDa. (c) Impact of DNA-PAE@BAY-876 on the PD-L1 glycosylation level in multiple cell types at 37°C for 24 h. The concentration of DNA-PAE@BAY-876 was 100 nM and the concentration of IFN- γ was 1 $\mu\text{g}\cdot\text{mL}^{-1}$. (d) Impact of UDP-GlcNAc and F6P on the PD-L1 glycosylation levels in 4T1 cells after BAY-876 treatment with high/low-glucose media. The concentration of BAY-876, UDP-GlcNAc and IFN- γ was 100 nM, 0.1 mM and 1 $\mu\text{g}\cdot\text{mL}^{-1}$, respectively. (e) UDP-GlcNAc abundance in 4T1 cells under different conditions (n=4). (I, V) Control; (II, VI) BAY-876; (III, VII) BAY-876+F6P; (IV, VIII) BAY-876+UDP-GlcNAc. (f) Schematic illustration of Hexosamine Biosynthesis Pathway (HBP). (g) 3D plot on the correlation between UDP-GlcNAc and F6P with 2-NBDG after treatment by BAY-876 under graded concentrations. (h-k) Impact of BAY-876 on the competitive combination between aptPD-L1/PD1 with PD-L1 on 4T1 cell surface.

Figure 6. Nanoassembly-mediated abolishment of Treg-mediated immunosuppression. (a) Schematic illustration of the nanoassembly-mediated reprogramming of immunosuppressive Tregs. (b) Foxp3 expression in Tregs after different treatments at 37°C for 24 h. (c) CTLA-4 and CD25 expression on Tregs after different treatments at 37°C for 24 h. (d-e) Quantitative profiling of Treg-mediated CTL suppression after different treatments with pH7.4/6.8 (n=3) via measuring IFN- γ levels. Tregs suppression (%) = (IFN- γ CD8-IFN- γ Treg+CD8) / IFN- γ CD8 (f-g) ELISA assay on the secretion levels of key immune-related cytokines by immune cells treated with DNA-

PAE@BAY-876 under different co-culture conditions at 37°C for 24 h (n=3). (h-j) Viability of 4T1 cells under different co-culture conditions at 37°C for 24 h (n=3). (k) Statistical analysis of data shown in panel b-c by flow cytometry (n=3). (1) Control, (2) aptCTLA-4, (3) BAY-876, (4) B+C, (5) DNA-PAE, (6) DNA-PAE@BAY-876. B+C indicates the mixture of BAY-876 and aptCTLA-4.

5. The authors activate CD4/CD8 T cells for 48 h and conclude that BAY does not affect viability/function. This does not agree with previous findings (PMID: 24930970), and it is important to know if longer activation times to generate Teff (e.g. 96h) affect this.

A: Thank you for your insight. We agree with the reviewer that BAY-876 could indeed impose negative impact on the viability and immune function of activated CD4+ or CD8+ T cells by inhibiting GLUT1-mediated glucose uptake according to previous studies. However, we would like to note that the adverse effects of BAY-876 on immune cells are highly dependent on its dosing conditions. Specifically, we incubated various major immune cell subtypes with BAY-876 under graded concentrations and found no significant impact on the viability and function of activated T cells when the BAY-876 concentration was below 100 nM after 24 h of incubation (Figure S14a). Considering that the equivalent BAY-876 concentration in the DNA-PAE@BAY-876 nanoassembly for in vitro and in vivo experiment was around 100 nM, it could be concluded that the DNA-PAE@BAY-876 nanoassembly would not impair the viability and functions of activated CD4+ and CD8+ T cells under the given dose conditions while not contradicting the findings in previous reports.

On the other hand, based on the reviewer's suggestion we have incubated the CD4+ and CD8+ T cells with anti-CD3 antibody and anti-CD28 antibody for 96 h and then subjected the activated T cells to 24 h of BAY-876 treatment under different concentrations to test their BAY-876 susceptibility. The data revealed a similar trend that BAY-876 did not show obvious toxicity to activated CD4+/CD8+ T cells under the BAY-876 concentration of 100 nM (Figure S14b), again supporting the immunostimulatory effect of the DNA-PAE@BAY-876 nanoassembly treatment in the present study.

The related data and discussions are shown here below for your review:

“MTT assay showed no significant changes in the viability of all major immune cell populations after BAY-876 treatment regardless of their activation status (Figure 3g-i and Figure S14).”

“**1.13 T cell activation:** The concentrations of anti-CD3 antibody and anti-CD28 antibody were respectively diluted to $5.5 \mu\text{g}\cdot\text{mL}^{-1}$ and $2 \mu\text{g}\cdot\text{mL}^{-1}$ by PBS. The above antibody diluent was added to 96-well plate at $70 \mu\text{L}$ per well and incubated at 4°C overnight. After incubation, the liquid was drained and the plate was cleaned twice with PBS. The sorted T cells were added and cultured in an 37°C incubator for 48 h or 96 h.”

“**Immunocytotoxicity assay:** Various immune cell populations were inoculated into 96-well plate at 1.0×10^5 units per well, and BAY-876 solutions with different concentrations and different pH values were added into the above plate for 24 h. 10% CCK-8 solution was then added and incubated at 37°C for 2 h. The absorbance was measured at 450 nm using a microplate reader.”

Figure S14. Concentration-dependent impact of BAY-876 to major immune cell populations under the activation period of 48 h (a) or 96 h (b) (n=3).

6. Figure 5a is not cited or discussed.

A: Thank you for your reminder. We are sorry for the careless mistake and have added the relevant discussion of Figure 5a in the original manuscript, which is also shown here below for your review:

“The observations above immediately suggested that the DNA-PAE@BAY-876 nanoassembly could enhance the PD-L1 antagonization performance of co-delivered aptPD-L1 against 4T1 cells in a highly coordinated manner, thus ameliorating the tumor cell-induced immunosuppression for more effective immunotherapy (Figure 5a).”

7. What is the difference experimentally between Figure S10A and Figure S17A? Figure S10A suggested to me that BAY hardly affects 2- NBDG uptake (MFI 5000->4500), while Figure 17A suggests a striking reduction in 2-NBDG uptake (95,000->50,000). Are more experiments needed to resolve this?

A: Thank you for your concern. We have carefully examined the experimental procedures and concluded that the difference was caused by the different 2-NBDG dosing conditions. Specifically, the experiment in Figure S10A was carried out using the co-incubation system of 4T1 and splenic cells under the 2-NBDG concentration of 15 mM, while the experiment in Figure S17A was carried out using only 4T1 cells under the 2-NBDG concentration of 25 mM, which was almost doubled. Therefore, the flow cytometric analysis on the 2-NBDG uptake by 4T1 cells in Figure S10A was more susceptible to the interference of background fluorescence. To avoid misunderstanding, we have redone both experiments using identical cell amount and 2-NBDG concentration, and the results confirmed that BAY-876 could indeed substantially inhibit 2-NBDG uptake by 4T1 cells. Specifically, the 2-NBDG uptake by 4T1 cells in the 4T1/splenocyte co-incubation system has decreased by 57.09% after incubation with BAY-876, which was at a comparable level to the reduction in 2-NBDG uptake in BAY-876-treated 4T1 cells. The new data have been incorporated into the manuscript and the relevant discussions have been optimized accordingly.

The related data and discussions are shown here below for your review:

“Notably, treating the co-culture system with 100 nM BAY-876 caused significant suppression of

4T1 cells while expanding the immune cell populations associated with the adaptive antitumor immunity including DCs, M1 macrophages, CD4+ T cells and CD8+ T cells, on account of the capacity of BAY-876 to selectively inhibit GLUT1 to block tumor cell-intrinsic glucose uptake. Contrastingly, both tumor cells and immune cells were substantially suppressed by KL-11743 treatment due to the simultaneous inhibition of GLUT1 and GLUT3, leading to universal glucose uptake blockade (Figure 31-m and Figure S16-S18). The evidence above supported the therapeutic capacity of the BAY-876-containing nanoassembly to rebalance glucose competition between TNBC cells and immune cells for mounting adaptive antitumor immune responses.”

“**Glucose uptake analysis of 4T1 cells and T cells:** 4T1 cells were inoculated in 12-well plate with 1.5×10^5 cells per well, cultured for 24 h, added with different samples and T cells of 30 times the amount of tumor cells. 30 mM 2-NBDG was subsequently added and incubated at 37°C for 24 h. After centrifuging at 1500 rpm for 5 min, all cells were added with APC-anti-CD45 antibody, incubated at room temperature for 30 min in the dark, then detected by flow cytometry (CytoFLEX, Beckman Coulter).

4T1 cells were inoculated in 12-well plate with 1.5×10^5 cells per well, then added with different concentrations of BAY-876 after cultured for 24 h. 30 mM 2-NBDG was subsequently added and incubated at 37°C for 24 h. Finally, 4T1 cells were collected and detected by flow cytometry (CytoFLEX, Beckman Coulter).”

Figure S16. Glucose uptake rate in cells after BAY-876 or KL-11743 treatment. (a-b) Impact of different GLUT inhibitors on 2-NBDG uptake in 4T1 cells or T cells in the co-culture system (n=3). (c) Extracellular glucose levels in 4T1 cell incubation media after treatment by BAY-876 or KL-11743 to 4T1 cells alone by 2-NBDG.

Figure S25. Changes in the abundance of key glycolysis products after treatment with BAY-876 at different concentrations. (a) UDP-GlcNAc (n=4); (b) F6P (n=4); (c) 2-NBDG (n=3).

8. The UDP- GlnNac rescue in Figure 5 is not convincing- perhaps it is not uptaken well?

A: Thank you for your suggestion. Indeed, we have repeated the rescue experiment using UDP-GlcNAc several times and confirmed that UDP-GlcNAc treatment induced substantial yet partial recovery of PD-L1 glycosylation. We have then carried out rescue experiment in BAY-876-treated 4T1 cells using UDP-GlcNAc under graded concentrations and found that increasing the UDP-GlcNAc dose could improve the glycosylation levels of membrane PD-L1. This is understood that we initially used low levels of UDP-GlcNAc for the recue experiment (0.1 mM) and adding extra exogenous UDP-GlcNAc could facilitate its uptake by tumor cells and enhance recovery of PD-L1 glycosylation levels. However, the recovery was still not complete even under a high UDP-GlcNAc concentration of around 0.5 mM (Figure S24a). From a biochemical perspective, protein glycosylation requires the support of key metabolites such as ATP and NADPH [Cell 143, 711-724 (2010); Cell Host Microbe 12, 47-59 (2012)], both which are critically dependent on the glycolysis activity in tumor cells. However, the BAY-876-mediated blockade of glucose uptake substantially inhibited the glycolysis activity in tumor cells, which not only suppressed the generation of UDP-GlcNAc but also reduced the production of ATP and NADPH, thus substantially reduced the capacity of tumor cells to execute glycosylation of membrane PD-L1 despite the supplementation of exogenous UDP-GlcNAc (Figure S24b-c). **The associated results and discussions have been incorporated into the manuscript and also shown here for your review:**

“Interestingly, western blotting and ELISA assay results collectively demonstrated that the BAY-876-induced PD-L1 deglycosylation could be reversed by adding additional F6P or UDP-GlcNAc,

evidently confirming that BAY-876 reduced PD-L1 glycosylation levels in 4T1 cells through negatively regulating the Glu-F6P-UDP-GlcNAc axis (Figure 5d-e and Figure S24). However, it is also noteworthy that even a large UDP-GlcNAc dose at around 0.5 mM could only induce partial recovery of PD-L1 glycosylation levels. This is understood that protein glycosylation in tumor cells is largely sustained by the glycolysis activities both in terms of UDP-GlcNAc precursors and metabolite supply including ATP and NADPH, both of which are significantly suppressed by the BAY-876-mediated blockade of glucose uptake.”

“1.20 ATP and NADPH abundance in 4T1 cells: 4T1 cells were inoculated into 6-well plates at 5×10^5 units per well. After culturing for 24 h, the cells were treated by BAY-876 with different concentration for 24 h. After cleaning with PBS, the cells were lysed with RIPA lysate (containing 1% PMSF) for 30 min at 4°C and centrifuged at 12000 rpm for 10 min. Eventually, the collected supernatant was assayed by the ATP or NADPH kit to determine the cellular ATP and NADPH abundance.”

“1.23 UDP-GlcNAc-mediated recovery of PD-L1 glycosylation in BAY-876-treated 4T1 cells: 4T1 cells were inoculated into 6-well plates at 5×10^5 units per well. After culturing for 24 h, the cells were treated by 100 nM BAY-876 for 12 h, then continually incubated with UDP-GlcNAc or F6P for 12 h. Finally, the PD-L1 glycosylation level was observed by Western Blotting and the cellular UDP-GlcNAc level was detected by the ELISA kit.”

Figure S24. Effects of BAY-876 or UDP-GlcNAc with different concentrations to 4T1 cells. (a) Effect of UDP-GlcNAc with different concentrations to the PD-L1 glycosylation on 4T1 cells treated with 100 nM BAY-876. Note: the molecular weight of PD-L1 is 33-70 KDa; the molecular weight of Tubulin is 50 KDa. (b) Changes of ATP content in 4T1 cells by BAY-876 with different concentrations (n=3). (c) Changes of NADPH content in 4T1 cells by BAY-876 with different concentrations (n=3).

9. I cannot find the proliferation data mentioned on line 338.

A: Thank you for your reminder. We are sorry for the careless mistake regarding the description as it was actually a typo that we missed during manuscript preparation. The wording has been optimized to avoid misunderstandings, **of which the corrected version is shown here below for your review.**

“The nanoassembly-induced reduction in Treg-mediated immunosuppression was further studied by monitoring the effector function of CD8+ T cells.”

10. In figure 6h-j it seems the majority of the 4T1 killing is mediated by BAY, with a minor effect of the CD8 T cells. Is this because the CD8 T cells are inhibited? How functional are these cells?

A: Thank you for your concern. We would like to first explain that the 4T1 cell viability cannot be numerically compared between different panels due to the difference in the set-up for individual control groups. Specifically, for the experiment in panel 6h, 4T1 cells were co-incubated with CD8+ T cells in all groups, while the 4T1 cells in panel i and j were incubated with Tregs and CD8+ T cells + Tregs, respectively. Consequently, 4T1 cells in the control groups of individual panels have distinctively different viability that makes the cross-panel comparison of the viability data irrelevant. Particularly, due to the inclusion of immunosuppressive Tregs in the co-incubation system in panel 6i and 6j, the viability of 4T1 cells in the control groups of these two panels were intrinsically higher than that in panel 6h. Therefore, it is logical to analyze the immunotherapeutic activity of BAY-876-containing nanoassemblies within the same panel. Regarding this issue, we would like to explain that on one hand, BAY-876 effectively inhibited the glycolysis in GLUT-1-overexpressing 4T1 cells and caused tumor starvation, leading to substantial direct damage to TNBCs. On the other hand, BAY-876-induced blockade of tumor-intrinsic glucose uptake would enhance the glucose

abundance in TME and cause metabolic instability of tumor-residing Tregs to impair their immunosuppressive function, which could synergize with the synchronically delivered anti-PD-L1 and anti-CTLA4 aptamers to evoke robust antitumor immunity. Indeed, in vitro evaluations showed that BAY-876 alone could elicit potent antitumor effect as the tumor survival rate in the DNA-PAE@BAY-876 was as low as 52.52% according to Figure S19e analysis. On the other hand, the mounted antitumor immunity caused a further increase in the tumor inhibition efficacy, of which the 4T1 cell survival rate dropped below 23.84% in Figure S20a, evidently supporting the potent antitumor effect of the nanoassembly-evoked adaptive immune reactions and validating its contribution to the eventual antitumor efficacy.

To further demonstrate the robustness of the nanoassembly-evoked immunotherapeutic effects, we have monitored the phenotypical changes in the activated T cells after nanoassembly treatment as suggested and observed that the DNA-PAE@BAY-876 treatment promoted the secretion of IFN- γ while reducing IL-10 secretion (Figure 6f-g). In addition, the therapeutic potency of the treatment-evoked antitumor immunity was also supported by the results of the bilateral tumor models, in which the DNA-PAE@BAY-876 treatment substantially suppressed the growth of secondary tumors and improved mouse survival. Overall, these results collectively demonstrated that DNA-PAE@BAY-876 nanoassembly did not induce obvious suppression effects on effector T cells and could effectively inhibit TNBC growth through the cooperation of tumor-specific glucose uptake inhibition and aptamer-enabled immune checkpoint blockade therapy.

Figure 6. Nanoassembly-mediated abolishment of Treg-mediated immunosuppression. (a) Schematic illustration of the nanoassembly-mediated reprogramming of immunosuppressive Tregs. (b) Foxp3 expression in Tregs after different treatments at 37°C for 24 h. (c) CTLA-4 and CD25 expression on Tregs after different treatments at 37°C for 24 h. (d-e) Quantitative profiling of Treg-mediated CTL suppression after different treatments with pH7.4/6.8 (n=3) via measuring IFN-γ levels. Tregs suppression (%) = (IFN-γCD8-IFN-γTreg+CD8) / IFN-γCD8 (f-g) ELISA assay on the secretion levels of key immune-related cytokines by immune cells treated with DNA-PAE@BAY-876 under different co-culture conditions at 37°C for 24 h (n=3). (h-j) Viability of 4T1

cells under different co-culture conditions at 37°C for 24 h (n=3). (k) Statistical analysis of data shown in panel b-c by flow cytometry (n=3). (1) Control, (2) aptCTLA-4, (3) BAY-876, (4) B+C, (5) DNA-PAE, (6) DNA-PAE@BAY-876. B+C indicates the mixture of BAY-876 and aptCTLA-4.

Figure S19. Biochemical changes and survival of 4T1 cells after different treatment. (a-d) Statistic analysis on the expression levels of key apoptosis-related markers in 4T1 cells after different

treatment via WB assay (n=3). (e-f) Survival rate of 4T1 cells and HC11 cells after different treatment (n=3).

Figure S20. Characterization of the 4T1 cell survival after different treatment. (a) 4T1 survival rate in the 4T1-splenic cell co-incubation system after different treatment (n=3). (b) Migration of T cells in the 4T1-splenic cell co-incubation system after various treatment (n=3). (c) Invasion capacity of 4T1 cells in the 4T1-splenic cell co-incubation system after different treatment.

11. In figure 7 it would be helpful to match the colours used across panels.

A: Thank you for your suggestion. Based on your advice we have unified the choices of colors for individual groups in Figure 7 and Figure 9, and the corrected figures are shown here below.

Figure 7. Antitumor evaluation of DNA-PAE@BAY-876 in vivo. (a) Schematic diagram of treatment schedule of DNA-PAE@BAY-876 nanoassemblies on 4T1 tumor-bearing mice. (b) Visual comparison of tumors from Balb/c mice after different treatment. (c) In vivo bioluminescence images on the tumor progression in Balb/c mice in different groups. (d) Tumor volume changes after different treatment (n=5). (e) Final tumor weight in different groups at the end of the 21 days period (n=5). (I) NC, (II) PAE, (III) aptCTLA-4, (IV) aptPD-L1, (V) BAY-876, (VI)

DNA-PAE, (VII) DNA-PAE@BAY-876. (f) Body weight changes of Balb/c mice in different groups during treatment (n=5). (g-h) TUNEL and H&E staining of tumor samples. (i-k) ELISA assay of glucose, lactic acid and UDP-GlcNAc abundance in tumor samples (n=3).

Figure 9. Therapeutic evaluation of DNA-PAE@BAY-876 on bilateral tumor models. (a) Schematic diagram of 4T1 bilateral tumor model construction and the treatment schedule. (b) Photographical images of the bilateral tumors through the treatment period. (I) NC, (II) PAE, (III) aptCTLA-4, (IV) aptPD-L1, (V) BAY-876, (VI) DNA-PAE, (VII) DNA-PAE@BAY-876. (c) Volume changes of the distal tumors after different treatment (n=3). (d) Survival curves of 4T1 bilateral tumor-bearing mice after different treatment (n=5). (e) Flow cytometry evaluation on the tumor-infiltration of total immune cells (CD45+). (f) Flow cytometry detection on the infiltration status of IFN- γ +CD8+T cells in distal tumor. (g-h) Flow cytometry evaluation on the tumor-infiltration of M1 macrophages (F4/80+CD86+) and CD8+ effector memory T cells (CD44+CD62L-). (i) Statistical analysis of flow cytometry data in panel e-h (n=3). (1) NC, (2) PAE, (3) aptCTLA-4, (4) aptPD-L1, (5) BAY-876, (6) DNA-PAE, (7) DNA-PAE@BAY-876.

12. It is not clear to me how much the nanoassembly is important here. Would combined PD-L1/CTLA4 antibodies plus BAY have a similar affect in vivo? The authors only compare only BAY, or only aptamer to the combination.

A: Thank you for the suggestion. We would like to apologize for the lack of pertinent control groups for the in vivo therapeutic evaluations and carried out additional in vivo experiment on 4T1 tumor-bearing mouse models by incorporating the suggested control groups including anti-PD-L1 antibody and anti-CTLA-4 antibody. We found that the therapeutic effect of aptamer group (aptPD-L1+aptCTLA-4) on 4T1 tumor-bearing mice was inferior than antibody group (abPD-L1+abCTLA-4), which could be readily explained by the low bioavailability of aptamers in vivo. In contrast, DNA-PAE@BAY-876 nanoassembly showed significantly superior therapeutic effect than antibodies+BAY-876 (Figure S41a-c). The difference in the therapeutic performance of individual groups could be explained by the pharmacokinetic behavior of individual components. According to the result of fluorescence imaging in vivo, it was found that most of anti-PD-L1 antibody reached the tumor site at 6 h, while BAY-876 mainly enriched in the kidney site at 6 h (Figure S41d), thus incapable of realizing the synchronization of BAY-876 and immune checkpoint inhibiting antibodies. In contrast, the DNA-PAE@BAY-876 nanoassemblies could efficiently become deposited in tumor tissues after 6 h (Figure S30), attributing to its tumor-targeting delivery efficacy in vivo. The superior antitumor efficacy of the DNA-PAE@BAY-876 nanoassembly compared with simple

combination of anti-PD-L1/anti-CTLA4 antibodies and BAY-876 was collectively supported by the analysis results of mouse survival rates, tumor volume and tumor weight (Figure S41e-g) as well as H&E and TUNEL staining results (Figure S41h-i).

Furthermore, we have also comparatively analyzed the immunostimulatory effect of DNA-PAE@BAY-876 nanoassembly and antibody+BAY-876 by flow cytometry. Compared with antibodies+BAY-876, DNA-PAE@BAY-876 nanoassembly could more effectively promote the infiltration and expansion of CD8⁺ T cells, CD4⁺T cells, dendritic cells and M1 macrophages in the tumor tissues (Figure S42a-c) while inhibiting M2 macrophages, Tregs and MDSCs (Figure S42d-f). Overall, the results above demonstrated that the nanoassembly is crucial for enabling the functional cooperation between BAY-876 and the PD-L1/CTLA-4 aptamers, which not only potentiated the nanointegrative incorporation of the therapeutic components but also allowed their targeted and synchronized action in tumor tissues for enhanced ICT.

The related data and discussions are shown here below for your review:

“7. Therapeutic comparison DNA-PAE@BAY-876 to immune checkpoint inhibiting antibodies in vivo

The antitumor efficacy of DNA-PAE@BAY-876 nanoassembly was further compared to commonly investigated immune checkpoint inhibiting antibodies including anti-PD-L1 antibody /anti-CTLA-4 antibody in vivo for elucidating their clinical potential. Notably, the antitumor efficacy of aptPD-L1+aptCTLA-4 combination on 4T1 tumor-bearing mice was inferior to the abPD-L1+abCTLA-4 combination. In contrast, DNA-PAE@BAY-876 nanoassembly showed much superior therapeutic effect than antibody combination+BAY-876 treatment (Figure S41a-c). To elucidate the difference in the therapeutic performance of individual groups, we carried out comprehensive analysis on the pharmacokinetic behavior of individual components and found that most of anti-PD-L1 antibody reached the tumor site at 6 h, while BAY-876 were mainly enriched in the kidney site at 6 h (Figure S41d), as BAY-876 lacks intrinsic tumor targeting capacity and therefore failed to synchronize the antitumor actions of BAY-876 and immune checkpoint inhibiting antibodies. Indeed, the DNA-PAE@BAY-876 group showed superior tumor inhibition efficacy than the antibody-based treatments in terms of the mouse survival rate, tumor volume, and tumor weight (Figure S41e-g), accompanied with more pronounced tumor cell death according to H&E

and TUNEL staining results (Figure S41h-i). The trends above were further supported by the evaluations on the immunostimulatory capacity of DNA-PAE@BAY-876 nanoassembly compared with antibody combination+BAY-876 by flow cytometry. Remarkably, compared with immune checkpoint inhibiting antibodies, DNA-PAE@BAY-876 nanoassembly more effectively promoted the infiltration and proliferation of CD8⁺ T cells, CD4⁺T cells, DCs and M1 macrophages at the tumor site (Figure S42a-c), while potently inhibiting M2 macrophages, Tregs and MDSCs (Figure S42d-f), evidently supporting its potential utility for enhanced ICT against TNBCs in the clinics.”

“Therapeutic comparison between DNA-PAE@BAY-876 nanoassembly and immune checkpoint inhibiting antibodies in vivo: Firstly, 100 μL 2×10^6 4T1 cells were injected subcutaneously into each Balb/c mouse, when the tumor grew to about 100 mm^3 , the mice were randomly divided into 5 groups at 6 mice per group and subjected to different treatment (Control, 50 $\mu\text{g} \cdot \text{mL}^{-1} \cdot \text{g}^{-1}$ anti-PD-L1 antibody and 50 $\mu\text{g} \cdot \text{mL}^{-1} \cdot \text{g}^{-1}$ anti-CTLA-4 antibody, 12.35 $\mu\text{g} \cdot \text{mL}^{-1} \cdot \text{g}^{-1}$ aptCTLA-4 and 12.22 $\mu\text{g} \cdot \text{mL}^{-1} \cdot \text{g}^{-1}$ aptPD-L1, 50 $\mu\text{g} \cdot \text{mL}^{-1} \cdot \text{g}^{-1}$ antibodies and 0.05 $\mu\text{g} \cdot \text{mL}^{-1} \cdot \text{g}^{-1}$ BAY-876, 48.87 $\mu\text{g} \cdot \text{mL}^{-1} \cdot \text{g}^{-1}$ DNA-PAE@BAY-876) through intravenous injection once every two days. After 21 days of treatment, all mice were sacrificed, their tumors were collected for analysis. The murine survival curves were plotted across a period of 60 days.

A portion of the collected tumor tissue was washed twice with PBS, placed on a 0.4 μm filter membrane, added with high-glucose DMEM medium, carefully pulverized with syringe head, mixed with 5 mL red blood cell lysis buffer, stood for 10 min, and centrifuged at 2000 rpm for 5 min. The supernatant was removed, the cells were re-suspended with PBS and then stained with fluorescent antibodies for 30 min to detect the infiltration of various immune cells in the tumor by flow cytometry (CytoFLEX, Beckman Coulter).

Another portion of the tumor tissue was frozen-sectioned, fixed with 4% paraformaldehyde at room temperature for 30 min. After permeation with 0.5% Triton X-100 for 10 min, the sections were incubated with TUNEL test solution at 37°C for 60 min, then added with DAPI for 10 min in the dark and sealed with glycerol anhydrous after washed with PBS. The mounted tissue sections were finally imaged by a laser confocal microscope (Leica TCS SP8, Germany).”

Figure S41. Comparison of DNA-PAE@BAY-876 to anti-PD-L1/anti-CTLA-4 antibody combination in 4T1 tumor-bearing mice. (a) Schematic diagram of the treatment schedule in vivo. (b) Photographical images of the 4T1 tumors through the treatment period. (I) Control, (II) ab, (III)

apt, (IV) ab+BAY-876, (V) DNA-PAE@BAY-876. (c) Visual comparison of tumors from Balb/c mice after different treatment. (d) Systemic distribution of BAY-876/abPD-L1 at 0/6/12/24h post intravenous injection. (e) Survival curves of 4T1 tumor-bearing mice after different treatment (n=5). (f-g) Final tumor volume and weight in different groups at the end of the 21 days period (n=5). (h-i) H&E and TUNEL staining of tumor samples. ab was anti-PD-L1 antibody and anti-CTLA-4 antibody; apt was aptPD-L1 and aptCTLA-4.

Figure S42. Comparative analysis on the antitumor efficacy of DNA-PAE@BAY-876 and anti-PD-L1/anti-CTLA-4 antibody in 4T1 tumor-bearing mice. (a) Flow cytometry detection on the tumor infiltration of CD8+ T cells and CD4+ T cells. (b) Flow cytometry detection on the tumor infiltration of mature DCs (CD11c+MHC-II+). (c) Flow cytometry detection on the tumor-infiltration of M1

macrophages (F4/80+CD86+). (d) Flow cytometry detection on the tumor infiltration of M2 macrophages (F4/80+CD206+). (e) Flow cytometry detection on the tumor infiltration of Tregs (CD4+CD25+Foxp3+). (f) Flow cytometry detection on the tumor infiltration of MDSCs (CD11b+GR1+). (1) Control, (2) ab, (3) apt, (4) ab+BAY-876, (5) DNA-PAE@BAY-876. ab was anti-PD-L1 antibody and anti-CTLA-4 antibody; apt was aptPD-L1 and aptCTLA-4.

Reviewer #3 (Remarks to the Author): with expertise in nano-technology for cancer therapy

Immune checkpoint inhibition therapy (ICT) is a novel therapeutic option for triple negative breast cancer (TNBC) that shows promising efficacy in clinical practice. However, post-transcriptional glycosylation of PD-L1 on TNBC cells may significantly impede the immune checkpoint blockade efficacy. In view of the central role of TNBC-intrinsic glycolysis in driving ICT resistance, this manuscript by Ren et al described a nanointegrated strategy for enhancing the ICT efficacy against TNBC with high PD-L1 glycosylation level, which is realized through the combined TNBC-targeted glycolysis inhibition and bispecific PD-L1/CTLA4 blockade using bioresponsive aptamer-based nanoassemblies. The on-demand activation of the nanoassembly in TME could release GLUT1-inhibiting BAY-876 drug and selectively abolish TNBC glycolysis, thus reshaping the immunosuppressive TME to facilitate the bispecific ICT. This study is interesting and of significance, which offers a novel multidisciplinary approach to address clinical problems. I recommend it for Nat Comm after addressing the following issues.

A: Thank you for the appreciation of our work as well as the constructive suggestions. Based on your advice we have thoroughly revised our manuscript according to your comments, and the detailed responses are shown here below for your review.

Major issues:

1. Drug release from the nanoassembly was triggered through PAE protonation in acidic environment. The charge status of PAE under pH 7.4 should also be investigated to validate the drug release mechanism.

A: Thank you for your advice. Since PAE itself is highly hydrophobic and its zeta potential cannot be directly measured on a zeta potential analyzer. Therefore, we measured the zeta potential changes of DNA-PAE under different pH conditions to indicate the charge status of PAE thereof. As shown in Figure S7a, zeta potential of DNA-PAE changed from negative to almost neutral when the pH value dropped from 7.4 to 6.8, showing that PAE has become protonated at pH 6.8. At the same time, the Tyndall effect of DNA-PAE-based nanoassemblies was gradually attenuated with the decreasing pH (Figure S7b), indicating that the electrostatic repulsion between the charged PAE

species destabilized the DNA-PAE nanoassemblies and may contribute to the eventual drug release performance under acidic conditions.

The related data and discussions are shown here below for your review:

“At the same time, DLS detection showed that the average hydrodynamic size of DNA-PAE increased from 122 to 328 nm (Figure 2m) when pH dropped from 7.4 to 6.8. Meanwhile, the zeta potential of DNA-PAE increased from -16.26 to -2.99 mV (Figure S7a), accompanied with diminished Tyndall effect (Figure S7b). These observations indicated that the electrostatic repulsion between the charged PAE species under acidic conditions destabilized the DNA-PAE nanoassemblies.”

Figure S7. Protonation of DNA-PAE nanoassemblies under acidic condition. (a) Zeta potential changes of DNA-PAE nanoassemblies at different pH conditions (n=3). (b) Tyndall effect of DNA-PAE nanoassemblies in PBS buffer at different pH values.

2. It is important to examine if the acidic pH would affect the bioactivity of the aptamers, including their secondary structures and binding affinity with cognate receptors.

A: Thanks for your comments. Based on your advice we have investigated the secondary structures and binding affinities of both aptPD-L1 and aptCTLA-4 with their cognate receptors as suggested. As shown in Figure S11a-b, variations in environmental pH values had no obvious effect to the targeting ability of aptPD-L1 and aptCTLA-4 in the co-culture system of 4T1 cells and spleen cells.

Specifically, the flow cytometric data showed that aptPD-L1 bond preferentially to PD-L1-overexpression 4T1 cells, while aptCTLA-4 predominantly bond to Tregs. Considering that there were many types of proteins on the cell membrane, we further incubated Cy5-aptPD-L1 and FAM-aptCTLA-4 with magnetic beads separately modified with purified mouse PD-L1 protein and mouse CTLA-4 protein, and then collected magnetic beads to detect the fluorescence intensity for evaluating the robustness of aptamer binding with their cognate receptors. As shown in the Figure S11c-d, similar trends were observed that Cy5-aptPD-L1 and FAM-aptCTLA-4 showed enhanced binding with PD-L1 and CTLA-4 modified magnetic beads, respectively, again validating that pH changes had no obvious influence to the targeting ability of aptPD-L1 and aptCTLA-4 in the present study.

The related data and discussions are shown here below for your review:

“In addition, flow cytometry and fluorescence spectroscopic data showed that the pH variations had no significant effect on the binding ability of the two aptamers (Figure S11), supporting their therapeutic robustness in the acidic environment.”

“Binding of aptPD-L1 and aptCTLA-4 in co-culture system: 4T1 cells and splenic immune cells were mixed in the 1.5 mL centrifuge tube at a ratio of 1:30 and treated with 5% BSA for 30 min, followed by the treatment of Cy5-aptPD-L1 or FAM-aptCTLA-4 for another 30 min of incubation. After washing twice with PBS, PC7-anti-CD45 antibody, PE-anti-CD4 antibody and APC-anti-CD25 antibody was added into above cells. Finally, the fluorescence intensity on 4T1 cells or immune cells was measured by flow cytometry (CytoFLEX, Beckman Coulter).”

“1.17 Binding of aptPD-L1/aptCTLA-4 to mouse PD-L1/CTLA-4 proteins via magnetic bead assay: 1 mL streptavidin-coated magnetic beads were stood for 3 min with the magnet, then washed by PBS for three times. The biotin-modified mouse PD-L1 or mouse CTLA-4 protein with a final concentration of 50 nM was mixed with the above magnetic beads, and incubated on the rotator for 60 min. After cleaning with PBS for three times, 150 nM Cy5-aptPD-L1 or FAM-aptCTLA-4 with different pH was added into the above mixture, then incubated at room temperature for 30 min. After washing twice with PBS, the fluorescence level was measured by a fluorescence spectrophotometer.”

Figure S11. Investigation of the impact of pH variations on the binding ability of aptPD-L1 and aptCTLA-4. (a-b) Cell binding behavior of aptPD-L1 and aptCTLA-4 in the co-incubation system of 4T1 cells and splenic immune cells under different pH conditions. (c-d) Evaluation on the binding of aptPD-L1/aptCTLA-4 to mouse PD-L1/CTLA-4 protein-modified magnetic beads under different pH conditions.

3. The binding of PD-L1 and CTLA-4 aptamers to 4T1 cells and Tregs was measured under separated conditions, which could not adequately illustrate the targeting ability in the complex tumor microenvironment, the authors need to investigate cell- aptamer binding under co- culture condition.

A: Thanks for your advice. Based on your advice we have established co-culture systems of 4T1 cells and splenic immune cells to observe the binding ability of Cy5-aptPD-L1 and FAM-aptCTLA-4 in TME-like conditions (Figure S9). Notably, both aptPD-L1 and apt-CTLA-4 showed preferential binding with the designated cell targets, supporting their targeting capacity in the complex TME.

The related data and discussions are shown here below for your review:

“Furthermore, we established co-incubation systems comprising 4T1 cells and splenic immune cells and confirmed that PAE-modified aptPD-L1 and aptCTLA-4 could preferentially complex

with 4T1 cells and Tregs, respectively, again validating the application potential of the PAE-modified aptamers for robust ICT against TNBCs (Figure S9).”

“**Binding of aptPD-L1 and aptCTLA-4 in co-culture system:** 4T1 cells and splenic immune cells were mixed in the 1.5 mL centrifuge tube at a ratio of 1:30 and treated with 5% BSA for 30 min, followed by the treatment of Cy5-aptPD-L1 or FAM-aptCTLA-4 for another 30 min of incubation. After washing twice with PBS, PC7-anti-CD45 antibody, PE-anti-CD4 antibody and APC-anti-CD25 antibody was added into above cells. Finally, the fluorescence intensity on 4T1 cells or immune cells was measured by flow cytometry (CytoFLEX, Beckman Coulter).”

Figure S9. Targeting ability of aptPD-L1 (a) and aptCTLA-4 (b) in the co-incubation system of 4T1 cells and splenic immune cells.

4. Based on the comparison between KL-11743 and BAY-876 in the co-culture system, the authors concluded that 100 nM BAY-876 could effectively promote the antitumor function of immune cells. However, only CD4+ T and CD8+ T cells were examined, the authors should measure other major relevant immune cell populations such as dendritic cells and M1 / M2 macrophages.

A: Thank you for your suggestion. Extending from the detection results on CD4⁺ T cells and CD8⁺ T cells, we further detected the changes in DC and macrophage populations after KL-11743 and BAY-876 treatment as suggested. As shown in Figure S18, both DC and M1 macrophage populations slightly increased after treatment with only BAY-876, indicating that even low concentrations of BAY-876 could promote immune cell function by inhibiting the glycolysis of tumor cells. In contrast, the simultaneous inhibition of GLUT1 and GLUT3 by KL-11743 resulted in a significant decrease in the frequency of DCs and M1 macrophages, which was consistent with the substantial role of GLUT3 as important glucose importer in these immune cell populations and again validating the therapeutic rationale of this study.

The related data and discussions are shown here below for your review:

“Notably, treating the co-culture system with 100 nM BAY-876 caused significant suppression of 4T1 cells while expanding the immune cell populations associated with the adaptive antitumor immunity including DCs, M1 macrophages, CD4⁺ T cells and CD8⁺ T cells, on account of the capacity of BAY-876 to selectively inhibit GLUT1 to block tumor cell-intrinsic glucose uptake. Contrastingly, both tumor cells and immune cells were substantially suppressed by KL-11743 treatment due to the simultaneous inhibition of GLUT1 and GLUT3, leading to universal glucose uptake blockade (Figure 3l-m and Figure S16-S18). The evidence above supported the therapeutic capacity of the BAY-876-containing nanoassembly to rebalance glucose competition between TNBC cells and immune cells for mounting adaptive antitumor immune responses.”

“Evaluation of the impact of BAY-876 and KL-11743 on major immune cell populations in the co-incubation system: 4T1 cells were inoculated into 12-well plate with 1.5×10^5 units per well. The cells were cultured for 24 h, added with splenocytes of 30 times the amount of tumor cells, and then incubated with 100 nM BAY-876 or 5 μ M KL-11743 at 37°C for 24 h. The splenocytes were collected and processed with the corresponding fluorescent antibodies at room temperature for 30 min in the dark. Finally, the cells were detected by flow cytometry (CytoFLEX, Beckman Coulter).”

Figure S18. Phenotypical changes of various immune cells in the co-incubation system of 4T1 cells and splenocytes after BAY-876 or KL-11743 treatment. (a) Flow cytometry evaluation on the expansion of CD8+/CD4+ T cells. (b) Flow cytometry evaluation of DC maturation (CD11c+MHC-II+). (c) Flow cytometry evaluation on the polarization state of macrophages (F4/80+CD86+). (d) Statistical analysis of flow cytometry data in panel a-c (n=3). (1) PBS, (2) BAY-876, (3) KL-11743, (4) BAY-876+KL-11743.

5. Only CD4+ T cells and CD8+ T cells were measured in Figure 4. The authors should also monitor the activation status of major APC populations, such as dendritic cells and M1 macrophages. ?

A: Thank you for your advice. Based on your suggestion we have quantitatively analyzed the phenotypical changes in major APC populations including DCs and M1 macrophages in the co-incubation system via flow cytometry. As shown in the Figure S21, DNA-PAE@BAY-876 had the greatest promotional effect on DC and M1 macrophage populations, of which the frequencies have increased by 35.38% and 33.89% respectively compared with the control group, further indicating that DNA-PAE@BAY-876 could alleviate the immunosuppression in the co-incubation system and boost the antitumor functions of APCs and effector T cells.

The related data and discussions are shown here below for your review:

“Meanwhile, DC and M1 macrophage frequencies in the DNA-PAE@BAY-876 with pH 6.8 group have increased by 35.38% and 33.89%, respectively (Figure S21), while the ratio of IFN- γ + CD8+ T cells has also increased by 33.96% (Figure 4g).”

“Evaluation of the phenotypical changes of major immune cell populations: 4T1 cells were inoculated into 12-well plate with 1.5×10^5 units per well. The cells were cultured for 24 h, added with different samples and splenocytes of 30 times the amount of tumor cells, and then incubated at 37°C for 24 h. The splenocytes were collected and processed by the corresponding fluorescent antibodies at room temperature for 30 min in the dark. Finally, the cells were detected by flow cytometry (CytoFLEX, Beckman Coulter).”

Figure S21. Nanoassembly treatment boosted the activation of APCs. Flow cytometry evaluation of DC maturation (CD11c⁺MHC-II⁺) (a) and M1 polarization of macrophages (F4/80⁺CD86⁺) after various treatment. (b) in the co-incubation system after different treatment. (c) Statistical analysis of flow cytometry data in panel a-b (n=3).

6. In Figure 6, the authors detected Foxp3, CD25 and CTLA-4 expression in Tregs. The changes in cytokine expressions should also be included via flow cytometry.

A: Thank you for your suggestion. Extending from the phenotypical changes in Tregs after various treatment, we have further detected the expression levels of pro-inflammatory IFN- γ and anti-inflammatory IL-10 in the co-incubation system via flow cytometry, both of which are commonly used markers to determine the immunostimulatory status [Cell Discov 9, 54 (2023); Angew Chem Int Ed Engl 61, e202109500 (2022)]. As shown in the Figure S29. The frequency of CD3⁺ IFN- γ ⁺ T cells has increased by 28.72% in the DNA-PAE@BAY-876 group compared with the control

group under pH 6.8, while the frequency of CD3⁺ IL-10⁺ T cells has decreased by 28.83%. The above experimental results demonstrated that DNA-PAE@BAY-876 could potently stimulate the antitumor activity of effector T cells.

The related data and discussions are shown here below for your review:

“To illustrate the immunostimulatory potential of the nanoassemblies, we tested the frequency of IFN- γ CD3⁺ and IL-10 CD3⁺ T cells in the co-culture system of 4T1 cells and splenic immune cells after various treatment. Compared with the Control group, the population of IFN- γ CD3⁺ T cells has increased by 28.72% in the pH6.8+DNA-PAE@BAY-876 group, while the IL-10+CD3⁺ T cell population has decreased by 28.83% (Figure S29).”

“Evaluation of IFN- γ /IL-10 expression levels in CD3⁺ T cells: 4T1 cells were inoculated into 12-well plate with 1.5×10^5 per well. The cells were cultured for 24 h, added with different samples and splenocytes of 30 times the amount of tumor cells, and then incubated at 37°C for 24 h. The splenocytes were collected and incubated with PC7-anti-CD45 antibody, APC-anti-CD3 antibody, FITC-anti-IFN- γ antibody and PE-anti-IL-10 antibody at room temperature for 30 min in the dark. Finally, the cells were detected by flow cytometry (CytoFLEX, Beckman Coulter).”

Figure S29. Activation status of T cells in the co-incubation system of 4T1 cells and spleen cells after different treatment. (a) IFN- γ secretion in the co-incubation system of 4T1 cells and spleen cells after different treatment. (b) IL-10 secretion in the co-incubation system of 4T1 cells and spleen cells after different treatment. (c) Statistical analysis the flow cytometric data in panel a-b (n=3). B+C means BAY-876 combined with aptCTLA-4.

Minor issues:

1. In FigureS2 H1 - NMR, the characteristic peak positions in the compounds should be assigned and labeled.

A: Thank you for your suggestion. Based on your advice we have analyzed and labeled the major characteristic peaks of each compound in Figure S2 to enhance its informativity, and the optimized figure is shown here below for your review.

Figure S2. ¹H-NMR spectra of key products in DNA-PAE synthesis. (a) 1,4-butanediol diacrylate. (b) 4,4'-trimethylenedipyridine. (c) N₃-PEG₂₀₀₀-ACA (N₃-PEG₂₀₀₀-acrylamide). (d) N₃-PEG₂₀₀₀-PAE (N₃-PEG₂₀₀₀-β amino ester). (e) The molecular mass of N₃-PEG₂₀₀₀-PAE by GPC.

2. Western blot data in Figure 4 b should be statistically analyzed.

A: Thank you for your concern. Based on your advice we have statistically analyzed the WB data in Figure 4b-c via gray value calculation method using Image J software, and the results are shown in Figure S19a-d. The quantitative analysis results were consistent with the visual trends that the

DNA-PAE@BAY-876 treatment could effectively trigger tumor cell apoptosis for inhibiting their growth.

Figure 4. DNA-PAE@BAY-876 treatment mounted potent antitumor effect in 4T1-splenic immune cell co-incubation system. (a) Apoptosis ratio of 4T1 cells in the co-incubation system after different treatment at 37°C for 24 h. (b-c) Apoptosis effect of 4T1 cells with pH7.4 (b) and

pH6.8 (c) by western blotting at 37°C for 24 h. (I) NC, (II) PAE, (III) aptCTLA-4, (IV) aptPD-L1, (V) BAY-876, (VI) DNA-PAE, (VII) DNA-PAE@BAY-876. Note: the molecular weight of Bcl-2 is 26 KDa; the molecular weight of Bax is 21 KDa; the molecular weight of Tubulin is 50 KDa. (d-e) Glucose uptake by 4T1 cells or T cells after different treatments under co-culture condition at 37°C for 24 h (n=3). (f-g) Flow cytometric analysis on the expansion of CD4⁺/CD8⁺ T cells, IFN- γ +CD8⁺ T cells in the co-incubation system after different treatment at 37°C for 24 h. (h) Statistical analysis of data shown in panel f-g by flow cytometry (n=3). (1) NC, (2) PAE, (3) aptCTLA-4, (4) aptPD-L1, (5) BAY-876, (6) DNA-PAE, (7) DNA-PAE@BAY-876.

Figure S19. Biochemical changes and survival of 4T1 cells after different treatment. (a-d) Statistic analysis on the expression levels of key apoptosis-related markers in 4T1 cells after different treatment via WB assay (n=3). (e-f) Survival rate of 4T1 cells and HC11 cells after different treatment (n=3).

3. The figure texts in Figure5 f is too small and needs to be adjusted.

A: Thank you for your suggestion. We have adjusted the font size of the figure text in Figure 5f of the original manuscript to enhance its readability.

Figure 5. Nanoassembly-enhanced 4T1 cell recognition and binding by aptPD-L1. (a) Schematic illustration on the nanoassembly-mediated metabolic rewiring of 4T1 cells for enhanced

ICT. (b) Evaluation on the dose-dependent impact of BAY-876 on PD-L1 glycosylation in 4T1 cells with pH7.4/6.8 at 37°C for 24 h. The concentration of IFN- γ was 1 $\mu\text{g}\cdot\text{mL}^{-1}$. Note: the molecular weight of PD-L1 is 33-70 KDa; the molecular weight of Tubulin is 50 KDa. (c) Impact of DNA-PAE@BAY-876 on the PD-L1 glycosylation level in multiple cell types at 37°C for 24 h. The concentration of DNA-PAE@BAY-876 was 100 nM and the concentration of IFN- γ was 1 $\mu\text{g}\cdot\text{mL}^{-1}$. (d) Impact of UDP-GlcNAc and F6P on the PD-L1 glycosylation levels in 4T1 cells after BAY-876 treatment with high/low-glucose media. The concentration of BAY-876, UDP-GlcNAc and IFN- γ was 100 nM, 0.1 mM and 1 $\mu\text{g}\cdot\text{mL}^{-1}$, respectively. (e) UDP-GlcNAc abundance in 4T1 cells under different conditions (n=4). (I, V) Control; (II, VI) BAY-876; (III, VII) BAY-876+F6P; (IV, VIII) BAY-876+UDP-GlcNAc. (f) Schematic illustration of Hexosamine Biosynthesis Pathway (HBP). (g) 3D plot on the correlation between UDP-GlcNAc and F6P with 2-NBDG after treatment by BAY-876 under graded concentrations. (h-k) Impact of BAY-876 on the competitive combination between aptPD-L1/PD1 with PD-L1 on 4T1 cell surface.

4 . Some literatures about the glucose metabolism of macrophages should be added.

A: Thank you for the suggestion. Several recent publications regarding the glucose metabolism patterns of macrophages have been cited in support of the therapeutic mechanism of this study. Their reference numbers are [20], [21], respectively.

5. H&E staining in Figure7 is unclear and should be redone.

A: Thank you for your advice. We have redone the H&E staining for the tissue samples to enhance their resolution and quality, and the replacement figure is shown here below for your review:

Figure 7. Antitumor evaluation of DNA-PAE@BAY-876 in vivo. (a) Schematic diagram of treatment schedule of DNA-PAE@BAY-876 nanoassemblies on 4T1 tumor-bearing mice. (b) Visual comparison of tumors from Balb/c mice after different treatment. (c) In vivo bioluminescence images on the tumor progression in Balb/c mice in different groups. (d) Tumor volume changes after different treatment (n=5). (e) Final tumor weight in different groups at the end of the 21 days period (n=5). (I) NC, (II) PAE, (III) aptCTLA-4, (IV) aptPD-L1, (V) BAY-876, (VI) DNA-PAE, (VII) DNA-PAE@BAY-876.

DNA-PAE, (VII) DNA-PAE@BAY-876. (f) Body weight changes of Balb/c mice in different groups during treatment (n=5). (g-h) TUNEL and H&E staining of tumor samples. (i-k) ELISA assay of glucose, lactic acid and UDP-GlcNAc abundance in tumor samples (n=3).

Reviewer #4 (Remarks to the Author): with expertise in breast cancer, immunology, metabolism

Ren et al develop a TME-responsive bispecific aptamer-based nanoassemblies encapsulating GLUT1 inhibitor BAY-876 (DNA-PAE@BAY-876), which could synergistically inhibit TNBC-selective glycolysis and promote immunostimulatory in the tumor microenvironment (TME). Although the theory is supported by plenty of experiments, the data presented are generally unconvincing, especially the exclusive tumor model used in this study. The study would benefit from additional experiments to establish the therapeutic relevance of DNA-PAE@BAY-876 in triple negative breast cancer (TNBC).

A: Thank you for the constructive comments for our study. We are grateful for the helpful criticisms and carried out substantial amount of experiments to enhance the convincingness and therapeutic relevance of this study. Specifically, we have constructed humanized mouse model bearing MDA-MB-231 TNBCs and tested the therapeutic effects of the nanoassembly in vivo. Meanwhile, the robustness of the nanoassembly-evoked systemic antitumor immunity was thoroughly evaluated by lung metastasis model. Furthermore, we have comprehensively analyzed the drug release and endocytosis features of the nanoassembly under tumor-relevant conditions. We ensure that every concern you kindly raised has been properly addressed in the revised manuscript. We hope that our revisions could address your concerns and mobilize this study closer to the high standard of Nat Comm.

Major points:

1. How efficient are the PAE-based handles conjugated onto aptPD-L1 and aptCTLA-4? What about the encapsulation efficiency of the BAY-876? And how efficiently is the released BAY-876 entering tumor cells in the TME?

A: Thank you for your concern. Based on your advice we have carried out substantial amount of experiments to characterize the physical and chemical features of the DNA-PAE@BAY-876 nanoassemblies. To start with, we first measured the conjugation efficiency of PAE-based handles onto Cy5-aptPD-L1 and FAM-aptCTLA-4 via fluorescence spectroscopy. As shown in Figure S5a-d, the conjugation efficiency of PAE onto aptamers was determined via standard calibration method, which was 81.24% and 84.79% for aptPD-L1 and aptCTLA-4, respectively.

To determine the encapsulation efficiency of BAY-876 by the nanoassemblies, we employed the UV-vis spectrophotometer to measure the BAY-876 abundance in nanoassemblies and the loading

buffer for qualitative analysis via standard curve method. As shown in Figure S5e, the encapsulation rate of DNA-PAE@BAY-876 was around 0.1% while the relative encapsulation amount of BAY-876 in the final nanoassembly was around 51%, which was attributed to the large molecular weight difference between BAY-876 and DNA-PAE. Furthermore, we have quantitatively measured in vivo delivery efficacy of the DNA-PAE@BAY-876 nanoassembly via liquid chromatography-triple quadrupole mass spectroscopy (Figure S31c-e) and found that the BAY-876 concentration in tumor tissues was around 126 nM, which was sufficient to block tumor-intrinsic glucose uptake while avoiding negative impact on tumor-resident immune cells. Overall, these results collectively supported the utility of the DNA-PAE@BAY-876 nanoassembly as a robust drug delivery platform for enhanced ICT.

The related data and discussions are shown here below for your review:

“In addition, using standard curve calibration method based on the fluorescence data, the grafting rates of aptPD-L1 and aptCTLA-4 with PAE handles were determined to be 81.24% and 84.79%, respectively (Figure S5a-d).”

“Quantitative UV-vis spectroscopic analysis via a standard curve calibration approach showed that the relative encapsulation amount of BAY-876 in the final nanoassembly was around 51% and the encapsulation efficiency was around 0.1% (Figure S5e), which was attributed to the large molecular weight difference between BAY-876 and DNA-PAE.”

“In addition, aptCTLA-4 and aptPD-L1 were labeled with FITC to synthesize DNA-PAE@BAY-876, and the fluorescence confocal microscopic data showed that aptCTLA-4 and aptPD-L1 successfully targeted CD45⁺ lymphocytes and 4T1 tumor cells, respectively (Figure S31a-b). Meanwhile, the concentration of BAY-876 in the tumor microenvironment was determined to be 126 nM using the liquid chromatography-triple quadrupole mass spectrometer (Figure S31c-e). The above experimental data supported the tumor-targeted delivery efficacy of the DNA-PAE@BAY-876 nanoassembly in vivo.”

“1.25 In vivo delivery analysis of DNA-PAE@BAY-876 Firstly, FITC-DNA-PAE@BAY-876 nanoassembly was synthesized using FITC-labeled aptPD-L1 or FITC-labeled aptCTLA-4. Meanwhile, the 4T1 tumor-bearing Balb/c mouse model was constructed and treated with FITC-

DNA-PAE@BAY-876 through intravenous injection. Then, the tumors at about 600 mm³ were collected.

A portion of the collected tumor tissue was frozen-sectioned and these sections were fixed with 4% paraformaldehyde at room temperature for 30 min. After cleaning with PBS, the sections were blocked with 5% BSA for 2 h, then added with APC-anti-CD45 antibody or APC-anti-PD-L1 antibody at 4°C overnight, then stained for 10 min with DAPI in the dark, finally sealed with anhydrous glycerol for observation by laser confocal microscope (Leica TCS SP8, Germany).

Another batch of the tumor tissue was ground to powder using liquid nitrogen, and then lysed with RIPA lysate (containing 1% PMSF) for 30 min at 4°C and centrifuged at 12000 rpm for 10 min. The collected supernatant was filtered by ultrafiltration tube (MWCO= 600 Da) and the BAY-876 content in tumors was quantified by a liquid chromatography-triple quadrupole mass spectrometer. At the same time, BAY-876-methanol solutions with different concentrations were prepared to establish the standard curve by liquid chromatography-triple quadrupole mass spectrometer.”

Figure S5. Standard curves of the fluorescence intensity for Cy5-aptPD-L1 (a-b) and FAM-aptCTLA-4 (c-d) at different concentrations (n=3). (e) Standard curves of the UV-vis absorption for BAY-876 at different concentrations (n=3).

Figure S31. The distribution of DNA-PAE@BAY-876 in the tumor of 4T1 tumor-bearing mice. (a-b) The distribution of aptCTLA-4 and aptPD-L1 in vivo. (c-e) The quantification of BAY-876 content in the tumor by liquid chromatography-triple quadrupole mass spectrometer.

2. The topic of the manuscript focuses on TNBC, however, just a 4T1 tumor model resembling human TNBC is used in in vitro and in vivo studies. Thus, the current data are not convincing enough to illustrate the clinical implication of DNA-PAE@BAY-876.

A: Thank you for your advice. We agree with the reviewer that the 4T1 tumor-only evaluations were insufficient to illustrate the therapeutic effects of the DNA-PAE@BAY-876 nanoassemblies and have thus constructed the humanized HSC-NOG-EXL mouse model bearing human MDA-MB-231 TNBCs to further demonstrate the therapeutic potency of the nanoassembly under clinically relevant conditions (Figure S43). Indeed, to test whether aptPD-L1 and aptCTLA-4 had targeting ability in humanized system, we first extracted splenic immune cells from humanized HSC-NOG-EXL mice and co-incubated them with MDA-MB-231 cells, then added with aptPD-L1 and aptCTLA-4 to observe their target ability, both of which showed preferential binding with their designated cellular

targets (Figure S44), suggesting their potential application in clinical scenarios. Similar to the in vivo experiment on 4T1 tumor-bearing mouse models, we first divided MDA-MB-231 tumor-bearing humanized mice into 7 groups and then administered NC, PAE, aptCTLA-4, aptPD-L1, BAY-876, DNA-PAE, DNA-PAE@BAY-876 every 2-3 days until day 21. Analysis on tumor volume and weight changes (Figure S45a-b) revealed an identical trend like the 4T1 tumor models. Specifically, PAE treatment had no significant inhibitory effect in MDA-MB-231 tumors due to the lack of therapeutic effects. Meanwhile, aptCTLA-4 and aptPD-L1 had poor anti-tumor effects due to the intrinsic ICT resistance of TNBCs in the clinics. BAY-876 showed obvious tumor inhibition on account of its glucose uptake blockade ability that caused tumor starvation. DNA-PAE@BAY-876 nanoassembly showed the best anti-tumor effect with final mean tumor volume and weight of 93 mm³ and 0.24 g, respectively (Figure S45c-d). Furthermore, H&E and TUNEL staining on extracted MDA-MB-231 tumors and revealed that tumor tissues in the DNA-PAE@BAY-876 group showed the highest apoptosis ratio (Figure S45e-f). Overall, these observations confirmed that DNA-PAE@BAY-876 nanoassemblies showed good antitumor potency against MDA-MB-231 tumors on humanized HSC-NOG-EXL mouse models.

In addition, we have analyzed the immune composition in MDA-MB-231 tumors after various treatment to determine the immunostimulatory effect of the DNA-PAE@BAY-876 nanoassemblies. Compared with the control group, tumor infiltration of DCs and M1 macrophages have increased by 17.89% and 21.96% in the DNA-PAE@BAY-876 group (Figure S46b-c), accompanied with a substantial expansion of tumor infiltrating CD8⁺ and CD4⁺ T cells by 9.03% and 18.06% (Figure S46a), while. In contrast, pro-tumorigenic immune cell populations including M2 macrophages, Tregs and MDSCs in MDA-MB-231 tumors after treatment with DNA-PAE@BAY-876 nanoassembly have decreased by 21.50%, 12.00% and 12.45%, respectively (Figure S46d-f), suggesting the successful remodeling of the MDA-MB-231 tumor immune microenvironment towards an anti-tumorigenic state. The above experimental results in MDA-MB-231 tumor-bearing humanized mouse model supported our hypothesis that DNA-PAE@BAY-876 nanoassembly could exert enhanced ICT effect for efficient TNBC inhibition under clinically relevant conditions.

The related data and discussions are shown here below for your review:

“8. Therapeutic evaluation of DNA-PAE@BAY-876 in humanized MDA-MB-231 tumor-

bearing HSC-NOG-EXL mice

To further demonstrate the clinical translational potential of the DNA-PAE@BAY-876 nanoassembly, we constructed the humanized HSC-NOG-EXL mouse model (Figure S43) and thoroughly investigate its therapeutic impact thereof. To start with, we first extracted splenic immune cells from the humanized HSC-NOG-EXL mice to establish co-incubation system with MDA-MB-231 cells to investigate the targeting capacity of aptPD-L1 and aptCTLA-4 to human TNBC and immune cells. Notably, both aptamers showed good binding affinity with the designated cellular targets with high specificity (Figure S44), providing the mechanistic basis for the subsequently evaluations on humanized TNBC mouse models. Subsequently, by adapting the experimental set-up of 4T1 tumor-bearing mouse models, we divided humanized MDA-MB-231 tumor-bearing mice into 7 groups (n=3) and then treated them with NC, PAE, aptCTLA-4, aptPD-L1, BAY-876, DNA-PAE, DNA-PAE@BAY-876 periodically until 21 days. Evaluations on the anti-TNBC efficacy of all sample groups revealed a similar trend like that on 4T1 tumor-bearing mouse models, where the DNA-PAE@BAY-876 nanoassembly showed the most pronounced anti-tumor effect with a final mean tumor volume and weight of 93 mm³ and 0.24 g, respectively (Figure S45c-d). The nanoassembly-induced potent inhibition of MDA-MB-231 tumors in humanized mouse models was further supported by the H&E and TUNEL staining on the extracted tumor tissues, which revealed that DNA-PAE@BAY-876 nanoassembly induced the largest dead MDA-MB-231 cell population among all groups (Figure S45e-f). Extending from the antitumor efficacy of individual groups on humanized TNBC mouse models, we further analyzed the immune status in MDA-MB-231 tumors to elucidate the therapeutic mechanisms of the nanoassembly. The DNA-PAE@BAY-876 substantially boosted the tumor infiltration of immune cells mediating the anti-tumorigenic immune responses. Specifically, the frequencies of mature DCs and M1 macrophages in the DNA-PAE@BAY-876 have increased by 17.89% and 21.96% compared with the control group, while frequencies of CD8⁺/CD4⁺ T cells have increased by 9.03% and 18.06% (Figure S46a-c). In contrast, frequencies of tumor-residing M2 macrophages, Tregs and MDSCs in DNA-PAE@BAY-876-treated MDA-MB-231 tumors have decreased by 21.50%, 12.00% and 12.45%, respectively (Figure S46d-f). In addition, the EICD in humanized MDA-MB-231 tumor-bearing mice after treatment with the nanoassemblies were evaluated using a similar set-up to the 4T1

tumor-bearing mouse models. Specifically, DC and Tcm frequencies in lymph nodes from the DNA-PAE@BAY-876 group were 11.29% and 15.39% higher than the control group, respectively, accompanied with a 28.97% and 25.00% increase in serum IFN- γ and TNF- α levels as well as a 11.50% decrease in serum IL-10 levels, indicating the successful nanoassembly-mediated regulation of EICD to boost the antitumor immunity. These observations collectively confirmed the antitumor potency of DNA-PAE@BAY-876 nanoassemblies against MDA-MB-231 tumors on humanized mouse models and supported its clinical potential for TNBC treatment on real-life patients.”

“The establishment of humanized MDA-MB-231 tumor-bearing HSC-NOG-EXL mouse model: Firstly, 100 μ L 2×10^6 MDA-MB-231 cells were injected subcutaneously into each humanized HSC-NOG-EXL mouse. When the tumors grew to about 100 mm³, the mice were randomly divided into 7 groups (n=3) and subjected to different treatment (PBS, 24.01 μ g·mL⁻¹·g⁻¹ PAE, 12.35 μ g·mL⁻¹·g⁻¹ aptCTLA-4, 12.22 μ g·mL⁻¹·g⁻¹ aptPD-L1, 0.05 μ g·mL⁻¹·g⁻¹ BAY-876, 48.31 μ g·mL⁻¹·g⁻¹ DNA-PAE, 48.87 μ g·mL⁻¹·g⁻¹ DNA-PAE@BAY-876) through intravenous injection once every two days. After 21 days of treatment, all mice were sacrificed, their tumor tissues and lymph nodes were collected for post-analysis while the blood of mice from the eyeball was collected using the heparin tube and then serum was collected after centrifugation for ELISA assay of multiple cytokines.

A portion of the collected tumor tissue and lymph nodes were washed twice with PBS, placed on 0.4 μ m filter membrane, added with high-glucose DMEM medium, carefully pulverized with the syringe head, then mixed with 5 mL red blood cell lysis buffer and stood for 10 min, and centrifuged at 2000 rpm for 5 min. The supernatant was removed, the cells were re-suspended with PBS, then stained with corresponding fluorescent antibodies for 30 min to detect the infiltration of various immune cells by flow cytometry.

Another portion of the tumor tissue was frozen-sectioned, fixed with 4% paraformaldehyde at room temperature for 30 min. After permeation with 0.5% Triton X-100 for 10 min, the sections were incubated with TUNEL test solution at 37°C for 60 min, then added with DAPI for 10 min in the dark and sealed with glycerol anhydrous after washed with PBS. The mounted tissue sections were finally imaged by a laser confocal microscope (Leica TCS SP8, Germany).”

Figure S43. Construction of humanized MDA-MB-231 tumor-bearing HSC-NOG-EXL mice. (a) Schematic diagram of humanized MDA-MB-231 tumor-bearing mice and the treatment schedule. (b) Flow cytometry detection of the peripheral blood of HSC-NOG-EXL mice after tail intravenous injection with HSC cells for 55 days for validation.

Figure S44. Targeting ability of aptPD-L1 (a) and aptCTLA-4 (b) in the co-incubation system of MDA-MB-231 cells and splenic immune cells from humanized HSC-NOG-EXL mice.

Figure S45. Antitumor evaluation of DNA-PAE@BAY-876 in humanized MDA-MB-231 tumor-bearing HSC-NOG-EXL mice. (a) Photographical images of the MDA-MB-231 tumors through the treatment period. (b) Visual comparison of tumors from HSC-NOG-EXL mice after different treatment. (c-d) Final tumor volume and weight in different groups at the end of the 21 days period (n=3). (e-f) H&E and TUNEL staining of tumor samples. (I) NC, (II) PAE, (III) aptCTLA-4, (IV) aptPD-L1, (V) BAY-876, (VI) DNA-PAE, (VII) DNA-PAE@BAY-876.

Figure S46. Evaluation on DNA-PAE@BAY-876-mediated immunotherapeutic effect in MDA-MB-231 tumor-bearing humanized HSC-NOG-EXL mice. (a) Flow cytometry detection on the tumor infiltration of CD8⁺ T cells and CD4⁺ T cells. (b) Flow cytometry detection on the tumor infiltration of DCs (CD11c+HLA-DR⁺). (c) Flow cytometry detection on the tumor-infiltration of M1 macrophages (CD14+CD80⁺). (d) Flow cytometry detection on the tumor infiltration of M2 macrophages (CD14+CD206⁺). (e) Flow cytometry detection on the tumor infiltration of Tregs (CD4+CD25+Foxp3⁺). (f) Flow cytometry detection on the tumor infiltration of MDSCs (HLA-DR-CD11b⁺). (g) Statistical analysis of the flow cytometric data in panel a-f (n=3). (1) NC, (2) PAE, (3) aptCTLA-4, (4) aptPD-L1, (5) BAY-876, (6) DNA-PAE, (7) DNA-PAE@BAY-876.

3. Figure 3 c: As the DNA aptamer and the cell membrane are both negatively charged, how are the DNA-PAE nanoassemblies endocytosed by cells under pH of 7.4?

A: Thank you for your concern. We understand the reviewer's concern regarding the potential electrostatic repulsion between the DNA aptamers and cell membranes that may prevent the endocytic uptake by tumor cells. We would like to state that the cell membrane is not a homogeneously negatively-charged entity but rather presents various neutrally or even positively charged areas such as lipid rafts and membrane-bound charged proteins [Nat Rev Microbiol 3, 238-250 (2005); Science 327, 46-50 (2010); Nature 387, 569-572 (1997)], which may interact with the exogenous nanospecies and facilitate their endocytic uptake [ACS Nano 12, 5078-5084 (2018); Nat Mater 8, 543-557 (2009)]. Similar effects have also been observed in various nanoformulations based on negatively charged poly (lactic-co-glycolic acid) [Nat Commun 7, 13193 (2016); J Control Release 161, 505-522 (2012); Biomaterials 26, 2713-2722 (2005)]. Notably, aptPD-L1 in the DNA-PAE nanoassembly could specifically bind with PD-L1 receptors overexpressed on TNBC cell surface and enable efficient receptor-mediated endocytosis even under neutral pH of 7.4. To test this hypothesis, we first employed PD-L1-inhibiting siRNAs (siPD-L1) to knock down PD-L1 expression in 4T1 cells [J Nanobiotechnology 20, 96 (2022)] (Figure S12a) and then incubated the PD-L1-KO 4T1 cells with FITC-labeled DNA-PAE nanoassemblies to monitor their uptake. As shown by the fluorescence imaging results, non-functional siRNA (siNC)-treated 4T1 cells showed significantly higher nanoassembly uptake than the PD-L1-KO 4T1 cells and confirmed the contribution of aptPD-L1-PD-L1 interaction to nanoassembly uptake. Overall, these observations collectively supported that DNA-PAE nanoassemblies could be taken in by 4T1 tumor cells under neutral pH despite the potential repulsive forces from the negative charges.

The related data and discussions are shown here below for your review:

“To further elucidate the uptake mechanism of the DNA-PAE nanoassemblies, we used siRNA to knock down the PD-L1 expression on the 4T1 cell surface and found that this caused a significant reduction in their uptake amount under neutral pH, showing that the DNA-PAE nanoassembly endocytosis by 4T1 cells under neutral pH was mediated by the interaction between aptPD-L1 and membrane PD-L1 (Figure S12).”

“**1.26 PD-L1 gene knock-down:** 4T1 cells were inoculated into 6-well plates with 5×10^5 per well or the 20 mm confocal dish with 1.5×10^5 per dish. After culturing for 24 h, the cells were treated by siNC or siPD-L1 with PEI for 8 h, then washed twice with PBS and incubated at 37°C for 24 h, finally observed by Western Blotting or photographed by a laser confocal microscope (Leica TCS SP8, Germany).”

Figure S12. Investigation of DNA-PAE endocytosis by 4T1 cells under neutral pH. (a) WB analysis on siPD-L1-mediated PD-L1 knock-down effect in 4T1 cells. (b) Endocytosis of DNA-PAE by 4T1 cells without/with PD-L1 knock-down.

4. Figure 4g: The ratio of IFN-g+CD8+ T cells in the control group under pH of 7.4 is two-fold higher than that under pH of 6.8, why that?

A: Thank you for the concern. We would like to state that the significantly lower frequency of IFN-g+CD8+ T cells in the control group under pH 6.8 was due to the acidity induced immunosuppression. Previous insights collectively demonstrated that the excessive acidity in TME is a major contributor to the immunosuppressive traits, which could engage immunosuppressor cells to impair the effector function of cytotoxic T cells [Cancer Cell 40, 1207-1222.e10 (2022); Nature 591, 652-658 (2021)]. Interestingly, the results also showed that the DNA-PAE@BAY-876 nanoassembly in the present study could overcome the acidity-induced immunosuppression and mount robust T cell-mediated antitumor immunity.

Figure 4. DNA-PAE@BAY-876 treatment mounted potent antitumor effect in 4T1-splenic immune cell co-incubation system. (a) Apoptosis ratio of 4T1 cells in the co-incubation system after different treatment at 37°C for 24 h. (b-c) Apoptosis effect of 4T1 cells with pH7.4 (b) and pH6.8 (c) by western blotting at 37°C for 24 h. (I) NC, (II) PAE, (III) aptCTLA-4, (IV) aptPD-L1, (V) BAY-876, (VI) DNA-PAE, (VII) DNA-PAE@BAY-876. Note: the molecular weight of Bcl-2

is 26 KDa; the molecular weight of Bax is 21 KDa; the molecular weight of Tubulin is 50 KDa. (d-e) Glucose uptake by 4T1 cells or T cells after different treatments under co-culture condition at 37°C for 24 h (n=3). (f-g) Flow cytometric analysis on the expansion of CD4⁺/CD8⁺ T cells, IFN- γ +CD8⁺ T cells in the co-incubation system after different treatment at 37°C for 24 h. (h) Statistical analysis of data shown in panel f-g by flow cytometry (n=3). (1) NC, (2) PAE, (3) aptCTLA-4, (4) aptPD-L1, (5) BAY-876, (6) DNA-PAE, (7) DNA-PAE@BAY-876.

5 . TNBC is associated with high risk of distant metastasis such as lung. Can DNA-PAE@BAY-876 treatment reduce metastatic lung burden through remodeling glycometabolism in TME?

A: Thank you for your suggestion. Based on your advice we have established 4T1 lung metastasis mouse model to test the robustness of the nanoassembly-evoked systemic antitumor immunity. On day 50, the lung metastasis model was collected from the 4T1 tumor-bearing mice, then fixed with 4% paraformaldehyde and photographed. As shown in Figure 8h, DNA-PAE@BAY-876 treatment effectively inhibited the lung metastasis of 4T1 tumors compared with the control group, for which the number of metastasis nodules was 71.42% lower than the PBS-treated controls. These observations evidently supported that treating TNBCs with DNA-PAE@BAY-876 nanoassemblies could elicit potent systemic antitumor immunity to reduce the risk of lung metastasis.

The related data and discussions are shown here below for your review:

“TNBC is associated with high risk of metastasis and recurrence. Extending from the immunostimulatory activity of the DNA-PAE@BAY-876 nanoassemblies in vivo, we further investigated if the nanoassembly-evoked systemic antitumor immunity could eliminate metastatic TNBCs using a 4T1 lung metastasis mouse model. As shown in Figure 8h, DNA-PAE@BAY-876 treatment effectively inhibited the lung metastasis of 4T1 tumors compared with the control group, for which the number of metastasis nodules was 71.42% lower than the PBS-treated controls. These observations evidently supported that treating TNBCs with DNA-PAE@BAY-876 nanoassemblies could elicit potent systemic antitumor immunity to reduce the risk of TNBC lung metastasis, which may add to the application potential of the nanoassembly-augmented ICT for TNBC management in the clinics.”

“**Evaluation of lung metastasis of TNBC after nanoassembly treatment:** Firstly, $100 \mu\text{L} \times 10^6$ 4T1 cells were injected subcutaneously into each Balb/c mouse, when the tumor grew to about 100 mm^3 , the mice were randomly divided into 7 groups at 6 mice per group and subjected to different treatment (PBS, $24.01 \mu\text{g} \cdot \text{mL}^{-1} \cdot \text{g}^{-1}$ PAE, $12.35 \mu\text{g} \cdot \text{mL}^{-1} \cdot \text{g}^{-1}$ aptCTLA-4, $12.22 \mu\text{g} \cdot \text{mL}^{-1} \cdot \text{g}^{-1}$ aptPD-L1, $0.05 \mu\text{g} \cdot \text{mL}^{-1} \cdot \text{g}^{-1}$ BAY-876, $48.31 \mu\text{g} \cdot \text{mL}^{-1} \cdot \text{g}^{-1}$ DNA-PAE, $48.87 \mu\text{g} \cdot \text{mL}^{-1} \cdot \text{g}^{-1}$ DNA-PAE@BAY-876) through intravenous injection once every two days. On day 50, the mice were sacrificed to extract the lungs, which fixed with 4% paraformaldehyde and photographed to count the number of lung metastasis nodules.”

Figure 8. Evaluation on DNA-PAE@BAY-876-mediated immunotherapy in vivo. (a-b) Flow cytometry detection on the tumor infiltration of Foxp3+ Tregs and IL-10+ Tregs. (c) Flow cytometry evaluation on the tumor-infiltration of total immune cells (CD45+). (d) Flow cytometry detection on the tumor infiltration of IFN- γ + CD8+ T cells. (e) Statistical analysis of data shown in panel a-d by flow cytometry (n=3). (1) NC, (2) PAE, (3) aptCTLA-4, (4) aptPD-L1, (5) BAY-876, (6) DNA-PAE, (7) DNA-PAE@BAY-876. (f) Immunofluorescence images on the tumor-infiltration of CD4+ and CD8+ T cells in different groups. (g) Immunofluorescence images of IL-10 in tumor tissues after different treatment. (h) Photographs of lung metastasis inhibition in 4T1 tumor-bearing mice after different treatments.

6. Effector immune cell deployment (EICD) is very important in immune checkpoint blockade therapy of cancers. The authors can discuss the role of DNA-PAE@BAY-876 in regulating EICD to enhance ICI efficacy.

A: Thank you for your advice. We totally agree with the reviewer that EICD is a revolutionary concept in immunotherapy that provides an overarching perspective on the initiation, activation, circulation, recruitment, infiltration and survival of effector immune cells in lymph nodes, peripheral blood and tumor microenvironment [Nat Commun 11, 6268 (2020); Sci China Life Sci 66, 1930-1933 (2023); Trends Immunol 43, 523-545 (2022)], which has emerged as a widely-accepted standard for the implementation of various immunotherapeutic modalities with substantial scientific and clinical significance. Based on the insights from relevant reports in recent years [Sci China Life Sci 66, 1930-1933 (2023); Trends Immunol 43, 523-545 (2022)], we have systematically investigated the EICD in the present studies using established schemes and protocols.

Specifically, we extracted lymph nodes of 4T1 tumor-bearing mice and MDA-MB-231 tumor-bearing humanized HSC-NOG-EXL mice to detect the phenotypical changes of DCs and central memory T cells (Tcms). Meanwhile, mouse serum was collected from the eyeballs of both mouse models to detect the secretion levels of key cytokines related to the immunotherapeutic effects. As shown in Figure S39, lymph nodes from the DNA-PAE@BAY-876 group in 4T1-bearing mice showed significantly larger DC and Tcm populations compared with the control group, which has increased by 14.67% and 14.26%, respectively. Meanwhile, the secretion levels of pro-inflammatory IFN- γ and TNF- α in serum have increased by 33.65% and 30.87%, while the secretion

level of anti-inflammatory IL-10 has decreased by 11.95%. Similarly, analysis on samples extracted from MDA-MB-231 tumor-bearing humanized HSC-NOG-EXL mice revealed an identical trend (Figure S47), for which the DC and Tcm frequencies in lymph nodes from the DNA-PAE@BAY-876 group were 11.29% and 15.39% higher than the control group, respectively, accompanied with a 28.97% and 25.00% increase in serum IFN- γ and TNF- α levels as well as a 11.50% decrease in serum IL-10 levels. These observations collectively demonstrated that the DNA-PAE@BAY-876 treatment could regulate EICD to mount potent local and systemic adaptive anti-tumor immunity for effective TNBC treatment.

The related data and discussions are shown here below for your review:

“In order to evaluate the overall effector immune cell deployment (EICD) in 4T1 tumor-bearing mice with DNA-PAE@BAY-876 treatment, the lymph nodes of 4T1 tumor-bearing mice to detect the phenotypical changes of DCs and central memory T cells (Tcms), while mouse serum was collected from the eyeballs of both mouse models to detect the secretion levels of key cytokines related to the immunotherapeutic effects. As shown in Figure S39, lymph nodes from the DNA-PAE@BAY-876 group showed significantly larger DC and Tcm populations compared with the control group, which has increased by 14.67% and 14.26%, respectively. Meanwhile, the secretion levels of pro-inflammatory IFN- γ and TNF- α in serum have increased by 33.65% and 30.87%, while the secretion level of anti-inflammatory IL-10 has decreased by 11.95%. These data showed that the DNA-PAE@BAY-876 treatment could regulate EICD to mount potent local and systemic adaptive anti-tumor immunity for effective TNBC treatment.”

“In addition, the EICD in humanized MDA-MB-231 tumor-bearing mice after treatment with the nanoassemblies were evaluated using a similar set-up to the 4T1 tumor-bearing mouse models. Specifically, DC and Tcm frequencies in lymph nodes from the DNA-PAE@BAY-876 group were 11.29% and 15.39% higher than the control group, respectively, accompanied with a 28.97% and 25.00% increase in serum IFN- γ and TNF- α levels as well as a 11.50% decrease in serum IL-10 levels, indicating the successful nanoassembly-mediated regulation of EICD to boost the antitumor immunity. These observations collectively confirmed the antitumor potency of DNA-PAE@BAY-876 nanoassemblies against MDA-MB-231 tumors on humanized mouse models and supported its

clinical potential for TNBC treatment on real-life patients.”

Figure S39. Effector immune cell deployment (EICD) in 4T1 tumor-bearing mice after various treatment. (a) Flow cytometry analysis on the maturation status of lymph node-resident DCs (CD11c+MHC-II+). (b) Flow cytometry analysis on the lymph node-resident central memory T cell populations (CD197+CD62L+). (c-e) Serum cytokine levels in mice after 21 days of different treatment (n=4). (f-g) Statistical analysis of the flow cytometric data in panel a-b (n=3).

Figure S47. Effector immune cell deployment (EICD) in MDA-MB-231 tumor-bearing humanized HSC-NOG-EXL mice. (a) Flow cytometry detection on the lymph node-resident DCs

(CD11c+HLA-DR+). (b) Flow cytometry detection on the lymph node-resident central memory T cells (CD45RA-CD197+). (c-e) Cytokine secretion levels in humanized mice serum after 21 days of different treatment (n=3). (f-g) Statistical analysis of the flow cytometric data in panel a-b (n=3).

Minor points:

1. The reference citations in the text should be in accordance with the instructions for authors.

A: Thank you for the correction. Based on your advice we have comprehensively corrected the reference format according to the author guidelines by Nature Communications. Some representative examples are shown here below for your review:

“12. Li CW, et al. Eradication of Triple-Negative Breast Cancer Cells by Targeting Glycosylated PD-L1. *Cancer Cell* **33**, 187-201.e10 (2018).

20. Shi Q, et al. Increased glucose metabolism in TAMs fuels O-GlcNAcylation of lysosomal Cathepsin B to promote cancer metastasis and chemoresistance. *Cancer Cell* **40**, 1207-1222.e10 (2022).

21. Liu J, Cao X. Glucose metabolism of TAMs in tumor chemoresistance and metastasis. *Trends Cell Biol*, S0962-8924(23)00049-1 (2023).

23. Li T, et al. O-GlcNAc Transferase Links Glucose Metabolism to MAVS-Mediated Antiviral Innate Immunity. *Cell Host Microbe* **24**, 791-803.e6 (2018).

41. Du Y, et al. Membrane-anchored DNA nanojunctions enable closer antigen-presenting cell-T-cell contact in elevated T-cell receptor triggering. *Nat Nanotechnol* **18**, 818-827 (2023).

51. Olszewski K, et al. Inhibition of glucose transport synergizes with chemical or genetic disruption of mitochondrial metabolism and suppresses TCA cycle-deficient tumors. *Cell Chem Biol* **29**, 423-435.e10 (2022).

57. Li L, et al. TLR8-Mediated Metabolic Control of Human Treg Function: A Mechanistic Target for Cancer Immunotherapy. *Cell Metab* **29**, 103-123.e5 (2019).”

2. In Figure 1 b, the CD28 icon is confused with that of BAY-876 due to their identical color.

A: Thank you for your correction. We have changed the color of CD28 icon from blue to green to avoid misunderstanding, and the corrected figure is shown here below for your review:

Figure 1. DNA-PAE@BAY-876 nanoassemblies reprogram immunosuppressive TME through TNBC-selective glycolysis inhibition and bispecific PD-L1/CTLA-4 antagonization, leading to enhanced ICT efficacy. (a) Synthetic route of DNA-PAE@BAY-876 nanoassemblies through the spontaneous organization of the molecular building blocks. (b) Schematic illustration of DNA-PAE@BAY-876-mediated TME remodeling and the associated immunostimulatory mechanisms.

3. In Figures 2 a and 2 b, color scales should be shown to understand what shades with different colors indicate.

A: Thank you for your concern. Based on your advice we have reprepared the two panels using NUPACK software and added the color scale. Meanwhile, the descriptions for the meaning of different colors have been added into the figure legends.

Figure 2. Physical and chemical characterization of DNA-PAE@BAY-876 nanoassemblies. (a-b) Secondary structures of PD-L1 and CTLA-4-antagonizing aptamers. Color intensity indicates base-pairing probability at equilibrium. (c-d) TEM image of DNA-PAE@BAY-876 nanoassemblies with different pH. (e) SEM image of DNA-PAE@BAY-876 nanoassemblies at pH7.4. (f) Size changes of DNA-PAE nanoassemblies before and after drug loading. (g) Zeta potential changes of DNA-PAE nanoassemblies before and after drug loading. (h) UV-vis

spectroscopic features of DNA-PAE nanoassemblies before and after drug loading. (i) Ultraviolet absorption of BAY-876 at different concentrations. (j) CMC of DNA-PAE nanoassemblies under pH 7.4. (k) Stability of DNA-PAE@BAY-876 nanoassemblies in PBS supplemented with 10% serum. (l) Critical pH value of DNA-PAE nanoassemblies showing their acidity-triggerable dissociation. (m) Size changes of DNA-PAE nanoassemblies at pH7.4 and 6.8. (n) Release kinetics of BAY-876 from DNA-PAE@BAY-876 nanoassemblies at pH7.4 and 6.8.

4 . Figure 2 e: Which PH condition does the SEM image correspond to?

A: Thank you for your advice. We are sorry for not mentioning the pH condition for the SEM images in Figure 2e, which was captured under the near neutral pH of 7.4. The related information has been added into Figure 2e.

Figure 2. Physical and chemical characterization of DNA-PAE@BAY-876 nanoassemblies. (a-b) Secondary structure of PD-L1 and CTLA-4-antagonizing aptamers. Color intensity indicates base-pairing probability at equilibrium. (c-d) TEM image of DNA-PAE@BAY-876 nanoassemblies with different pH. (e) SEM image of DNA-PAE@BAY-876 nanoassemblies at pH7.4. (f) Size changes of DNA-PAE nanoassemblies before and after drug loading. (g) Zeta potential changes of DNA-PAE nanoassemblies before and after drug loading. (h) UV-vis spectroscopic features of

DNA-PAE nanoassemblies before and after drug loading. (i) Ultraviolet absorption of BAY-876 at different concentrations. (j) CMC of DNA-PAE nanoassemblies under pH 7.4. (k) Stability of DNA-PAE@BAY-876 nanoassemblies in PBS supplemented with 10% serum. (l) Critical pH value of DNA-PAE nanoassemblies showing their acidity-triggerable dissociation. (m) Size changes of DNA-PAE nanoassemblies at pH7.4 and 6.8. (n) Release kinetics of BAY-876 from DNA-PAE@BAY-876 nanoassemblies at pH7.4 and 6.8.

5. Figure 4 b and 4 c, what does the “I II III IV V VI VII” mean?

A: Thank you for your reminder. We are sorry for the lack of detailed labeling for individual groups.

The related information has been added into the figure legends accordingly.

is 26 KDa; the molecular weight of Bax is 21 KDa; the molecular weight of Tubulin is 50 KDa. (d-e) Glucose uptake by 4T1 cells or T cells after different treatments under co-culture condition at 37°C for 24 h (n=3). (f-g) Flow cytometric analysis on the expansion of CD4+/CD8+ T cells, IFN- γ +CD8+ T cells in the co-incubation system after different treatment at 37°C for 24 h. (h) Statistical analysis of data shown in panel f-g by flow cytometry (n=3). (1) NC, (2) PAE, (3) aptCTLA-4, (4) aptPD-L1, (5) BAY-876, (6) DNA-PAE, (7) DNA-PAE@BAY-876.

6 . For western blotting, dashes should be added to locate the exact position of the marker bands indicated.

A: Thank you for your reminder. Dashes and labels have been added for relevant WB data in Figures 4 and 6 to enhance its informativity.

is 26 KDa; the molecular weight of Bax is 21 KDa; the molecular weight of Tubulin is 50 KDa. (d-e) Glucose uptake by 4T1 cells or T cells after different treatments under co-culture condition at 37°C for 24 h (n=3). (f-g) Flow cytometric analysis on the expansion of CD4+/CD8+ T cells, IFN-γ+CD8+ T cells in the co-incubation system after different treatment at 37°C for 24 h. (h) Statistical analysis of data shown in panel f-g by flow cytometry (n=3). (1) NC, (2) PAE, (3) aptCTLA-4, (4) aptPD-L1, (5) BAY-876, (6) DNA-PAE, (7) DNA-PAE@BAY-876.

Figure 5. Nanoassembly-enhanced 4T1 cell recognition and binding by aptPD-L1. (a) Schematic illustration on the nanoassembly-mediated metabolic rewiring of 4T1 cells for enhanced ICT. (b) Evaluation on the dose-dependent impact of BAY-876 on PD-L1 glycosylation in 4T1 cells

with pH7.4/6.8 at 37°C for 24 h. The concentration of IFN- γ was 1 $\mu\text{g}\cdot\text{mL}^{-1}$. Note: the molecular weight of PD-L1 is 33-70 KDa; the molecular weight of Tubulin is 50 KDa. (c) Impact of DNA-PAE@BAY-876 on the PD-L1 glycosylation level in multiple cell types at 37°C for 24 h. The concentration of DNA-PAE@BAY-876 was 100 nM and the concentration of IFN- γ was 1 $\mu\text{g}\cdot\text{mL}^{-1}$. (d) Impact of UDP-GlcNAc and F6P on the PD-L1 glycosylation levels in 4T1 cells after BAY-876 treatment with high/low-glucose media. The concentration of BAY-876, UDP-GlcNAc and IFN- γ was 100 nM, 0.1 mM and 1 $\mu\text{g}\cdot\text{mL}^{-1}$, respectively. (e) UDP-GlcNAc abundance in 4T1 cells under different conditions (n=4). (I, V) Control; (II, VI) BAY-876; (III, VII) BAY-876+F6P; (IV, VIII) BAY-876+UDP-GlcNAc. (f) Schematic illustration of Hexosamine Biosynthesis Pathway (HBP). (g) 3D plot on the correlation between UDP-GlcNAc and F6P with 2-NBDG after treatment by BAY-876 under graded concentrations. (h-k) Impact of BAY-876 on the competitive combination between aptPD-L1/PD1 with PD-L1 on 4T1 cell surface.

7. The authors should revise the language and correct grammatical mistakes as well as formal errors in the text.

A: Thank you for the reminder. Based on your advice we have thoroughly corrected the writing of this manuscript. **Some representative corrections are shown here below for your review:**

Linguistic corrections:

“Interestingly, immune checkpoint inhibition therapy (ICT), a form of immunotherapy that restores antitumor immunity by blocking negative immune checkpoints, has shown promise for effective TNBC treatment.”

“From a clinical perspective, TNBC presents a plethora of immunogenic traits...”

“Consequently, aptamer-based nanotherapeutics could be an alternative strategy to develop more advanced ICTs for overcoming the challenges in TNBC immunotherapy.”

“which may provide new insights on developing effective ICTs for TNBC treatment in the clinics.”

“To endow the PD-L1 and CTLA-4-targeting aptamers with self-assembly capacity and TME responsiveness...”

REVIEWERS' COMMENTS

Reviewer #1 (Remarks to the Author):

The authors have addressed all my concerns, and now it can be accepted for publication.

Reviewer #2 (Remarks to the Author):

I am satisfied with the reviewers response to my concerns regarding experimental clarity. Based on the changes made and new experimental data I would recommend this manuscript for publication.

Reviewer #3 (Remarks to the Author):

The author has well answered all questions raised by reviewers and supplemented the corresponding statistical analysis and supporting data. The work is meaningful. It is suitable for publication

Reviewer #4 (Remarks to the Author):

The revision has addressed most of my major concerns in the first round of the review and the manuscript is much improved. I have no further comments.